# Early contact between late farming and pastoralist societies in southeastern Europe

Sandra Penske[1 ✉], Adam B. Rohrlach[1,2], Ainash Childebayeva[1], Guido Gnecchi-Ruscone[1], Clemens Schmid[1], Maria A. Spyrou[1,3], Gunnar U. Neumann[1], Nadezhda Atanassova[4], Katrin Beutler[5], Kamen Boyadzhiev[6], Yavor Boyadzhiev[6], Igor Bruyako[7], Alexander Chohadzhiev[8], Blagoje Govedarica[8], Mehmet Karaucak[5], Raiko Krauss[9], Maleen Leppek[10], Igor Manzura[11], Karen Privat[12,13], Shawn Ross[14], Vladimir Slavchev[15], Adéla Sobotkova[16], Meda Toderaş[17], Todor Valchev[18], Harald Ringbauer[1], Philipp W. Stockhammer[1,10], Svend Hansen[5], Johannes Krause[1] & Wolfgang Haak[1 ✉]

Archaeogenetic studies have described two main genetic turnover events in prehistoric western Eurasia: one associated with the spread of farming and a sedentary lifestyle starting around 7000–6000 BC (refs. 1–3) and a second with the expansion of pastoralist groups from the Eurasian steppes starting around 3300 BC (refs. 4,5). The period between these events saw new economies emerging on the basis of key innovations, including metallurgy, wheel and wagon and horse domestication[6–9]. However, what happened between the demise of the Copper Age settlements around 4250 BC and the expansion of pastoralists remains poorly understood. To address this question, we analysed genome-wide data from 135 ancient individuals from the contact zone between southeastern Europe and the northwestern Black Sea region spanning this critical time period. While we observe genetic continuity between Neolithic and Copper Age groups from major sites in the same region, from around 4500 BC on, groups from the northwestern Black Sea region carried varying amounts of mixed ancestries derived from Copper Age groups and those from the forest/steppe zones, indicating genetic and cultural contact over a period of around 1,000 years earlier than anticipated. We propose that the transfer of critical innovations between farmers and transitional foragers/herders from different ecogeographic zones during this early contact was integral to the formation, rise and expansion of pastoralist groups around 3300 BC.

During the fifth and fourth millennia BC, key technological and social changes took place in southeastern Europe (SEE) which profoundly transformed prehistoric societies. Metal production was among the most important innovations; copper was mined, smelted and used to make axes, jewellery and small tools. The discovery of the necropolis of Varna (4600–4300 BC) on the Black Sea coast led to a reassessment of social inequality in human prehistory, with large quantities of gold and other symbols of power and wealth suggesting unprecedented levels of social stratification[10–12]. The many tell settlements that emerged during the Copper Age (CA, 4900–3800 BC) in SEE, involved in the proto-industrial exploitation of copper[13], gold and salt, highlight this advanced social organization and the blossoming of social, political, economic and artisanal activities. Eminent tell sites include Mound Măgura Gorgana near Pietrele on the Lower Danube in Romania[14], associated with the Gumelniţa culture and Tell Yunatsite in Bulgaria, associated with the Karanovo culture (Fig. 1 and Extended Data Fig. 1), which were occupied for several centuries[15]. From around 4600 BC, the similarity and continuous development of material culture and exchange of raw materials in the so-called Gumelniţa–Kodžadermen–Karanovo VI complex across southern Romania (Gumelniţa), northern Bulgaria (Kodžadermen) and Thrace (Karanovo) indicate transregional connectedness and suggest a relatively stable sociopolitical network. Consequently, the roughly simultaneous abandonment of the numerous tell settlements and cemeteries around 4250/4200 BC appears enigmatic (Fig. 1a,c). The underlying circumstances are unclear and might have involved the depletion of resources, the deterioration of soils and

[1]Department of Archaeogenetics, Max Planck Institute for Evolutionary Anthropology, Leipzig, Germany. [2]School of Computer and Mathematical Sciences, University of Adelaide, Adelaide, South Australia, Australia. [3]Institute for Archaeological Sciences, Eberhard Karls University of Tübingen, Tübingen, Germany. [4]Institute of Experimental Morphology, Pathology and Anthropology with Museum, Bulgarian Academy of Sciences, Sofia, Bulgaria. [5]Eurasia Department, German Archaeological Institute, Berlin, Germany. [6]National Archaeological Institute with Museum at the Bulgarian Academy of Sciences, Sofia, Bulgaria. [7]Odesa Archaeological Museum, Odesa, Ukraine. [8]Regional History Museum, Veliko Tarnovo, Bulgaria. [9]Institute for Prehistory, Early History and Medieval Archaeology, Tübingen, Germany. [10]Institute for Pre- and Protohistoric Archaeology and Archaeology of the Roman Provinces, Ludwig Maximilian University Munich, Munich, Germany. [11]National Museum of History of Moldova, Chişinău, Republic of Moldova. [12]Electron Microscope Unit, Mark Wainwright Analytical Centre, University of New South Wales, Sydney, New South Wales, Australia. [13]Earth and Sustainability Science Research Centre, School of Biological, Earth and Environmental Sciences, University of New South Wales, Sydney, New South Wales, Australia. [14]Department of History and Archaeology, Macquarie University, Sydney, New South Wales, Australia. [15]Varna Regional Historical Museum, Varna, Bulgaria. [16]Aarhus University, Aarhus, Denmark. [17]Institutul de Arheologie "Vasile Pârvan" Academia Română, Bucharest, Romania. [18]Yambol Regional Historical Museum, Yambol, Bulgaria. ✉e-mail: sandra_ellen_penske@eva.mpg.de; wolfgang_haak@eva.mpg.de

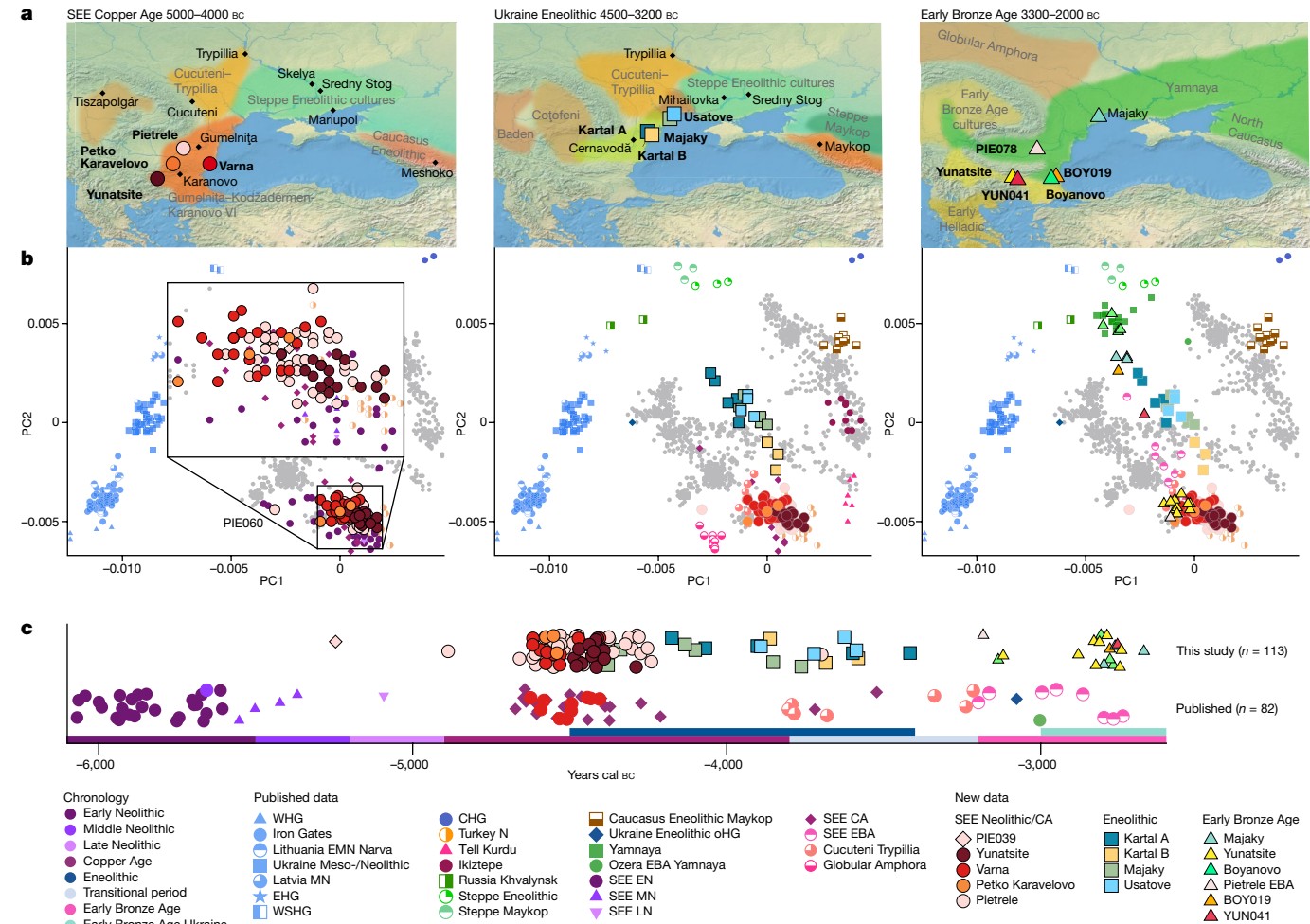

**Fig. 1 | Geographical locations, genetic analyses and chronology of newly reported ancient Copper Age, Eneolithic and Early Bronze Age individuals.**
**a**, Map of sites and relevant archaeological cultures discussed in the text. Maps were made with Natural Earth (https://naturalearthdata.com). **b**, PCA of newly reported individuals (coloured symbols with black outline) and relevant published groups (coloured symbols, no outline) projected onto the West Eurasian genetic variation of 1,253 individuals from 77 populations. **c**, Mean radiocarbon dates of relevant published and newly reported individuals from southeastern Europe plotted according to the regional chronology. The suffixes in the group labels present archaeological time periods and geographical regions: N, Neolithic; EN, MN, LN, Early, Middle, Late Neolithic, respectively; CA, Copper Age; EBA, Early Bronze Age; SEE, southeastern Europe; WHG, EHG, WSHG, CHG, oHG, Western, Eastern, West Siberian, Caucasus, outlier Hunter–Gatherers, respectively.

possibly also violent conflicts, as evidenced by the destruction horizon at Tell Yunatsite[16,17]. Historically, this demise was associated with the arrival of new groups from the steppe[18] but this proposal has lacked sufficient evidence. However, settlement activity over the following centuries was scarce in the entire western Black Sea region, indicating a 'dark' millennium with, for example, Yunatsite not being resettled until about 1,000 years later during the Early Bronze Age (EBA)[15].

Following the end of the CA, the centre of settlement activity shifted further northeast towards the forest–steppe region, where huge settlements, with thousands of houses, the so-called megasites of the Cucuteni–Trypillia complex (around 4100–3800 BC), emerged[19,20]. This northwestern Black Sea region represents an interaction zone between late CA farming-associated groups and those of the adjacent steppe region with different ecogeographic conditions. Continued innovations increased human mobility and the exploration of lands hitherto not amenable to agrarian lifestyles, as practised in the regions in SEE and south of the Caucasus for many millennia before. From the contact zones in the northwestern Black Sea region and the Caucasus, a gradual transition from foraging to semi-nomadic pastoralism also followed in the North Pontic region during the sixth and fourth millennium BC, triggered by continued innovations[6], transfer of

livestock and advances in herd management, food processing, dairying practices[8,9,19] and the development of arsenical-copper alloys[21]. The North Pontic region played a central role in the development of the oldest wheeled vehicles[22], while the North Caucasian Maykop culture was critical in the further development of metal alloys, as well as early horse domestication[8,9] and a sheep-wool economy combined with extensive dairy production[9]. The Maykop culture had extremely rich burials with metal weapons indicating 'high status' individuals, attesting to social inequality and upheaval during this time[23], as such social elites were also found in southern Romania and Bulgaria[24]. The Cernavodă I (around 4000–3200 BC) and Usatove cultures (3600/3500–3200/3100 BC) in the northwestern Black Sea region played a chief role in the east–west exchange between the Danube and the lower Dniepr[25] and these formations, while presumably indigenous, received strong contributions from the Trypillian tradition[26].

Similar to the SEE CA tell sites, the megasites and cultural phenomena of the northwestern Pontic region suddenly disappeared and were succeeded around 3300 BC by fully established pastoralists associated with the Yamnaya cultural complex. The expansion of North Pontic pastoralists to the west has been studied in many regions of Europe in recent years[27], whereas their emergence and impact on societies in

SEE is barely understood. This is relevant insofar as the archaeological record for the subsequent EBA (around 3200–2500 BC), indicates a concomitant rise in settlement activity for the first time since the demise of the CA settlements in the eastern Balkan region[28]. Burial mounds, associated with the Yamnaya cultural complex, appear frequently and extend along the Danube valley into the Carpathian Basin during the third millennium BC (refs. 4,29). By contrast, the resettlement of sites like Tell Yunatsite involved groups with burial rites not associated with the incoming steppe groups[30] (Extended Data Fig. 1c,h).

Archaeogenomic studies have shown that SEE CA individuals carry genetic profiles that resemble those of Neolithic farmers expanding from western Anatolia into Europe[31], distinct from both the earlier pre-agropastoralist (Western and Eastern Hunter–Gatherers; WHG/EHG) and later EBA pastoralist groups[1,2,4,5], who carried 'steppe' ancestry. Individuals from well-known, contemporaneous CA settlements (Pietrele and Yunatsite) and outstanding burial sites (Varna) provide a unique opportunity to study the genetic variation in and between sites at their peak settlement densities. However, the developments following early interactions, which had later given rise to the expansion of pastoralists and their genetic ancestry across Europe, remain unknown. Critically, individuals from the key period of the fifth and fourth millennium BC from the contact zone between SEE, the Trypillian megasites and the steppes have not been analysed genetically. Here, we address this spatial and temporal sampling gap by studying individuals associated with the Cernavodă I and Usatove cultures from the northwestern Black Sea region in today's Ukraine. Additionally, we analyse EBA individuals from the tell sites Yunatsite and Pietrele, following a possible resettlement of the sites after several centuries of abandonment. We compare these to Yamnaya-associated individuals from eastern Bulgaria, who were buried in mounds typically associated with steppe pastoralists during the third millennium BC and to individuals postdating the Usatove horizon in the northwestern Black Sea area.

In total, we report genome-wide data for 135 (out of 216 attempted) individuals from eight distinct sites (Fig. 1) ranging from around 5400 to 2400 BC: Neolithic ($n = 1$), CA ($n = 95$), Eneolithic ($n = 18$) and EBA ($n = 21$). All samples were enriched for a panel of 1.24 million single-nucleotide polymorphisms (1,240,000 SNP panel[32]), ranging from 61,000 to 947,000 SNPs with an average SNP coverage between 0.01× and 3.4×. We used a cut-off of 400,000 SNPs for hapROH and imputation and filtered for >550,000 SNPs for identity-by-descent (IBD) analyses (Supplementary Table A; Methods). We also report 113 new radiocarbon dates (Fig. 1c and Supplementary Table A). To assess the genetic ancestry and variation of the newly typed individuals we first performed principal component analysis (PCA) constructed from 1,253 modern-day West Eurasians from 77 different populations, onto which data from the ancient individuals were projected (Fig. 1b and Supplementary Table B; Methods).

## Neolithic and Copper Age ancestries

The earliest-dated individual in our dataset, PIE039 from Pietrele, falls in the expected range of other SEE Neolithic individuals in PCA space, with whom she also shares affinities according to outgroup $f_3$ statistics (Fig. 1b, Fig. 2 and Supplementary Table C). We used $f_4$ statistics of the form $f_4$ (test, PIE039; HGs, Mbuti), where 'test' are different Neolithic groups, to identify the genetically most similar Neolithic groups, which were then used as local proxies for quantitative ancestry modelling. We found Hungary_LN_Sopot and Malak Preslavets N to be most symmetrically related to PIE039 with respect to all HG comparisons ($|Z| \leq 1$) and thus combined them into local group SEE 1, which could be used as a single source for proximal qpAdm modelling ($P = 0.41$), confirming shared local ancestry (Extended Data Fig. 2, Fig. 3d, Supplementary Tables D, E, H and Supplementary Information 5).

In PCA space, the chronologically younger SEE CA individuals from the emblematic sites of Yunatsite (YUN), Varna (VAR), Pietrele (PIE)

and the multiple burial from Tell Petko Karavelovo (PTK), form a tight cluster that also overlaps with published Neolithic individuals from Anatolia and SEE[29] (Fig. 1b). Moreover, outgroup $f_3$ statistics suggest local genetic homogeneity throughout the CA in this region (Fig. 2 and Supplementary Table C). However, all SEE CA groups are slightly shifted towards the EHG/WHG cline in both PC1 and PC2 compared to most published Neolithic individuals. Distal qpAdm modelling (Fig. 3a and Supplementary Table G) confirmed minimal amounts of EHG-, CHG- and WHG-like ancestry, in addition to predominantly Turkey_N-like ancestry. This ancestry composition is already present during the Neolithic[29] and confirmed by the test $f_4$ (test, CA; HGs, Mbuti) in which Neolithic groups form a clade with SEE CA with respect to HG groups (Extended Data Fig. 2, Supplementary Tables D and E and Supplementary Information 5). This allows us to identify the best local Neolithic proxy for each SEE CA group and to account for the subtle differences in ancestries. Using the respective, locally preceding, Neolithic groups for proximal qpAdm modelling, we could model all SEE CA groups as a single-source model (Fig. 3d and Supplementary Table H), suggesting genetic continuity at the local scale.

The outlier individual PIE060 is shifted further towards the WHG/EHG cluster in PCA, suggesting an excess of this type of ancestry, which could be confirmed by $f_4$ statistics of the form $f_4$(SEE N, PIE060; HGs, Mbuti) (($|Z| \geq 3$); Supplementary Table F). Ancestry modelling with qpAdm supports a two-way model (Fig. 3d) with SEE N (around 65%) and Iron Gates HG or KO1 (around 35%) as the best proxies. Using DATES[33] to determine the time of admixture between SEE N and Iron Gates HG as a local HG ancestry, we obtained an admixture estimate of $16.3 \pm 13.4$ generations ($Z = 1.213$), which corresponds to around 81–832 years before the mean $^{14}$C date of PIE060, when a generation time of 28 years is assumed[34]. A flat decay curve (Extended Data Fig. 3a) supports the interpretation of a recent admixture date, which suggests that PIE060 came from a community outside Pietrele with recent contact with HGs. Indeed, individuals with similarly high amounts of HG ancestry have been reported from nearby sites in Malak Preslavets (around 70 km) and Dzhulyunitsa (around 140 km)[29].

In line with the autosomal data, the Y-chromosomal and mitochondrial DNA lineages are common in nearly all Neolithic and CA groups studied until now, albeit with several males also carrying typical Mesolithic (C1a and I2a) Y lineages[35], including individual PIE060 (Extended Data Fig. 3b and Supplementary Table A). With seven different main lineages among 29 males in Pietrele (I2a1, C1a, G2a, H2, T1a, J2a and R1b-V88), six among 15 males in Varna (I2a1, I2a2, G2a, T1a, E1b1 and R1b-V88) and four among six males at Yunatsite (C1a, G2a, H2, J2a), the Y-chromosomal diversity during the SEE CA was higher than in central/western Europe[36–38].

When testing for genetic relatedness in each of the SEE CA sites using READ, we detected only three first-degree and two second-degree relationships in total (Supplementary Table I; Methods). To specifically test for links between the contemporaneous SEE CA sites and for more distant genetic relatedness we explored signals of IBD sharing between individuals in and between all sites (Methods). We found no evidence for between-site links up to the fourth to fifth degree and only two pairs of individuals (PIE003-VAR010 and YUN005-VAR030) shared at least two blocks greater than 20 cM indicative of a fifth to seventh degree relationship (Extended Data Fig. 4a and Supplementary Table J). Integrating the normalized sum and number of shared blocks we find higher background relatedness at the intrasite level at Yunatsite and Varna compared to Pietrele, which can be explained by the structure of the sites (a destruction horizon of households and a burial ground with shorter use, respectively, versus tell and settlement burials spanning 350–400 years) (Fig. 1c and Extended Data Fig. 4b). However, analysis of the runs of homozygosity (ROH) per individual using hapROH indicates low levels of parental background relatedness suggesting relatively large effective population sizes, consistent with previous observation across early farming societies (Methods;

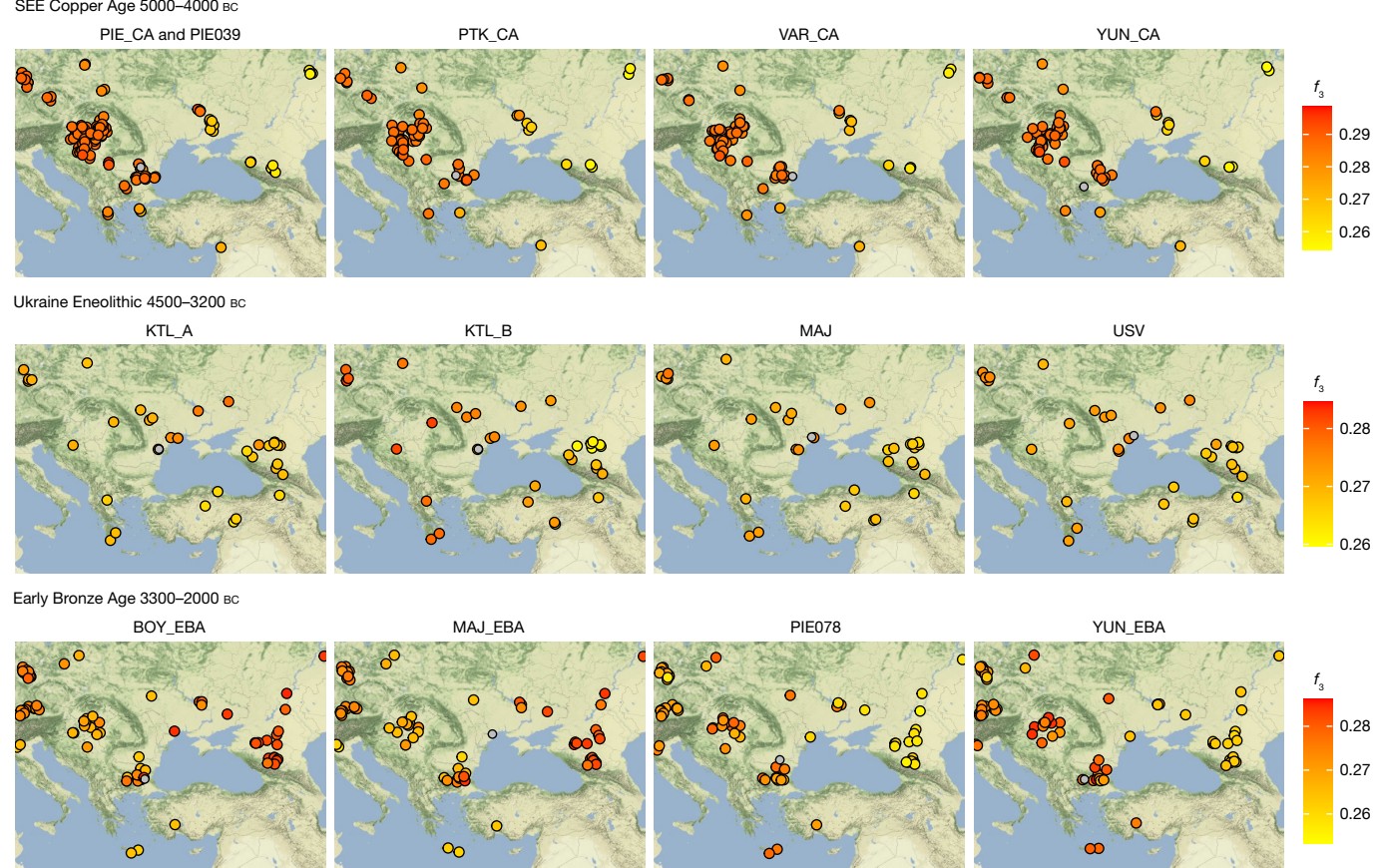

**Fig. 2 | Outgroup $f_3$ statistics for SEE CA, Ukraine Eneolithic and Early Bronze Age groups.** Outgroup $f_3$ statistics of the form $f_3$(test1, test2; Mbuti) plotted onto maps of central/SEE highlighting the shifting genetic affinities over time, separated temporally. Test1 includes groups and individuals newly reported in this study (headers) and their respective location is given by a grey circle. Test2 includes all relevant ancient populations from the respective time period (Supplementary Table C) and their locations are indicated as filled circles with black outlines. Higher $f_3$ statistics (red colours) indicate more shared drift with the respective group in Test1. All $f_3$ statistics, including outlier individuals and published ancient populations further west, can be found in Supplementary Table C. The maps were made in R[57] and the tile sets are copyright of Stamen Design, under a Creative Commons Attribution (CC BY 3.0) licence.

Extended Data Fig. 5). These findings reflect the settlement density and the wide-spread cultural, rather than close genetic, connectedness of the Gumelnița–Kodžadermen–Karanovo VI complex, in line with the cross-regional significance of SEE tell sites[26].

## Early contacts during the Eneolithic

Eneolithic individuals from Ukraine (Ukraine Eneolithic), dated from around 4500–3500 BC, associated with the Cernavodă I and Usatove cultures, form a genetic cline in PCA space (Fig. 1b) between Neolithic/ SEE CA individuals and published Eneolithic steppe individuals from the North Caucasus[39] and Khvalynsk in western Russia[32]. This indicates possible admixture between CA farmer-related groups and Eneolithic steppe groups, as in line with cultural interactions described in the archaeological record[40–42]. The observed genetic cline reflects developments over a wide chronological range of around 1,000 years (Fig. 1c and Supplementary Table A). Some of the newly reported [14]C dates could be affected by a freshwater reservoir effect[43], common in Steppe Eneolithic sites[44,45] and could therefore be several centuries younger than their reported dates. However, accounting for this possibility, an offset of around 500 years would still date most of the Ukraine Eneolithic individuals to the fourth millennium BC and thus considerably earlier than the Yamnaya-associated steppe pastoralist expansion.

Individuals from Kartal (around 4150–3400 BC), associated with the Cernavodă I culture, are genetically highly heterogeneous, with five individuals (Kartal A) forming a cline between 'Steppe Eneolithic'/'Steppe Maykop' individuals and Early Neolithic groups, while three other individuals (Kartal B) fall closer to the latter (Supplementary Tables L and M). The five contemporaneous individuals from Majaky (MAJ), are genetically more homogeneous and fall together with the four individuals from the late Eneolithic Usatove type-site (USV/UBK; Supplementary Table A) in the middle of the 'Kartal cline'. We tested for a correlation between positions of the Ukraine Eneolithic individuals in PC2 and their [14]C dates and found none (Spearman's $\rho = 0.113, P = 0.6656$). The broadscale shift in genetic affinities between the CA and the Eneolithic, from SEE to the steppe zone, is also clearly visible in outgroup $f_3$ statistics when mapped geographically (Fig. 2 and Supplementary Table C).

To formally characterize the Ukraine Eneolithic individuals, we tested for excess shared ancestry with four Holocene 'cornerstone' populations (Turkey_N, WHG, EHG/WSHG and CHG) (Supplementary Information 1.2), using $f_4$-symmetry statistics of the form $f_4$(test, Ukraine Eneolithic; cornerstone, Mbuti) and conditioning on three test populations (Extended Data Fig. 6, Supplementary Tables E, M and N and Supplementary Information 6). First, compared to Turkey_N, Ukraine Eneolithic individuals show excess affinity to all HG groups, as indicated by significantly negative $f_4$ statistics ($|Z| \geq 3$) (Extended Data Fig. 6a). Second, conditioning on Steppe Eneolithic (Extended Data Fig. 6b), we observe excess affinity of Ukraine Eneolithic to Turkey_N, a symmetrical relatedness to CHG and WHG, while Steppe Eneolithic

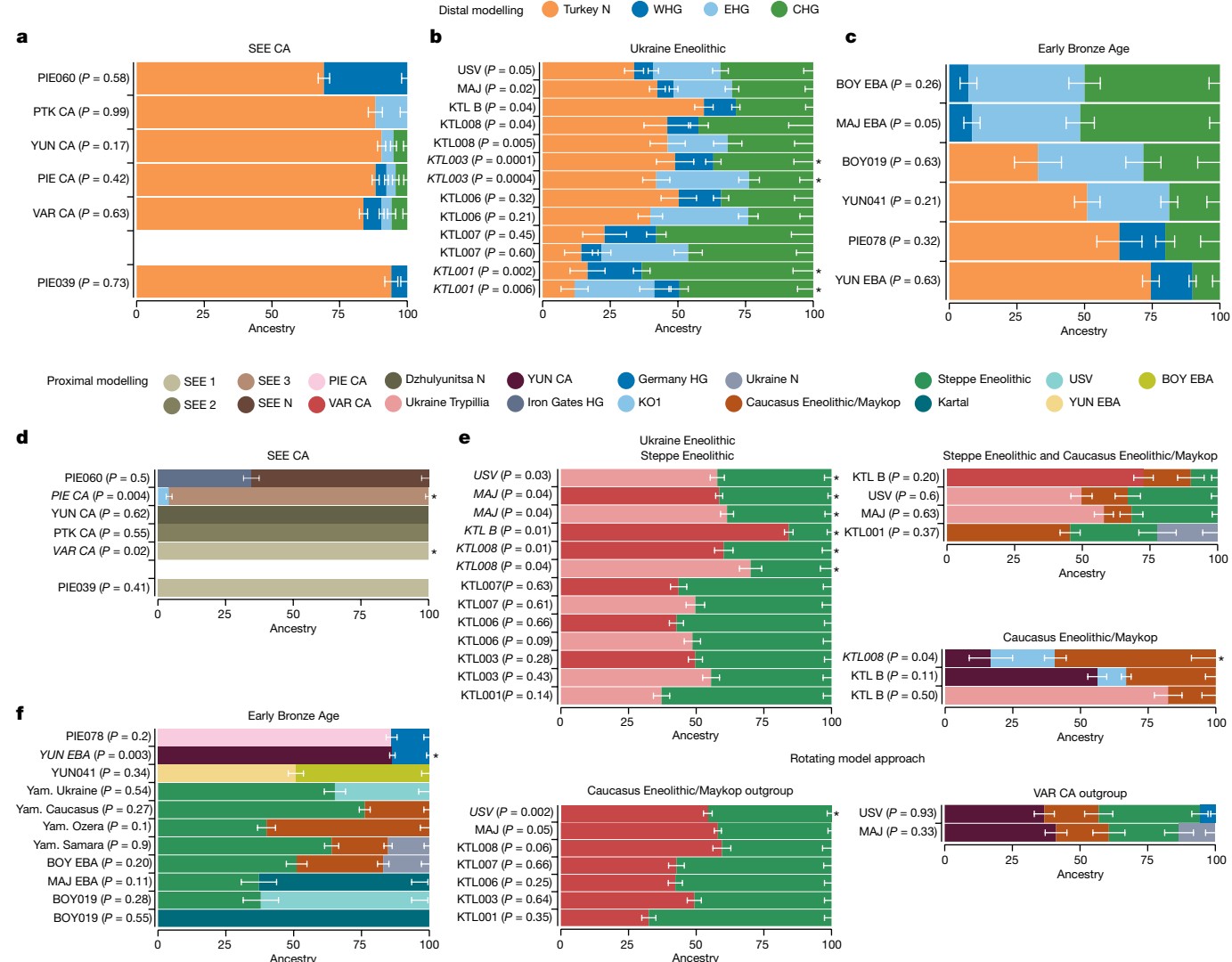

**Fig. 3 | Distal and proximal qpAdm results for the Copper Age, Ukraine Eneolithic and Early Bronze Age groups. a–c**, Distal models with Turkey_N, WHG, EHG and CHG as sources for the three sampled time periods: SEE CA (**a**); Ukraine Eneolithic (**b**); and Early Bronze Age (**c**). **d–f**, Geographically and temporally proximal models of the three sampled time periods: SEE CA (**d**); Ukraine Eneolithic (**e**); and Early Bronze Age (**f**) (Supplementary Tables G, H, P, Q, V and W). All results shown here were run with the parameter 'allSNPs: NO' (Supplementary Information 6). * Indicates non-supported/rejected/failed models when applying a $P$ value cut-off of less than 0.05 (shown in italics). Ancestry proportions are shown with one standard error. Standard errors were computed with the default block jackknife approach.

groups carry more EHG/WSHG ancestry. On the basis of cultural influences which also link the northern Black Sea through the steppe belt to the North Caucasus region[40–42], we also test for potential influence of North Caucasian groups. Using Caucasus Eneolithic/Maykop as test (Extended Data Fig. 6c) we find excess affinity of Ukraine Eneolithic to EHG and WHG and Turkey_N, while Caucasus Eneolithic/Maykop share more drift with CHG.

The archaeological record identifies the northwestern Black Sea region as an interaction zone between late CA farming and groups from the steppe region[19,20]. Such an early interaction has been postulated by Immel et al.[46], who have reported Yamnaya-related ancestry in individuals associated with the Cucuteni–Trypillia complex from today's Moldova. However, on re-analysis of these data we find that this signal can be explained solely by an increase in especially EHG-rich ancestry (Supplementary Information 4 and Supplementary Table O).

To characterize the role of Cernavodă I and Usatove-associated individuals from the postulated interaction zone, who show a clear signal of admixture, we formally tested the contribution of diverse ancestry sources using $f_4$ (Steppe Eneolithic/Caucasus Eneolithic/

Maykop, Ukraine Eneolithic; test, Mbuti), where test represents SEE and Anatolian CA farmer groups (Supplementary Table P). With respect to Steppe Eneolithic, all Ukraine Eneolithic individuals show excess affinity to all tested CA groups. With respect to Caucasus Eneolithic/Maykop, USV, MAJ, KTL_B, KTL003 and KTL008 show excess affinity to all SEE CA farmer groups, while KTL006 and KTL007 only share drift with Ukraine Trypillia (Supplementary Table P).

Of note, all $f_4$-symmetry tests with Caucasus Eneolithic/Maykop and SEE CA indicate an additional attraction of Ukraine Eneolithic to WHG/EHG (Supplementary Table N), with Iron Gates HG or Ukraine N showing the highest affinity (Supplementary Table Q). This affinity towards WHG/EHG is absent when Steppe Eneolithic is used (Supplementary Table N), implying that scenarios involving potential gene flow from the Caucasus would require an additional source carrying WHG-/EHG-like ancestry as this ancestry is not sufficiently represented by SEE CA or Caucasus Maykop groups.

Using distal qpAdm modelling we find support for a four-way admixture of Turkey_N, EHG, CHG and WHG for KTL001, KTL007, MAJ and USV (Fig. 3b and Supplementary Table R), while individuals KTL003,

KTL006 and KTL008 can be modelled alternatively with three sources (Turkey_N+EHG+CHG) and KTL_B individuals only with Turkey_N (around 60%), CHG (around 28%) and WHG (around 12%) ancestry. Following up with proximal qpAdm models to explore potential contribution(s) of temporally and geographically closer groups (Fig. 3e and Supplementary Tables E and S), we find that all Ukraine Eneolithic individuals can be modelled as a two-way model of either VAR_CA or Ukraine Trypillia as farmer-related ancestry source and Steppe Eneolithic as a source of mixed EHG+CHG ancestry.

Since archaeological research suggests a cultural contribution of Steppe Eneolithic and Maykop groups (Supplementary Information 2.2), we specifically tested for alternative scenarios which involved admixture between both groups north of the Caucasus and subsequent spread westwards. Using both associated ancestries and different HGs and SEE CA-related groups as sources in qpAdm modelling (Fig. 3e and Supplementary Table S), we find that KTL001 can indeed be modelled as a three-way mixture of Steppe Eneolithic (around 32%), Caucasus Eneolithic/Maykop (around 46%) and Ukraine N foragers (around 22%), to the exclusion of a SEE CA source. By contrast, MAJ and USV can be modelled as VAR_CA or Ukraine Trypillia (around 50%), Steppe Eneolithic (around 35%) and Caucasus Eneolithic/Maykop (around 15%) as minor third component. KTL_B results in the same model but with a higher VAR_CA component (around 73%) and a minor contribution of Steppe Eneolithic (around 10%) ancestry (Supplementary Table S).

Exploring an alternative scenario which excludes Steppe Eneolithic as a source, we find a well-fit model for KTL008 with YUN_CA (around 17%), Caucasus Eneolithic/Maykop (around 60%) and KO1 (around 23%). Further, KTL_B can be modelled with Ukraine Trypillia (around 82%) and Caucasus Eneolithic/Maykop (around 18%) as second source, which is consistent with the omission of EHG ancestry in the distal qpAdm results for KTL_B (Fig. 3e and Supplementary Table S).

Finally, to test whether we can distinguish between the farmer-related ancestry contributed by SEE CA- or Maykop-associated groups from the Caucasus, or by both, we rotated each source to the outgroups, alternatingly, keeping Steppe Eneolithic as a constant. Here, we find strong support for a genetic contribution from SEE CA rather than Caucasus Eneolithic/Maykop for most KTL individuals (except KTL_B), which can be modelled as Steppe Eneolithic and VAR_CA (Fig. 3e and Supplementary Table S). The same model is supported for MAJ ($P = 0.05$) but rejected for USV, which indicates that Maykop-associated ancestry is needed for the latter. Indeed, the competing model, with Maykop as an additional source and VAR_CA as an outgroup, results in a well-fit four-way mixture model for USV ($P = 0.93$) and improved model fit for MAJ ($P = 0.33$), whereas the models for the remaining KTL individuals are rejected (Supplementary Table S). This provides strong support for an alternative admixture history for USV and MAJ, involving local SEE CA, Steppe Eneolithic, Caucasus Eneolithic/Maykop and a HG-related source, a combination that is distinct from KTL individuals.

The similarities in genetic ancestry presented for MAJ and USV are also observed in the results from the IBD analysis (Extended Data Fig. 4a and Supplementary Table J) in which we find a fourth to sixth degree relationship between MAJ023 and USV006, which reflects the close geographical vicinity of the two sites. The normalized sum and number of shared blocks for Ukraine Eneolithic show a higher background relatedness in USV compared to the other sites (Extended Data Fig. 4b) but also between USV and MAJ and USV and KTL, respectively, which matches the relative chronological overlap of the three sites (Fig. 1c and Supplementary Table A). However, in comparison to the preceding CA and heterogenous KTL individuals, ROH indicate a slightly elevated parental background relatedness for MAJ and USV (Extended Data Fig. 5), suggesting smaller effective population sizes in Usatove-associated groups.

Y-chromosomal evidence from the six Ukraine Eneolithic males reflects lineages from each of the contributing sources (Extended Data Fig. 3b): G2a is probably a Neolithic legacy, while three males

carrying I2a1 could be attributed to the local Ukrainian Neolithic or HG groups in general. KTL005 and MAJ009 carry haplotypes R1b/M343(×P297) and R1b1/L754(×M269), respectively, which are ancestral for the pre-M269 branch (P297) and the M269 branch. Importantly, we do not observe R1b-Z2103 or immediate R1b-M269 precursor lineages, which originated in the steppe and are later linked with expansion of steppe-related ancestry.

## Genetic ancestries during the Bronze Age

The EBA individuals in this study are characterized by two contrasting clusters of genetic ancestry in PCA space (Fig. 1b) and different genetic affinities in outgroup $f_3$ statistics (Fig. 2 and Supplementary Table C). Individuals from YUN and individual PIE078, who date to the first half of the third millennium BC, resemble the SEE CA groups, whereas BOY_EBA and MAJ_EBA individuals fall within the 'steppe ancestry' cluster, commonly associated with the Yamnaya cultural complex. Two outlier individuals, BOY019 and YUN041, fall in the space between. Intriguingly, the males from YUN_EBA/PIE078 carried Y-chromosome lineages I2a, suggestive of a HG legacy, while the males from BOY/MAJ_EBA carried R1b-Z2103 or derived lineages, a characteristic hallmark of Yamnaya-associated ancestry (Extended Data Fig. 3b).

On the basis of these observations we tested for additional attraction towards HG-related groups in YUN_EBA and PIE078 compared to their CA predecessors by using $f_4$(CA, EBA; HGs, Mbuti) and confirmed the excess HG ancestry in EBA individuals from YUN and PIE with significant negative results ($|Z| \leq 3$) (Extended Data Fig. 7 and Supplementary Table T). By contrast, for MAJ_EBA, BOY_EBA, BOY019 and YUN041, we tested for additional attraction towards farmer-related groups represented by VAR_CA when compared to Yamnaya-associated groups (test) using $f_4$(test, EBA, VAR_CA, Mbuti) (Supplementary Table E). Here, only the outlier individual YUN041 has a higher affinity to VAR_CA than to other EBA groups (Extended Data Fig. 8 and Supplementary Table U). Distal qpAdm modelling with cornerstone populations confirms the contrasting ancestries of the two main EBA clusters. PIE078 and YUN_EBA can be modelled with Turkey_N, CHG and WHG (Fig. 3c and Supplementary Table X), whereas MAJ_EBA, BOY_EBA, BOY019 and YUN041 require EHG ancestry as an additional source (Fig. 3c).

We then explored the apparent homogeneity of Yamnaya-associated EBA steppe pastoralist groups, by testing for possible contribution(s) from four sources: Ukraine Eneolithic as a proxy for mixed Turkey_N/CHG/EHG ancestry, Ukraine N as an HG-related group, Steppe Eneolithic as pre-Yamnaya genetic substrate and Caucasus Eneolithic/Maykop as a proxy for mixed Turkey_N/CHG-related South Caucasus ancestry, as suggested by ref. [47] and directly supported by our results for the preceding Eneolithic period. First, we formally tested for shared drift between all EBA Yamnaya-associated individuals and Steppe Eneolithic/Caucasus Eneolithic/Maykop with respect to cornerstone populations by using $f_4$(Steppe Eneolithic/Caucasus Eneolithic/Maykop, EBA; cornerstones, Mbuti). With the exception of Yamnaya Caucasus, all EBA individuals show an excess affinity to Turkey_N when compared to Steppe Eneolithic (Extended Data Fig. 9 and Supplementary Table V). Further, when compared to Caucasus Eneolithic/Maykop all EBA individuals share drift with WHG and EHG/WSHG and only YUN041 is also significant for Turkey_N (Extended Data Fig. 9 and Supplementary Table V). Second, we used $f_4$-symmetry statistics of the form $f_4$(steppe1, steppe2; test, Mbuti) where test includes Ukraine N, Ukraine Eneolithic, Caucasus Eneolithic/Maykop and Steppe Eneolithic. Here, with the exception of outlier individual Ukraine_Ozera_EBA_Yamnaya, all $f_4$ statistics are non-significant ($|Z| \leq 3$) (Supplementary Table W), which indicates that all Yamnaya-associated individuals including those from Ukraine and Bulgaria are genetically highly similar.

Applying the same rationale and sources to proximal qpAdm modelling to uncover subtle signals (Fig. 3f and Supplementary Table Y), we find that BOY_EBA and Yamnaya Samara can be modelled as a three-way

mixture of Steppe Eneolithic, Caucasus Eneolithic/Maykop and Ukraine N. We note that the same three sources contributed to the preceding Ukraine Eneolithic individuals from USV and MAJ (in addition to SEE CA ancestry), which suggests that similar processes had led to the tripartite ancestry formation in the steppe zone during the fourth millennium BC. Indeed, we find that BOY_EBA, MAY_EBA and Yamnaya Samara can also be modelled as a two-way mixture of Steppe Eneolithic and KTL001 (who lacked SEE_CA ancestry). For Ukraine_EBA_Yamnaya, we find support for a three-way model ($P$ = 0.07) with Steppe Eneolithic (around 75%), Caucasus Eneolithic/Maykop (around 14%) and Globular Amphora (around 11%) as a western source but also improved model fit ($P$ = 0.5) for a two-way mixture of Steppe Eneolithic (around 65%) and USV (around 35%) (Supplementary Table Y), which suggests a possible direct contribution of Ukraine Eneolithic groups to steppe pastoralists in the third millennium BC. By contrast, Yamnaya Caucasus individuals from the southern steppe can be modelled as a two-way model of around 76% Steppe Eneolithic and 26% Caucasus Eneolithic/Maykop, confirming the findings of Lazaridis and colleagues[47]. This two-way mix (40% + 60%, respectively) also provides a well-fit model ($P$ = 0.09) for the Ozera outlier individual, consistent with the position in PCA and corroborating an influence from the Caucasus. Despite the overlap in PCA, these results suggest subtle geographical structure, involving local genetic strata and influences from neighbouring groups in western and southern contact zones, respectively. Individual BOY019 can be modelled successfully with around 63% USV and around 37% Steppe Eneolithic ancestry or around 40% Ukraine Trypillia and around 60% Steppe Eneolithic, suggesting interaction between these two neighbouring groups in the western contact zone or alternatively direct descent from admixed groups (for example, KTL001). Finally, individual YUN041 can be modelled as around 50% local YUN_EBA ancestry and 50% of either BOY_EBA or another Yamnaya-associated source.

## Discussion

The genetic homogeneity observed in and across the four CA sites (PIE, YUN, PTK and VAR) of the fifth millennium BC matches the cultural homogeneity of the archaeological records and suggests an extended period of a relative stable sociopolitical network and absence of large-scale cultural and genetic transformations. Shared shorter IBD tracts between sites are consistent with the transregional connectivity visible in the material culture. We can only speculate about the reasons that led to decreasing settlement densities at the end of the CA. Conflict arising from an early expansion of supposedly 'Indo-European' groups from the steppe, an idea that was put forward by M. Gimbutas[18], is possible but internal competition and strife between CA groups is equally likely. In fact, given the near-identical genetic ancestry profiles of SEE CA groups, we caution that genetic analyses would be blind to internal conflicts, causing the replacement of one CA group by another. Long-lasting droughts and forest fires[16] or infectious diseases and ensuing epidemics are other factors that could deplete lands. Indeed, evidence for early forms of *Yersinia pestis* as old as 5,000 years has been reported[48–50] and even further back in time for *Salmonella enterica*[51] for individuals associated with transitional foraging and pastoralism. Despite the systematic screening of teeth, we found no evidence for pathogens among the CA individuals of the fifth and fourth millennium BC, apart from two individuals (YUN048 and VAR021), who were positive for the Hepatitis B virus (HBV)[52], while individual VAR021 was also positive for *Salmonella enterica*.

A principal finding from our study indicates early contact and admixture between CA farming groups from SEE and Eneolithic groups from the steppe zone in today's southern Ukraine, possibly starting in the middle of the fifth millennium BC when settlement densities shifted further north, connecting the lower Danube region with the coastal steppe and Cucuteni–Trypillia groups of the forest–steppe. Archaeological evidence shows that the early CA Gumelniţa groups had already settled deep into the steppe zone by the mid-fifth millennium BC, introducing elements of a farming lifestyle but also carrying cultural influences from local HG groups[53]. The succeeding Cernavodă I and Usatove archaeological cultures were heavily influenced by local CA cultures and surrounds. During the fourth millennium BC, the northwestern Pontic region experienced intensified contact with Steppe Eneolithic groups, while these in turn also had contact with groups in the North Caucasus, such as Maykop, all of which are mirrored by the genomic data presented here. Moreover, despite the close geographical proximity of the Ukrainian sites studied, we were able to trace different admixture histories. Here, the heterogeneity of the individuals from the site Kartal stands out, which is located on the Danube delta at the northern end of the former distribution of the Chalcolithic Gumelniţa–Kodžadermen–Karanovo VI complex and thus represents the transformative nature and dynamics of the fourth millennium BC in action. By contrast, the more homogenous Majaky and Usatove groups, located north of the Dniester River, show that such assimilation processes had already occurred, suggesting that contact and exchange between transitional foragers and early pastoralist groups from the forest–steppe zone and non-local SEE farmer-associated groups had started already in the late fifth millennium BC. Moreover, variable cultural influences attested by the archaeological record[40,41,53] are also traceable genetically. We argue that livestock, innovations and technological advances were exchanged through these zones of interaction, which then led to the establishment of fully developed pastoralism in the steppe by the end of the fourth millennium BC. Gene flow from both contact zones into the steppe could also explain the small amounts of farmer-related ancestry in the emerging Yamnaya pastoralists, which differentiates them from the Steppe Eneolithic substrate and accounts for subtle geographical structure in the vastly expanding territory/range.

The early admixture during the Eneolithic presented in this study appears to be local to the northwestern Black Sea region of the fourth millennium BC and did not affect the hinterland in SEE. In fact, EBA individuals from the fourth and third millennia BC from YUN and PIE do not show traces of steppe-like ancestry but instead a resurgence of HG ancestry observed widely in Europe during the fourth millennium BC (refs. 4,29,54,55). This indicates the presence of remnant HG groups in various non-farmed regions, for example, highlands and uplands or densely forested zones and wetlands and a mosaic of ancestries rather than a genetically uniform CA and EBA Europe.

While only a few tell sites have been resettled by local and/or incoming groups who did not originate in the North Pontic region, we can trace the appearance of migrants from the steppe, clearly attributed to Yamnaya culturally and genetically, in the local time transect at Majaky but also at Boyanovo in the Bulgarian lowlands of the Thracian Plain. The subtle differences in genetic ancestries between these two when compared to different Yamnaya-associated groups account for their geographical locations and different stages of genetic and perhaps, cultural assimilation. Two outlier individuals from EBA YUN and BOY bear witness to occasional admixture between inhabitants of EBA tells and incoming steppe pastoralists. Ultimately, the third millennium BC form of 'steppe'-ancestry is expected to have reached the Great Hungarian plain, from where it diversified and spread further west. The interaction between local and incoming groups in SEE did not result in archaeologically visible conflicts or a near-complete autosomal genetic turnover as observed in Britain or a replacement of the Y-chromosome lineages in the Iberian Peninsula[36,56].

Further integrated archaeogenomic studies are needed to disentangle the dynamics at play around the Black Sea during the formative periods of the admixture clines demonstrated in this study. High-quality genome-wide data from the fifth and fourth millennia BC that allow the direct tracing of IBD blocks shared by contributing groups will hold the key to understanding the population history of West Eurasia.

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

## Methods

### Permission statement

Permission to work on the archaeological samples was granted by the respective excavators, archaeologist and curators and museum directors of the sites, who are co-authoring the study.

### Radiocarbon dating

Of the 135 individuals reported in this study we obtained new direct [14]C dates for 113 individuals. Radiocarbon dating was carried out using accelerated mass spectrometry at the Curt-Engelhorn-Zentrum Archäometrie gGmbH in Mannheim, Germany (Fig. 1c and Supplementary Table A). All samples were calibrated on the basis of the IntCal20 database and using OxCal v.4.4.2. All [14]C dates in this study are consistent with the archaeological chronology based on stratigraphy and grave goods. We also included 11 published, direct [14]C dates for individuals from Varna[58–60] (Fig. 1c and Supplementary Table A).

### Ancient DNA laboratory procedures

Ancient DNA work was carried out in dedicated clean room facilities of the Max Planck Institute for Evolutionary Anthropology (MPI-EVA), Leipzig and Jena, Germany. We processed 168 petrous bones and 129 teeth in total. Petrous bones were sampled with a minimal invasive method[61] and, for the sampling of the teeth, the crown was separated from the root and the inner pulp chamber was drilled out[62]. DNA was extracted from all samples following a modified protocol refs. 63,64. DNA double-stranded libraries were built using a partial uracil-DNA-glycosylase (UDG-half) treatment[65]. For samples that did not meet the threshold for further analysis, we attempted to increase the DNA yield by using an automated protocol for producing single-stranded, non-UDG libraries[66,67]. All libraries were double-indexed with a unique pair of indices[68].

First, all indexed libraries were screened by means of shotgun sequencing of 5 million reads on an Illumina HiSeq4000 or NextSeq500 sequencing platform using a single end (1 × 75 base pair (bp) reads) kit, followed by an assessment of human DNA content and DNA damage profiles (initial quality criteria). Libraries above the threshold of 0.1% endogenous DNA were enriched for around 1.2 million SNPs in a targeted in-solution capture (1,240,000 SNP capture)[31]. Enriched libraries were sequenced on HiSeq4000 and NextSeq500 Illumina platforms using a single-read (SR 75) kit and sequencing 40 million reads for libraries between 0.1% and 2% or 20 million reads for libraries above 2%, resulting in a mean coverage of 0.7× (Supplementary Table A). An additional mitochondrial capture[4,69] was performed for individuals for which enough reads could not be obtained as by-catch of the 1,240,000 capture, resulting in an average coverage of 64×. For selected male individuals we also performed an inhouse capture assay for the Y chromosome (YMCA)[70] which targets around 10.445 kB on the non-combining region of the Y chromosome and which resulted in a mean coverage of 0.08×. Together, a total of 135 individuals yielded sufficient genomic data for downstream analysis.

### Sequence data processing

After demultiplexing, EAGER v.1.92.56 (ref. 71) was used to process raw ancient DNA sequence data. Raw reads were trimmed for Illumina adaptor sequences using AdapterRemoval v.2.3.0 (ref. 72). Subsequently, reads were mapped to the human reference genome hs37d5 using BWA v.0.7.12 (ref. 73) and duplicates were removed using DeDup v.0.12.1 (ref. 71). To analyse characteristic DNA damage in the form of G to A and C to T substitutions, mapDamage v.2.0.9 (ref. 74) was used. The effect of postmortem DNA damage on genotyping was minimized by removing 2 bp from the 3′ and 5′ ends of reads from double-stranded UDG-half-treated libraries ($n = 131$) using the trimbam function included in bamUtils v.1.0.13 (ref. 75). The resulting filtered bam files were genotyped with pileupCaller v.1.4.0.2 (ref. 76) by randomly calling one allele per position considering the human genome as a pseudohaploid genome (–randomHaploid). Only for quality controls 10 bp were removed from the 3′ and the 5′ ends for non-UDG treated single-stranded libraries, whereas the untrimmed bam files were treated with the –singleStrandMode in pileupCaller for genotyping. Coverage statistics calculations and bam filtering were done using samtools (v.1.3; ref. 77).

### Ancient DNA authentication

All libraries, except PTK001, yielded damage patterns characteristic of ancient DNA, which includes short DNA fragment lengths (45–65 bp on average) and postmortem deamination at the end of the molecules (6–17% for partial UDG treatment, 30–38% for non-UDG treatment). We merged Shotgun, 1,240,000 and mitochondrial capture data for each individual, mapped this against the revised Cambridge Reference Sequence for the complete human mitochondrial genome (NC 012920.1) and estimated contamination for both sexes on the mitochondrium using ContamMix[78] (Supplementary Table A), ranging from 0.086% to –9.2%. The nuclear contamination for males was estimated using ANGSD[79] and ranged from 0.2% to 2%. PTK001 yielded a contamination estimate of around 18% and therefore was excluded from all further analysis. We estimated the genetic sex by calculating the coverage on the X, Y and the autosomal chromosomes, for which the X and Y coverage is normalized by the autosomal coverage and the relative length of each sex chromosome[80].

### DNA reference datasets

The new genotype data were restricted to two sets of reference panels, the Affymetrix Axiom Genome-wide Human Origins1 array (HO; 593,124 autosomal SNPs)[2,81] and the 1,240,000 panel (1.233,013 autosomal SNPs including all of the HO SNPs)[31]. The number of SNPs covered at least once for each of these reference panels is given in Supplementary Table A.

### Genetic relatedness analysis

Genetic relatedness was estimated using READ[82], using default parameter settings. Background relatedness was estimated using the median value, across all sites per temporal group (Supplementary Information 7). From pairs of first-degree relatives, the individual with lower number of SNPs on the 1,240,000 target region was excluded from downstream analysis. Three individuals from PIE were identified as identical and were therefore merged for downstream analysis. Two pairs of the newly published samples from YUN CA had to be merged as they were revealed to be the same individuals. One individual from VAR and one from YUN were merged with previously published individuals from each site because they were sampled from the same individual and therefore identical[29] (Supplementary Table I).

### Assignment of uniparentally inherited haplogroups

Trimmed Shotgun, 1,240,000 and mitochondrial capture reads were aligned to the revised Cambridge Reference Sequence for the complete human mitochondrial genome (NC 012920.1) and a consensus sequence for each individual was retrieved using Geneious v.2019.2.3 (ref. 83). HaploGrep2 (v.2.4.0; ref. 84) was used to assign each consensus sequence to a specific mitochondrial haplogroup (Supplementary Table A). Y-chromosome haplogroups for all male individuals were assigned using the manual assignment method of Y-haplogroup calling as described in ref. 70 (Supplementary Table A). In the case of non-UDG treated sequence, YMCA data were filtered to exclude C to T and G to A transitions on the forward and reverse strands, respectively.

### Population genetic analysis

For genome-wide analyses the new data from this study were merged with published ancient and modern data from the Allen Ancient DNA

Resource (AADR) v.44.3 (https://reich.hms.harvard.edu/allen-ancient-dna-resource-aadr-downloadable-genotypespresent-day-and-ancient-dna-data). Data on the HO panel (around 600,000 SNPs) were used for PCA using the program 'smartpca' v.16000 (EIGENSOFT[85]). Principal components were computed for 1,253 present-day western Eurasians from 77 different populations (Supplementary Table B) on which ancient individuals were projected, using the options 'lsqproject: YES' and 'shrinkmode: YES'. Individuals with fewer than 30,000 SNPs on the HO-dataset covered were excluded from the PCA. All other analyses were performed on the above merged dataset on the 1,240,000 SNP panel (around 1.24 million SNPs). Outgroup $f_3$ statistics[86] were calculated using qp3Pop to obtain the genetic relatedness of a target population to a set of ancient Eurasian populations since the divergence from an African outgroup. The $f_4$ and $f_3$ statistics were calculated using qpDstat and the f4mode: YES function. Standard errors were computed with the default block jackknife approach and 3 s.e. are reported and plotted. The $f_3$ and $f_4$ statistics were calculated using the ADMIXTOOLS[81] package.

### Genetic admixture modelling
Ancestry modelling and ancestry proportion estimation on the 1,240,000 SNP dataset was performed using qpAdm in ADMIX-TOOLS (v.5.1; ref. 4). The following groups were used as a basic set of outgroups for distal modelling: Mbuti.DG, Turkey_Epipaleolithic, Iran_GanjDareh_N, Russia_MA1_HG.SG, Russia_Kostenki14, Italy_North_Villabruna_HG. Depending on the time period, the outgroup set was adjusted according to the specific test. A detailed list of outgroups per test can be found in Supplementary Tables G, H, R, S, X and Y.

### Admixture date estimation with DATES for PIE060
The software DATES (v.753)[33] was used to estimate the time of the admixture events of ancient populations under the assumption that gene flow occurred as a single event and that the generation time is 28 years[34]. DATES measures the decay of ancestry covariance to infer the admixture time and estimates the variance of this admixture using a jackknife approach. The following parameters were used for every run: binsize 0.001; maxdis 1; qbin 10; lovalfit 0.45. For PIE060, the two reference populations were chosen on the basis of the best-fitting ancestry model from qpAdm.

### Imputation
Samples were imputed using GLIMPSE (v.1.0.1) with the default parameters[87,88]. Briefly, bam files were trimmed 2 bp to remove ancient DNA damage. We then determined genotype likelihoods from trimmed bam files using bcftools[89] with the 1,000G panel (The 1,000 Genomes Project consortium[90]) as a reference. We used GLIMPSE_impute on genomic chunks of 2,000,000 bp with the buffer size of 200,000 bp to perform imputation. We then ligated the chunks using GLIMPSE_ligate and determined the most likely haplotypes using GLIMPSE_sample. Samples with more than 0.5× coverage on the 1,240,000 positions (around 550,000 SNPs) after imputation were included in IBD analysis. No MAF filtering was performed, since only 1,240,000 positions were retained after imputation.

### Runs of homozygosity
The software package HapROH (v.0.64) was used to analyse ROH on pseudohaploid 1,240,000 SNP capture data[91]. Only samples with more than 400,000 SNPs were included in the analysis to prevent potential false positives (Supplementary Table K).

### IBD sharing
IBD sharing analysis was done using ancIBD (v.0.4)[92] on individuals with more than 600,000 SNPs and genotype probabilities > 0.99 after imputation with GLIMPSE[87,88]. We used HapBLOCK to perform the IBD sharing estimation. Imputed samples were merged, then the

vcf_to_1240K_hdf command was used to convert the vcf files to the hdf5 format. The hapBLOCK_chroms command was used to perform the IBD sharing analysis for each chromosome at a time using the default parameters. Following that, only shared blocks of more than 220 SNPs per centimorgan and shared blocks of more than 5 cM were kept for data quality purposes and used for plotting (Supplementary Table J).

### Metagenomic pathogen screening
Shotgun sequencing data were screened for the presence of pathogen DNA with the screening pipeline HOPS (v.0.2)[93]. First, adaptor-clipped reads were mapped to a custom-made RefSeq database using MALT v.0.4.0 (ref. 94) in BlastN mode and with semiglobal alignment type and default pipeline settings. The used database included all available complete bacterial and viral genomes as of 2017 in addition to selected eukaryotic pathogen genomes and the human reference sequence GRCh38. The results were filtered with a predefined list of pathogens of interest and possible candidates authenticated on the basis of edit distance distribution, ancient DNA damage pattern and read distribution along the reference genome.

### Reporting summary
Further information on research design is available in the Nature Portfolio Reporting Summary linked to this article.

## Data availability
The DNA sequences reported in this paper have been deposited in the European Nucleotide Archive under the accession number PRJEB62503.

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

**Acknowledgements** We thank E. Kukral, F. Aron, L. Semerau, A. Wissgott, G. Brandt, S. Nagel and E. Essel for their support in the ancient DNA analyses and S. Clayton and K. Prüfer for the processing of the raw sequence data. We thank the teams at (former) MPI-SHH-Archaeogenetics and MPI-EVA-Archaeogenetics for continued support and discussion. This research was funded by the Max Planck Society, the European Research Council under the European Union's Horizon 2020 research and innovation programme (grant agreement nos 771234-PALEoRIDER, W.H.; 834616-ARCHCAUCASUS, S.H.; and 678901-FoodTransforms, P.W.S.) and the ERA.Net RUS Plus initiative (S&T-277-BIOARCCAUCASUS, S.H.).

**Author contributions** W.H., S.H., P.W.S. and J.K. designed the study. R.K., G.G.-R., C.S., M.A.S., G.U.N., A. Chohadzhiev, A.S., K.P., S.R., M.L., I.M., B.G., K. Boyadzhiev, Y.B., K. Beutler and I.B. provided materials and resources. S.P. performed laboratory experiments. S.P., A.B.R., G.G.-R., C.S., G.U.N. and A. Childebayeva analysed the data. W.H., I.M., S.H., P.W.S., Y.B. and K. Boyadzhiev assisted with data interpretation. S.P., S.H., P.W.S., I.M., A.B.R. and W.H. wrote the manuscript with contributions from all co-authors.

**Funding** Open access funding provided by Max Planck Society.

**Competing interests** The authors declare no competing interests.

**Additional information**
**Correspondence and requests for materials** should be addressed to Sandra Penske or Wolfgang Haak.

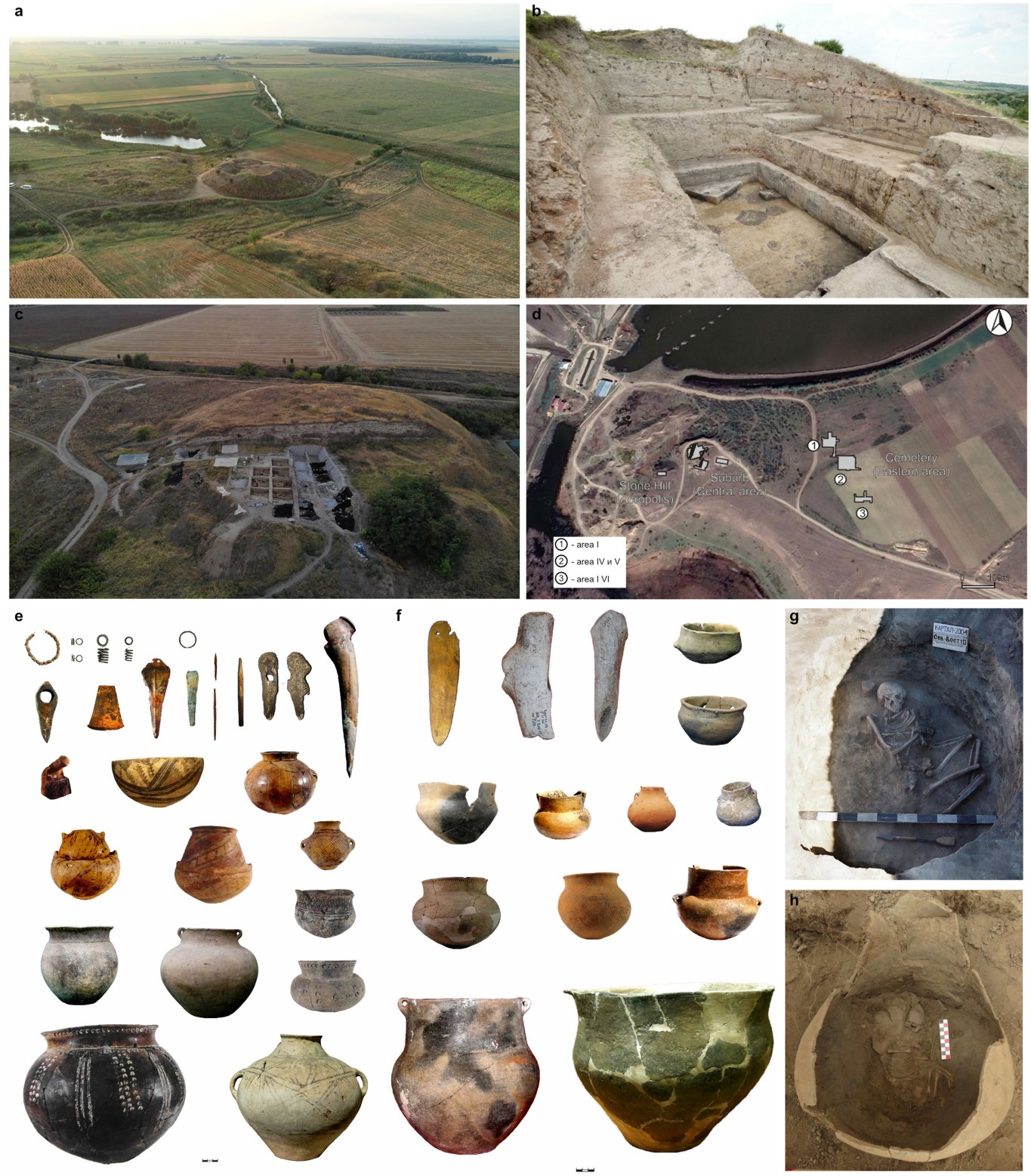

**Extended Data Fig. 1 | Tell settlements and burial sites in Southeastern Europe. a**, Aerial view of Tell Măgura Gorgana near Pietrele, Romania (© Konstantin Scheele, German Archaeological Institute, Eurasia Department). **b**, Detailed view of the 11m stratigraphy at Pietrele (© Svend Hansen, German Archaeological Institute, Eurasia Department). **c**, Aerial view of Tell Yunatsite, Bulgaria (© Kamen Boyadzhiev). **d**, Map of the site Orlovka-Kartal, Ukraine.

The base map was sourced from Google Earth https://www.google.com/earth/index.html. **e**, Characteristic finds from the Eneolithic type-site Usatove. **f**, Characteristic finds attributed to the Cernavoda I phase. **g**, Burial in flexed position from grave 10 at Kartal (© Igor Bruyako). **h**, Infant urn-burial from the Early Bronze Age layer south of tell Yunatsite (© Kamen Boyadzhiev).

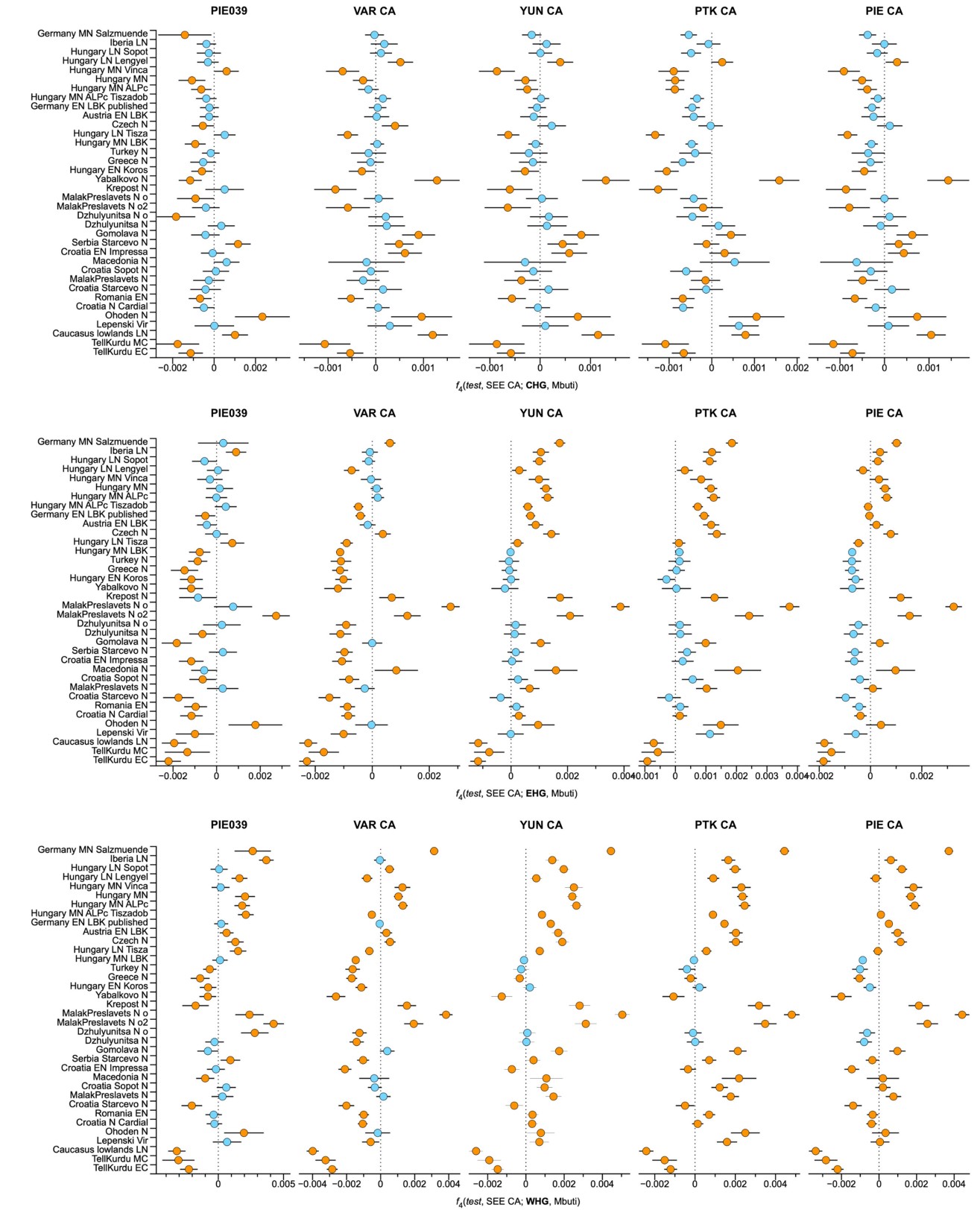

**Extended Data Fig. 2 | $F_4$-statistics for CA groups to determine Neolithic proxies.** $F_4$-statistics show different attraction of the CA to Neolithic groups conditioned on HG groups. Z-scores outside the threshold of ($|Z| \geq 1$) are highlighted in orange, $f_4$-values are shown with one standard error. *Test* populations are given on the y-axis. Standard errors (SE) were computed with the default block jackknife approach.

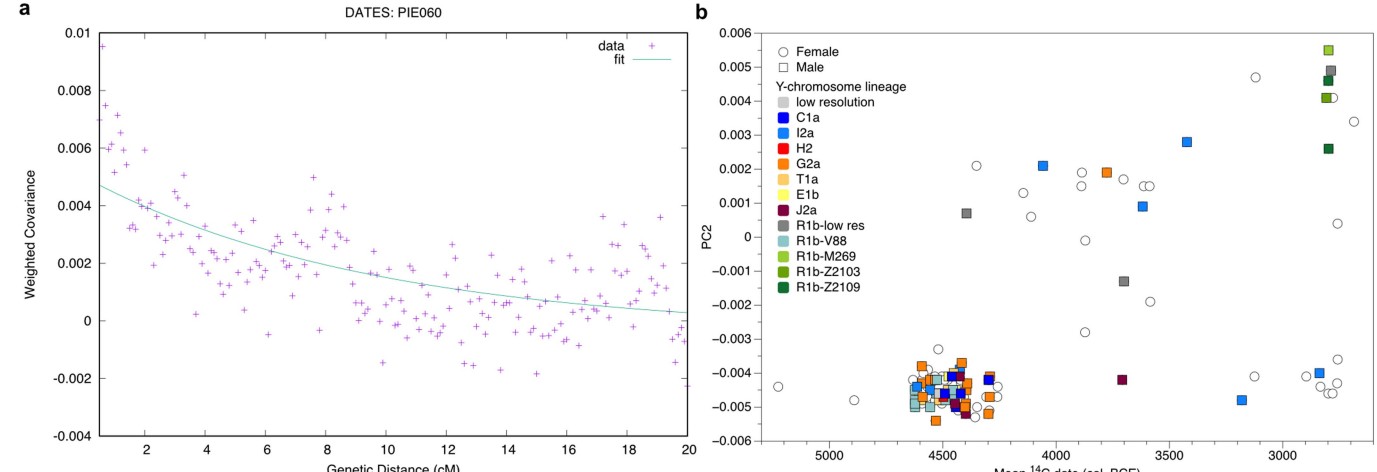

**Extended Data Fig. 3 | DATES and Y-chromosomal haplogroup diversity.**
**a**, DATES decay curve for the CA outlier individual PIE060 with SEE N and Iron Gates HGs as source populations. **b**, Changes of Y-chromosomal haplogroup diversity (colour fills) over time (mean ${}^{14}$C dates cal. BC; x-axis) with respect to changes in autosomal ancestry as reflected in PC2 (y-axis), based on the relative density of female (open circles) and male (colour filled squares) of all newly reported individuals in this study.

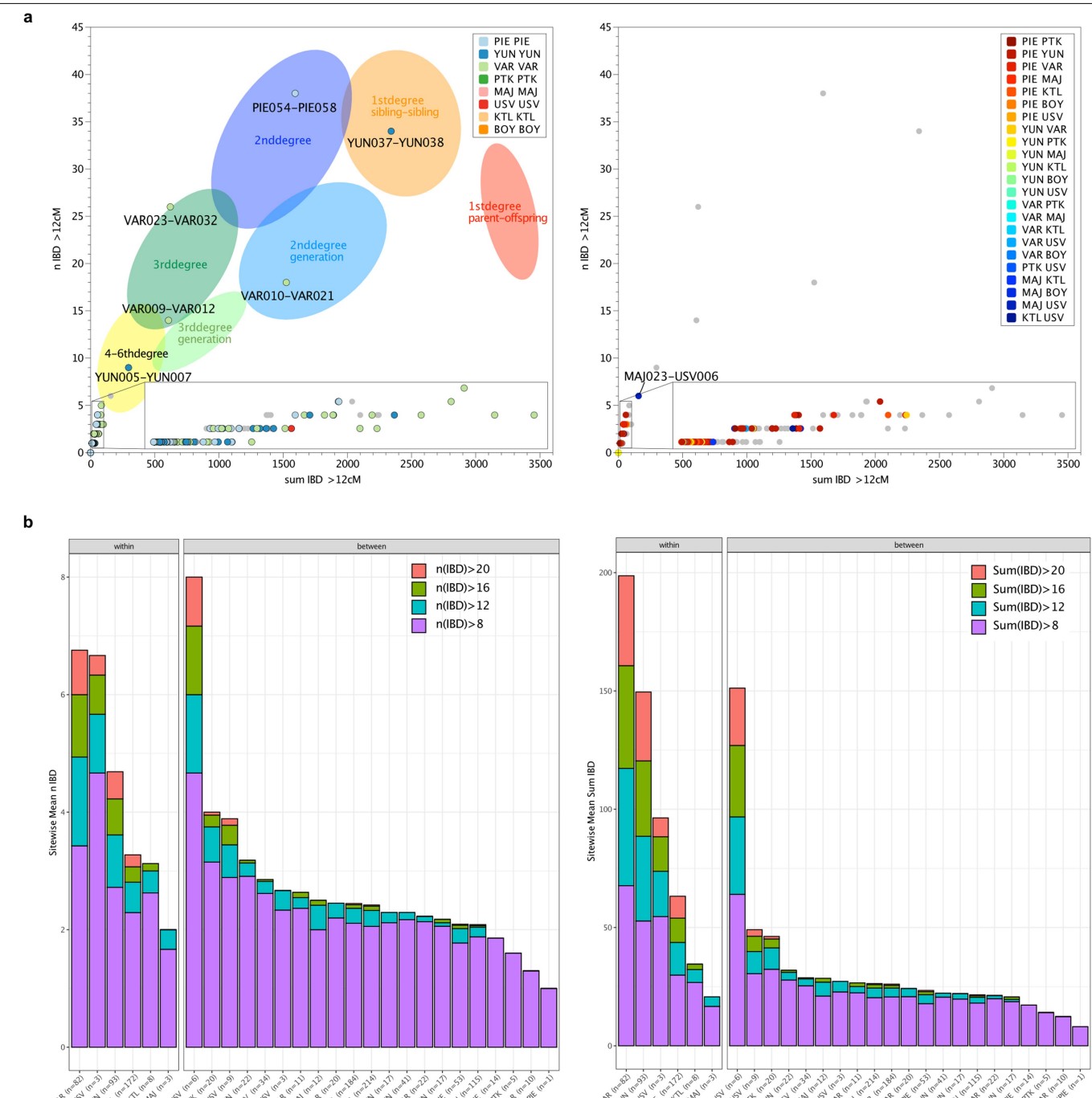

**Extended Data Fig. 4 | Identity-by-descent within and between sites.**
**a**, Results of identity-by-descent (IBD) analysis per pair of individuals. Plotting the sum versus the number of the shared chunks of IBD in window sizes of >12cM resolves degrees of biological relatedness up to the 4-6th degree.

Within (left) and between site (right) relationships are highlighted separately. **b**, Stacked bar plot showing the number (left) and the cumulative distribution of the sum (right) of IBD blocks that are shared between all individuals within and between sites.

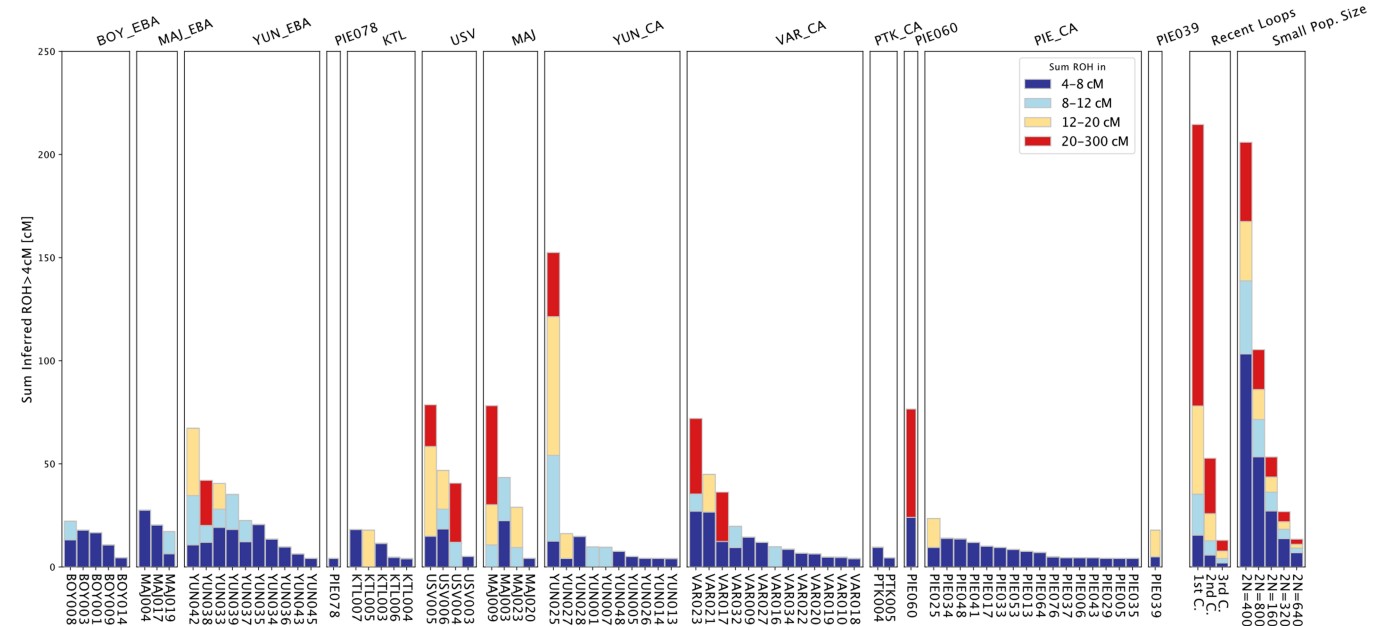

**Extended Data Fig. 5 | Cumulative distribution of the runs of homozygosity tracts of all newly reported individuals.** Runs of homozygosity were estimated with hapROH. Individuals are grouped in relative chronological order from right to left. Expected parental relationship and simulated effective populations sizes are given.

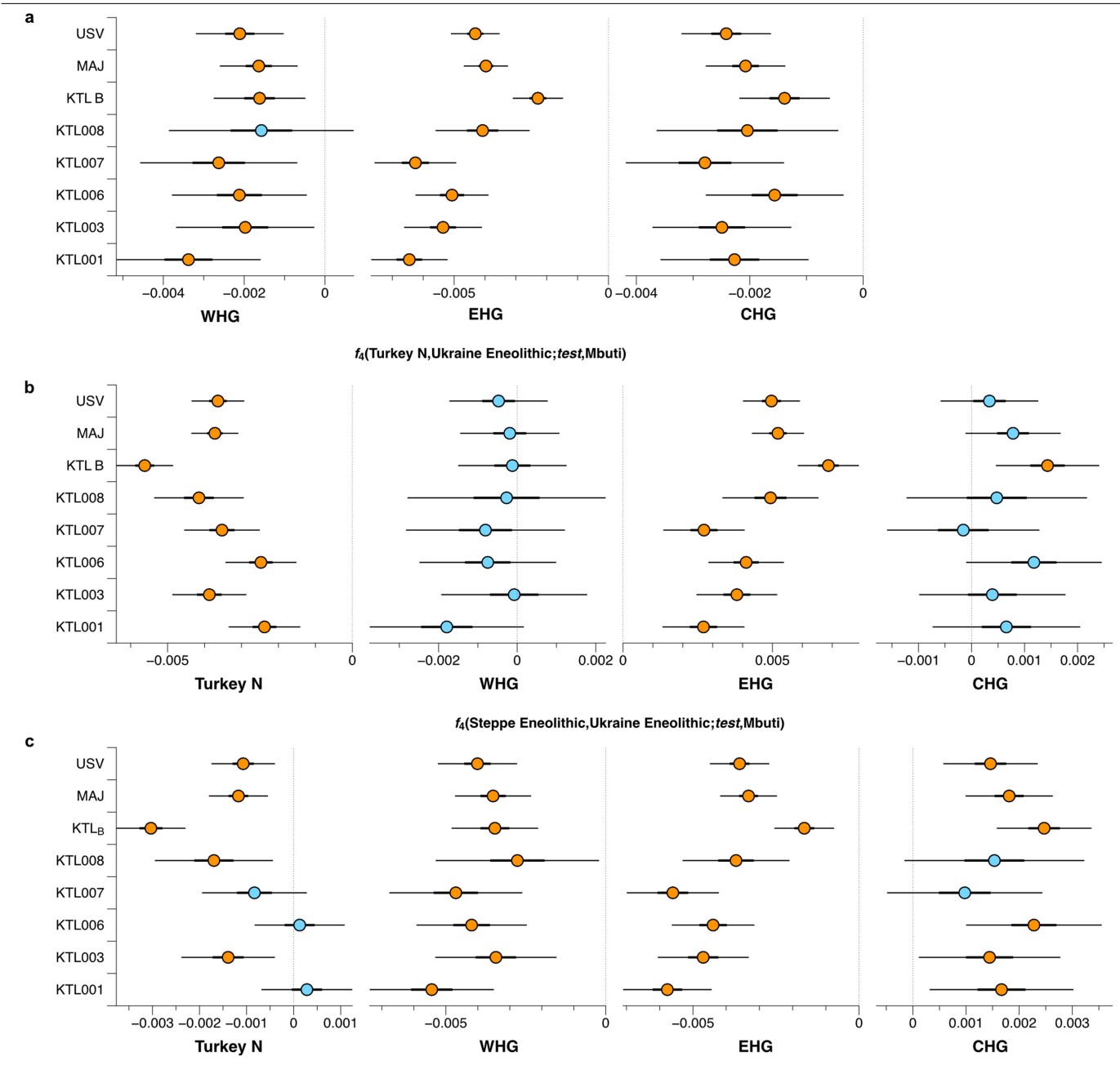

**Extended Data Fig. 6 | $F_4$ statistics for Ukraine Eneolithic groups to determine attraction to 'cornerstone' populations.** $F_4$ statistics show different attractions of Ukraine Eneolithic groups to 'cornerstone' ancestry groups conditioned on **a**, Maykop-associated groups, **b**, Steppe Eneolithic, and **c**, SEE CA. Significant Z-scores (|Z|≥3) are highlighted in orange, $f_4$ values are shown with three standard errors. *Test* populations are given on the y-axis. Standard errors (SE) were computed with the default block jackknife approach.

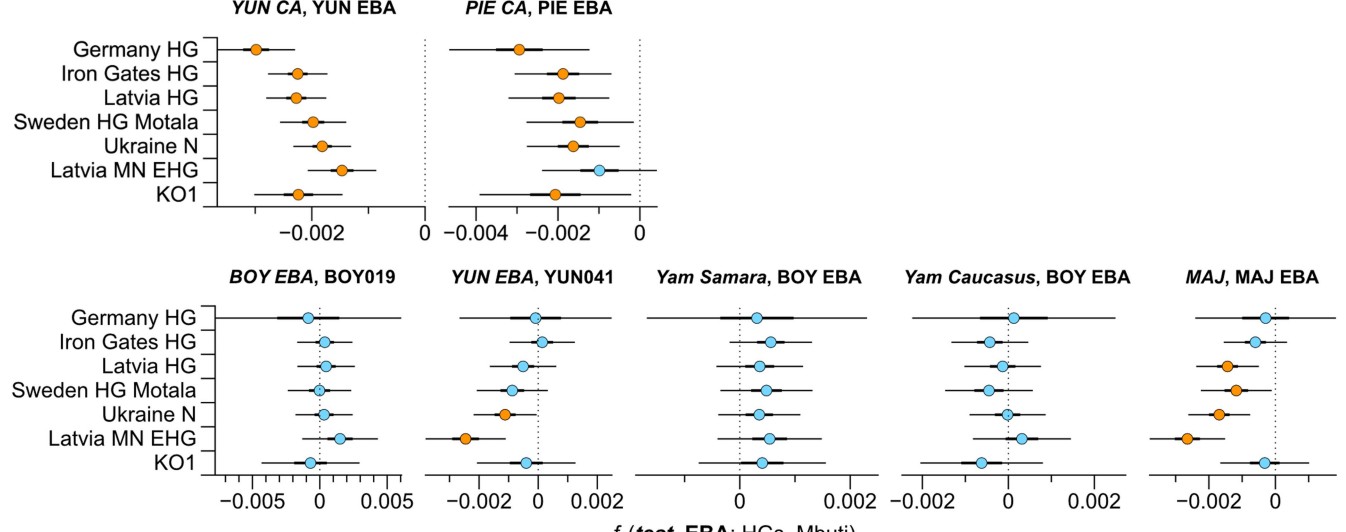

**Extended Data Fig. 7 | $F_4$ statistics for EBA groups to test for excess HG attraction.** $F_4$ statistics show different attractions of EBA groups to HG groups conditioned on their respective preceding or contemporaneous group. Significant Z-scores ($|Z| \geq 3$) are highlighted in orange, $f_4$ values are shown with three standard errors. *Test* populations are given on the y-axis. Standard errors (SE) were computed with the default block jackknife approach.

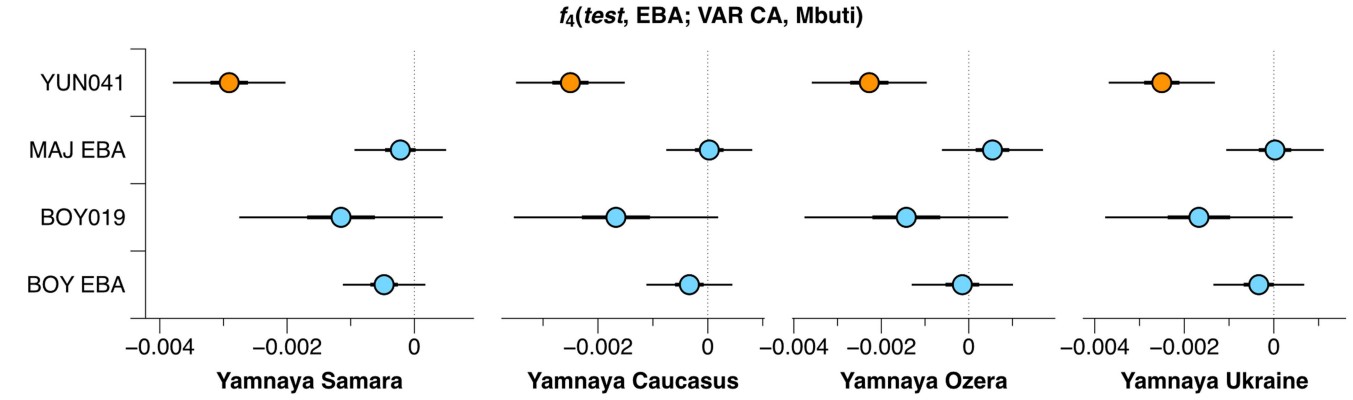

**Extended Data Fig. 8 | Testing for Anatolian farmer-related affinity in EBA individuals.** $F_4$ statistics show different attractions of the EBA groups to VAR_CA conditioned on Yamnaya-associated groups. Significant Z-scores ($|Z|{\geq}3$) are highlighted in orange, $f_4$ values are shown with three standard errors. *Test* populations are given on the y-axis. Standard errors (SE) were computed with the default block jackknife approach.

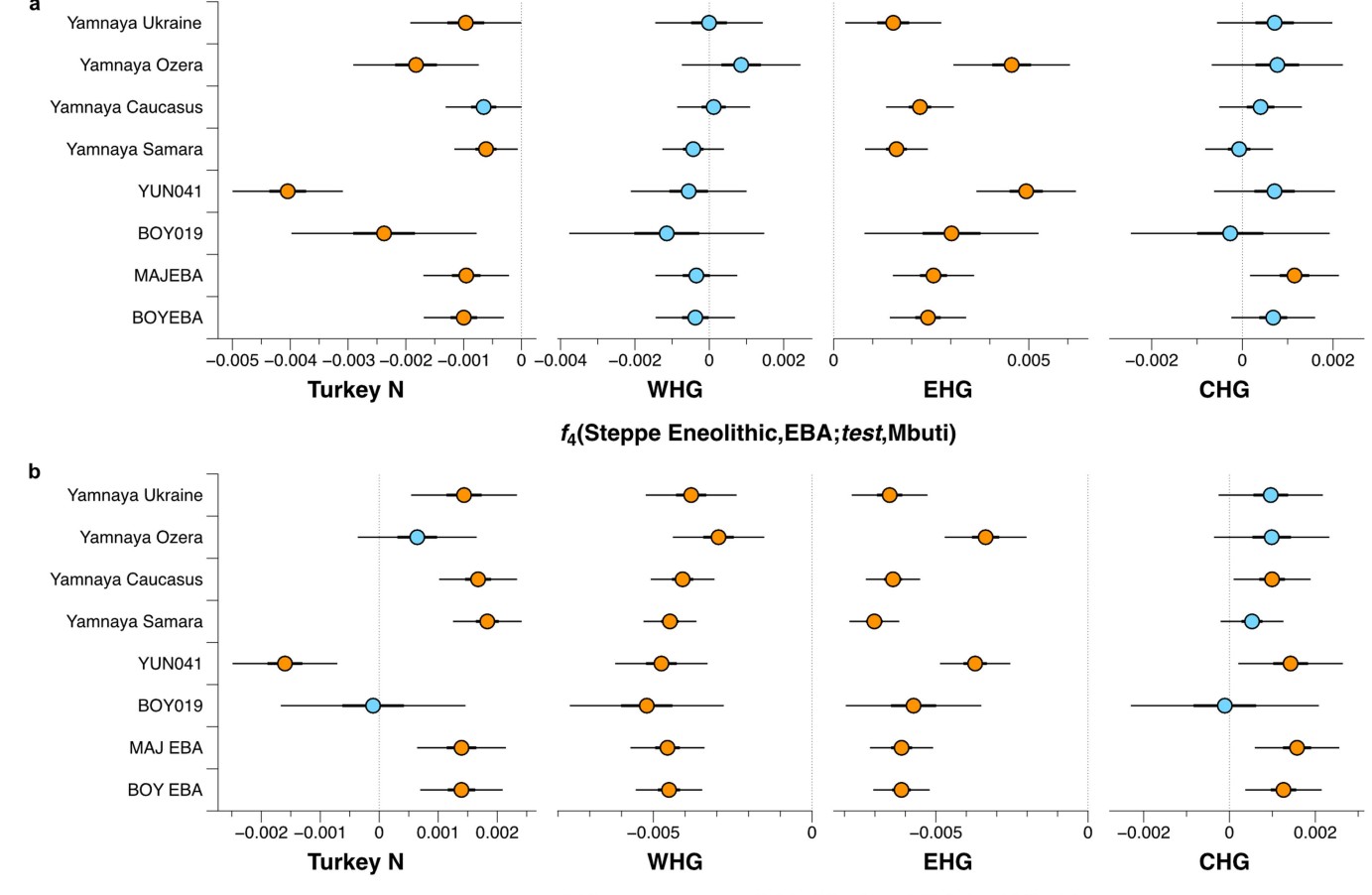

**Extended Data Fig. 9 | $F_4$ statistics to explore excess affinity of EBA 'steppe ancestry' groups conditioned on preceding Eneolithic groups from the steppe and the Caucasus.** $F_4$ statistics show different attractions of the EBA groups to *'cornerstone'* populations conditioned on pre-Yamnaya groups, Steppe Eneolithic and Caucasus Eneolithic/Maykop. Significant Z-scores ($|Z|{\geq}3$) are highlighted in orange, $f_4$ values are shown with three standard errors. EBA groups are given on the y-axis and *test* populations are given on the x-axis. Standard errors (SE) were computed with the default block jackknife approach.

| | |
|---|---|

# Reporting Summary

## Statistics

For all statistical analyses, confirm that the following items are present in the figure legend, table legend, main text, or Methods section.

| n/a | Confirmed | |
|---|---|---|
| ☐ | ☒ | The exact sample size ($n$) for each experimental group/condition, given as a discrete number and unit of measurement |
| ☐ | ☒ | A statement on whether measurements were taken from distinct samples or whether the same sample was measured repeatedly |
| ☐ | ☒ | The statistical test(s) used AND whether they are one- or two-sided<br>*Only common tests should be described solely by name; describe more complex techniques in the Methods section.* |
| ☒ | ☐ | A description of all covariates tested |
| ☐ | ☒ | A description of any assumptions or corrections, such as tests of normality and adjustment for multiple comparisons |
| ☐ | ☒ | A full description of the statistical parameters including central tendency (e.g. means) or other basic estimates (e.g. regression coefficient) AND variation (e.g. standard deviation) or associated estimates of uncertainty (e.g. confidence intervals) |
| ☐ | ☒ | For null hypothesis testing, the test statistic (e.g. $F$, $t$, $r$) with confidence intervals, effect sizes, degrees of freedom and $P$ value noted<br>*Give P values as exact values whenever suitable.* |
| ☒ | ☐ | For Bayesian analysis, information on the choice of priors and Markov chain Monte Carlo settings |
| ☒ | ☐ | For hierarchical and complex designs, identification of the appropriate level for tests and full reporting of outcomes |
| ☒ | ☐ | Estimates of effect sizes (e.g. Cohen's $d$, Pearson's $r$), indicating how they were calculated |

*Our web collection on statistics for biologists contains articles on many of the points above.*

## Software and code

Policy information about availability of computer code

| | |
|---|---|
| Data collection | No specific software was used for data collection. All software used for the processing of raw sequencing data and generation of genotype files is listed below. |
| Data analysis | The following freely available software was used for data analyses. The corresponding citations are provided in the Material & Methods section: EAGER (1.92.56), AdapterRemoval (v2.3.0), BWA (v 0.7.12), DeDup (v 0.12.1), MapDamage (v 2.0.9), samtools (v 1.3), pileupCaller (v 1.4.0.2), bamUtils (v 1.0.13), ANGSD (0.910), contamMix (v1.0 12), ADMIXTOOLS (5.1) (qp3Pop, qpDstats, qpWave, qpAdm), EIGENSOFT package (v 7.2.1), smartpca (v 16000),  Haplogrep 2 (v2.4.0), R (v 3.6.2), DATES (v 753), ancIBD (0.4), hapROH (0.64), GLIMPSE (1.0.1), OxCal (v4.4.2), READ (no versioning), Geneious (v 2019.2.3), HOPS (v 0.2), MALT (v.0.4.0) |

For manuscripts utilizing custom algorithms or software that are central to the research but not yet described in published literature, software must be made available to editors and reviewers. We strongly encourage code deposition in a community repository (e.g. GitHub). See the Nature Portfolio guidelines for submitting code & software for further information.

## Data

Policy information about availability of data

All manuscripts must include a data availability statement. This statement should provide the following information, where applicable:
- Accession codes, unique identifiers, or web links for publicly available datasets
- A description of any restrictions on data availability
- For clinical datasets or third party data, please ensure that the statement adheres to our policy

Genomic sequence data (BAM format) is available at the European Nucleotide Archive under project accession number PRJEB62503. The published genotype data compiled and annotated as Allen Ancient DNA Resource (AADR v44.3) and that was used for comparative analyses is available here: https://reich.hms.harvard.edu/allen-ancient-dna-resource-aadr-downloadable-genotypes-present-day-and-ancient-dna-data.
The human mitochondrial revised Cambridge Reference Sequence (NC 012920.1): https://www.ncbi.nlm.nih.gov/nuccore/251831106.
The human reference genome GrCh38 (hg38): https://www.ncbi.nlm.nih.gov/datasets/genome/GCF_000001405.26/
The human reference genome GrCh37 (hg19): https://www.ncbi.nlm.nih.gov/datasets/genome/GCF_000001405.13/
The 1000 human genomes reference panel: http://hgdownload.cse.ucsc.edu/gbdb/hg19/1000Genomes/phase3/

## Human research participants

Policy information about studies involving human research participants and Sex and Gender in Research.

| Reporting on sex and gender | not applicable |
| --- | --- |
| Population characteristics | not applicable |
| Recruitment | not applicable |
| Ethics oversight | not applicable |

Note that full information on the approval of the study protocol must also be provided in the manuscript.

# Field-specific reporting

Please select the one below that is the best fit for your research. If you are not sure, read the appropriate sections before making your selection.

☒ Life sciences ☐ Behavioural & social sciences ☐ Ecological, evolutionary & environmental sciences

For a reference copy of the document with all sections, see nature.com/documents/nr-reporting-summary-flat.pdf

# Life sciences study design

All studies must disclose on these points even when the disclosure is negative.

| Sample size | No statistical methods were used to determine ancient DNA sample size a priori. Sample sizes for ancient groups/population depends entirely on availability and preservation of human skeletal remains associated to archaeologically described cultures or techno-complexes. |
| --- | --- |
| Data exclusions | We processed and screened samples from 216 prehistoric individuals, of which 135 were deemed suitable for downstream analyses, following pre-defined data quality and authentication criteria, described in the Methods section. Data from specimens that showed insufficient levels of ancient DNA content or high levels of DNA contamination were excluded from further analyses. |
| Replication | We study unique entities of past populations and did not use different treatments or variations of data analyses, so replication is not applicable. We recognize that individuals from the same region and time period of the past show similarities, and that their particular ancestry composition does not exist in the same form anymore today. Genome-wide data with hundreds of thousands of SNPs allows for multiple realisations of the sample history. |
| Randomization | Individuals are grouped by time period (archaeological culture, radiocarbon date), geographic region and genetic similarity. Randomisation is thus not relevant to this study. |
| Blinding | As the archaeological and anthropological context of our samples (date and location, etc.) is critical to the interpretation of the data, blinding is not applicable to our study. |

# Reporting for specific materials, systems and methods

We require information from authors about some types of materials, experimental systems and methods used in many studies. Here, indicate whether each material, system or method listed is relevant to your study. If you are not sure if a list item applies to your research, read the appropriate section before selecting a response.

## Materials & experimental systems

| n/a | Involved in the study |
|-----|----------------------|
| ☒ | ☐ Antibodies |
| ☒ | ☐ Eukaryotic cell lines |
| ☐ | ☒ Palaeontology and archaeology |
| ☒ | ☐ Animals and other organisms |
| ☒ | ☐ Clinical data |
| ☒ | ☐ Dual use research of concern |

## Methods

| n/a | Involved in the study |
|-----|----------------------|
| ☒ | ☐ ChIP-seq |
| ☒ | ☐ Flow cytometry |
| ☒ | ☐ MRI-based neuroimaging |

# Palaeontology and Archaeology

**Specimen provenance**

Specimen come from excavations supported by the Regional Museum of History - Veliko Tarnovo, the German Archaeological Institute - Berlin (Germany), the Institute for Archaeology "Vasile Pârvan" of the Romanian Academy of Sciences - Bucharest (Romania), the Institute for Geography of the Goethe University - Frankfurt/Main (Germany), the Varna Regional Museum of History - Varna (Bulgaria), the National Institute of Archaeology with the Museum at the Bulgarian Academy of Sciences (IAM-BAS) - Sofia (Bulgaria),  the Regional Museum of History - Pazardzhik (Bulgaria), the  Yambol Regional Historical Museum - Yambol (Bulgaria), and the Odesa Archaeological Museum - Odesa (Ukraine), with primary contact persons provided in Supplementary Table A.
All specimens were collected and analyzed with permissions from the respective local organisations for the handling of the archaeological material, and represented by local curators and collaboration partners who are listed among the co-authors of this study.

**Specimen deposition**

Specimens are stored at the Regional Museum of History - Veliko Tarnovo, the German Archaeological Institute - Berlin (Germany), the Institute for Archaeology "Vasile Pârvan" of the Romanian Academy of Sciences - Bucharest (Romania), the Institute for Geography of the Goethe University - Frankfurt/Main (Germany), the Varna Regional Museum of History - Varna (Bulgaria), the National  Institute of Archaeology with the Museum at the Bulgarian Academy of Sciences (IAM-BAS) - Sofia (Bulgaria),  the Regional Museum of History - Pazardzhik (Bulgaria), the  Yambol Regional Historical Museum - Yambol (Bulgaria), and the Odesa Archaeological Museum - Odesa (Ukraine).
Specimens will be returned to the respective heritage organization and museums after completion of the joint collaborations. DNA extract and libraries will remain stored at the ancient DNA laboratories of the Max Planck Institute for Evolutionary Anthropology, Jena & Leipizg, Germany.

**Dating methods**

New AMS 14C dates were obtained from ultra-filtrated collagen. Collagen extraction and 14C measurements were carried out at the Curt-Engelhorn-Zentrum Archäometrie gGmbH, Mannheim, Germany.

☒ Tick this box to confirm that the raw and calibrated dates are available in the paper or in Supplementary Information.

**Ethics oversight**

No ethics oversight was required.
The archaeological and anthropological researchers included as coauthors of this study were granted permissions to sample skeletal elements for the purpose of ancient DNA analyses. All steps in the analyses followed standard ethical guidelines with regards to respectful handling, documentation, storage, transport, sampling and processing of human skeletal elements, with support from the Regional Museum of History - Veliko Tarnovo, the German Archaeological Institute - Berlin (Germany), the Institute for Archaeology "Vasile Pârvan" of the Romanian Academy of Sciences - Bucharest (Romania), the Institute for Geography of the Goethe University - Frankfurt/Main (Germany), the Varna Regional Museum of History - Varna (Bulgaria), the National Institute of Archaeology with the Museum at the Bulgarian Academy of Sciences (IAM-BAS) - Sofia (Bulgaria), the Regional Museum of History - Pazardzhik (Bulgaria), the  Yambol Regional Historical Museum - Yambol (Bulgaria), the Odesa Archaeological Museum - Odesa (Ukraine) who approved and provided guidance on the study protocol.

Note that full information on the approval of the study protocol must also be provided in the manuscript.

