## [Peer Review File · Nature]

Manuscript Title: Early contact between late farming and pastoralist societies in southeastern Europe

Reviewer Comments & Author Rebuttals

Reviewer Reports on the Initial Version:

Referee expertise:

Referee #1: archaeology

Referee #2: evolutionary genetics, aDNA

Referee #3: aDNA

Referees' comments:

Referee #1 (Remarks to the Author):

A

Concise summary of the archaeogenetic data and results. The results of the genome-wide analyses are linked to the transfer of innovations and ideas in the 4th millennium BCE, an aspect that is briefly addressed in the 'Discussion' section but not elaborated on.

B

This archaeogenetic study is of great relevance as it provides new insights by presenting genome-wide analyses for the eastern Balkan region and the northwestern Black Sea area during the 5th, 4th and early 3rd millennium BCE. The northwestern Black Sea region has so far been a region for which only a very small amount of archaeogenetic data is available. With 134 sampled individuals from both areas, a new large data base is provided which traces genetic continuities and integrates into archaeological knowledge.

C&D

Lines 162 to 371 present the genetic data analysed using bioinformatics. As an archaeologist, I cannot competently judge the validity of the statistical methods and the quality of the data. Most of the statistical methods are generally known to me from other population genetics articles. Therefore, I can only comment briefly on the presentation of the data, again from an archaeologists' perspective. The data and their results are presented in the text in a well comprehensible way. Frequent reference is made to the relevant figures and tables within the text and in the extended and supplementary materials.

E The study with its three main statements is divided into three chronological sections. For the individual chronological sections, there is a bias in the geographical regions from which samples are

available. This is certainly related to the availability of suitable sample material, but is not addressed explicitly in the text.

The largest data set is available for the eastern Balkan region in the 5th millennium BCE. The genome-wide analyses prove genetic continuity, for single individuals from tell settlements relatedness could be proven. An important additional piece of information from genetics is that no evidence of pathogens has been found. So no evidence for an epi- or pandemic could be provided from genetics which is of interest for the ongoing discussion about the causes of the end of the tell in the Copper Age.

The second data set is available for the 4th millennium BCE. For this, only individuals from the northwestern Black Sea area were analysed. The fact that there was contact between the cultural groups distinguished by archaeologists in this period, which is also reflected genetically, is not really surprising in view of the regional proximity of the sampled sites. More noteworthy here is the diversity of genetic components that point to Exchange with communities in different directions. There might be a connection here with the rather heterogeneous burial customs, as described by Y. Rassamakin for the steppe Region, which is not addressed in the paper.

For the early 3rd millennium BCE, data are mainly presented from the Balkan area, only from Mayaki in the Odessa region were two individuals of the Early Bronze Age genome-wide analysed. A 'mosaic of ancestries' (line 425) has emerged in the Balkans in this period. This too seems to go hand in hand with the archaeological evidence, looking at the new funerary customs in burial mounds associated with the Yamnaya Culture. On the other hand, some tell settlements were being resettled, and their pottery is clearly different from the vessels that are found in Yamnaya graves. In the East European steppe, on the other hand, the 'mosaic of ancestries' described in this study for the 4th millennium BCE for the north-western Black Sea area seems to disappear. The aspect of a relatively uniform genetic 'steppe' or 'Yamnaya' ancestry is not addressed in this paper. Can this change in genetic composition from the 4th to the 3rd millennium BCE be assessed more concretely for the steppe area on the basis of the data presented here?

Overall, the conclusions are fortunately written in a very differentiated way. Too far-reaching interpretations are avoided. I fully agree with the last paragraph, because only by expanding the genetic data sets and their interdisciplinary interpretation complex cultural dynamics in different parts of Eurasia can be understood.

F

Article - Figure 1b green squares: probably 'Yamnaya' or 'Steppe' ancestry, but not mentioned in the list of published data

Extended data - Fig. 1 presents finds from a) Pietrele and b) Usatovo and c) a grave from tell Yunatsite. The illustrations of artefacts have no relation to the text. Why were these finds depicted, what should they show? More appropriate might be illustrations showing the tell settlements in a photograph or topographical plans or a grave of the Usatovo culture. It would probably even make more sense to add illustrations to the sites from which samples were taken and which are described in 'Supplementary information'. In this case Figure 1 in 'extended data' could be dispensed.

G

The literature cited is well selected and corresponds to the (archaeological) state of research. In some places the numbering seems to be mixed up, please check: e.g., ref. 55 probably refers to 54; ref. 56-60 probably to 55-59, etc.

H

A clearly structured and well written paper.

Referee #2 (Remarks to the Author):

Penske et al. present genome-wide data and C14 dates from 100+ individuals spanning 3000 years from the Early Chalcolithic to the EBA from East Balkans to the North-West Pontic region. The manuscript effectively describes changes in material culture in the region through this period. The authors then perform in depth analysis of their dataset to demonstrate admixture between Anatolia_N-related groups and groups related to pre-Neolithic likely local HG groups in the region, and later, variable levels of Yamnaya ancestry in the broad region. They also report interesting cases of single individuals with distinct ancestry. The work further involves description of IBD sharing within and between Chalcolithic villages.

The population genomic analyses are comprehensive. However, I felt that the manuscript's main claim, i.e. that it connects major changes in technology and other aspects of material culture during this period, with human movement patterns in the region, needed further support beyond broad parallels. For instance, this could involve testing hypotheses related to sources and timing admixture that could explain specific developments.

#####

I have some questions and suggestions below for the authors:

L228: "and burial rites independent of genetic relatedness (Extended Data Fig. S4c)."

I cannot see how Extended Data Fig. S4c supports this statement. If the authors tested whether rites were unrelated to genetic groups, it would be helpful to provide this.

L229: "Taken together, these findings support the cross-regional significance of southeastern European tell sites, and reflect the cultural interconnectedness documented in the archaeological record.

This statement would benefit from some comparative support, e.g. how do the patterns observed differ from those in other regions and/or periods, and how do the patterns suggest "cultural interconnectedness"?

In the Ukrainian Eneolithic, Kartal is described as showing high heterogeneity in ancestry, which is very interesting. But the reader is left wondering what this exactly implies. For example, are there material culture differences between Kartal and Majaky that could be related to differences in ancestry heterogeneity?

The described heterogeneity in ancestry among EBA sites could also be studied in the same manner.

L269: "we also tested alternative scenarios of population interactions and cultural influences that

link the northern Black Sea to the North Caucasus"

It would be very interesting to test "cultural influences" indeed, but was this really done in this work?

L374: "The genetic continuity observed among the individuals at the four CA sites [...] extended periods of stability for several centuries as attested to by the archaeological record."

This statement suggests lack of inter-regional migration is linked with "stability", which is not necessarily true.

The authors could also explain concretely what the archaeological record is exactly showing (for those not knowledgeable in the field). How does it concretely differ from the Ukrainian Eneolithic and or the EBA?

I thought that the results in section "Genetic influence from the east during the Eneolithic in Ukraine" could be explained in a more succinct manner, and the main conclusion, that the Ukrainian Eneolithic could be explained as admixture between SEE_CA-related groups and Steppe Maykop-related groups, could be more clearly explained (some more specific notes below). Some of the f4 tests are difficult to interpret, at least for me, and I am not sure how they contribute to the picture. Also, whether the intriguing genetic heterogeneity within the Kartal population can be attributed to temporal differences is an obvious question, but is not addressed.

#####

Minor points/suggestions:

Fig 1. Perhaps indicate that titles of panels c refer also to those of panels b?

Perhaps better show the other way round, with older on top?

Dark green small squares - I couldn't find them in the legend. Is it Novosvobodnaya?

The authors could also consider multiplying PC1 by -1 to have the axis follow the east-west direction, just to help the reader accustomed to such PCA plots - just a suggestion.

L154: numbers appear inconsistent with those in L145. Perhaps the 1324 individuals are both modern-day and ancient?

Some numbers related to sample sizes seemed inconsistent:

L140: "we report genome-wide data for 134 (out of 216 attempted individuals)"

L645: "Together, a total of 130 individuals yielded sufficient genomic data for downstream analysis."

L143: "114 new radiocarbon dates"

L613: "we obtained direct 14C dates for 122 individuals"

"Trypillia culture": The approximate period could be mentioned in the text?

Extended Data Fig. S2 and some of the other supp figures: The f4 configuration is such that positive values (to the *right*) indicate affinity to "test", while the test pop names are on the *left* axis. This was confusing for me.

Fig 2: The sample used in the figure (test2 populations) should be described in the caption. Honestly, I was also unsure how much information the figure conveyed, so the authors may also consider moving this to the supplement - just a suggestion.

L176: "Moreover, outgroup-f3 statistics suggest local genetic continuity throughout the CA in this region"

The f3 pattern looks fine, but would it not be preferable to test this by showing f4-based affinity between SEE CA groups relative to neighbouring groups and/or time periods? Or compare f3 values statistically?

L184: The mentioned shift on the PC space is not super clear. Perhaps include a more focused version of the PCA?

L194: Perhaps add another figure to point out PIE060 in PC space? Or add a marker on Fig 1 to indicate this individual?

L200: A single outlier individual, PIE060, with 30% WHG ancestry and an admixture estimate of 300 years is somewhat surprising. A 300-year-old admixture event should have diffused within the group. Hence, does this imply the individual was a migrant from a region where WHG admixture was more common?

L202: "Haplotype blocks as long as" - perhaps add that these are total numbers, not lengths of single blocks.

L208: "typical Mesolithic [...] Y lineage" - could the authors provide a reference or explanation as to how this "typical" was defined?

L224: It was not clear to me why burials in a burial ground would be more closely related than burials within a settlement. Some further explanation would be welcome.

L275 and L292: the same f4 model is reported, but the text reads as if they were different.

Both use f4(Steppe Eneolithic/Maykop, Ukraine_Eneolithic; test, Mbuti).

Is it a mistake or am I missing something?

In general, this section was a bit confusing to me and might be revised to better explain the analyses and the motivation behind them.

L293 and L293: "farmer-related groups" and "HG" - I feel "farmer" and "HG" are somewhat too ambiguous terms. Would be helpful to define these more strictly, e.g. Anatolia_N-related or SEE_N-related.

L298: "Ukraine_Eneolithic individuals have a higher affinity towards HG groups when compared to Novosvobodnaya Maykop, but are cladal with other 'Steppe Eneolithic/Maykop' groups". I feel that the use of "cladal" here is misleading; perhaps it could be rephrased as "cladal w.r.t. HG affinity." Further, there is quite some heterogeneity in the f4 scores, so it might be better to be more cautious

in interpretation.

L301: "Only KTL_A produced significant negative statistics". This is in comparison to Khvalynsk, is that correct? If so, I cannot see how this is supported in Extended Data Fig. S6.

L307: "Significant negative results ($|Z| \geq 3$) indicate shared drift between KTL_B/KTL005 and the tested HGs which is the case for Tell Kurdu and for two HG groups when compared to YUN_CA." - I'm sorry but I can't see this result in Extended Data Figs. S5 and S6. Would be great if the authors clarify this further.

L304-309: "KTL005 and KTL_B fell close to the SEE_CA cluster in PCA and therefore we tested for a possible attraction to HGs with the test $f_4(\text{SEE_CA, Ukraine_Eneolithic; HGs, Mbuti})$. Significant negative results ($|Z| \geq 3$) indicate shared drift between KTL_B/KTL005 and the tested HGs which is the case for Tell Kurdu and for two HG groups when compared to YUN_CA. All other tested SEE_CA share either more drift with HGs or are cladal with KTL_B/KTL005 ($|Z| \leq 3$) (Extended Data Fig. S7, Supplementary Data P)."

This part was highly confusing for me. The motivation appears to be finding the HG contributor to KTL_B relative to SEE_CA. Why KTL_A and other Ukraine_Eneolithic groups were not used should be better explained.

Obviously $f_4(\text{SEE_CA, KTL B; HGs, Mbuti})$ is generally not significant, which I felt should be the main result here.

Also, Extended Data Fig. S7 includes the test $f_4(\text{Tell Kurdu, KTL B; HGs, Mbuti})$, but obviously Tell Kurdu is not SEE_CA.

Further, $f_4(\text{Tell Kurdu, KTL B; HGs, Mbuti})$ being negative - what this implies was not clear to me. Finally, to aid the reader, Extended Data Fig. S7 should be referenced at the end of the first sentence.

And again, I'm not comfortable with the (for me) loose use of the term "cladal".

L306: typo in "Ukraine_Eneolithic"

L311: "explore a variety of models" - please add what is aimed to be explained by the models

L316: "ancestry for the Ukraine_Eneolithic." - perhaps say "for other Ukraine_Eneolithic populations", given that KTL_B and KTL005 are also part of this group, right?

L318: point missing.

L351: "we observed significantly negative f_4 -statistics for Ikiztepe_LC, indicating that YUN_CA possibly carries more CHG-related ancestry than YUN_EBA." - This might simply reflect the fact that YUN_EBA can be modelled as YUN_CA + Baltic ancestry so that the CHG (or perhaps Anatolia_N) ancestry in YUN_CA has been diluted over time. In any case, I find such an interpretation confusing, honestly.

L376: "Long distance IBD relationships between sites are consistent with the transregional connectivity visible in the material culture."

This is very nice, but to convince the reader, I believe the authors should at least attempt to discuss the genetic and material culture observations more concretely and in a comparative framework (even if qualitatively). For instance, they could provide some examples where long-distance IBD relationships are rarely observed and transregional connectivity is also not visible.

L391: "A new major finding from our study indicates early contact and admixture between CA farming groups from SEE and Eneolithic groups from the steppe zone in today's southern Ukraine during the 4th millennium BCE at a time when settlement densities shifted further north." Unfortunately, I don't see how this statement of "early contact" is directly supported by the results. If the basis is the observation that SEE_CA could be used as a source for modelling Ukrainian Eneolithic, then: 1) this should be explicitly stated, and 2) alternative sources should also be tested, I believe. It is possible I am missing the point here, but it would surely help the authors to clarify this point.

L397: reference to literature missing.

L402: "Y-haplogroups indicative of a farming legacy," - again I feel the use of "farming" is too lax; please specify which archaeological groups are referred to explicitly.

Fig 3:

Green, pink and yellow tones are extremely difficult to distinguish. Please add another visual marker (e.g. stripes).

Please briefly mention in the legend what the population names in the panel titles refer to (e.g. "use of Russia Khvalynsk or Russia Steppe Maykop as sources").

L683: Providing READ parameters and details would be helpful - were the PO values normalized using the median of each site, or region, or the full dataset?

L717: I presume there was no multiple testing correction applied - would be good to indicate this.

L735: "We attempted to estimate the admixture time for each target population but it resulted in large estimates with standard error being almost as high as the estimate itself."

It would be good to know how many trials were conducted, and how many were "infeasible". If most trials did not work, this actually reduces the reliability of the admixture date estimate reported in L198. (Our experience with DATES has also been similar, with a very high frequency of unreasonable estimates, which leads to the question of whether the "reasonable" looking ones should be taken seriously.)

L900: "Significant" - better remove or replace by "nominally significant"

Extended Data Fig. S1. This is very nice, but a more comprehensive description of pottery from all the sites involved would have been even more informative.

Extended Data Fig. S3.

RFmix of what individual?

What is the sample in panel b?

Also not clear to me how panel b reflects a M:F "ratio".

Extended Data Fig. S7: Why is Tell Kurdu but not e.g. Ikiztepe_LC or Kumtepe used as a possible proxy for southern gene flow?

Extended Data Fig. S8: Pink tones are too close. Perhaps use stripes or other additional markers? "Maikop" in the figure legend.

Supplement:

- The ecological and archaeological background descriptions were generally clear and well-written. Though some parts were confusing, e.g. when the text on the SEE jumps to Northern Pontic evidence - the connections could be better explained (p 4).

It would also be helpful to support the information with maps showing localities and a timeline.

- "At the moment of their death, these individuals must have been in miserable condition, most probably due to hard physical work." - Further description could be helpful.

Also, the authors could consider providing images to support the skeletal evidence.

- Do differences with respect to social status or conditions of death (trauma) or burial rituals at all correlate with genetic ancestry? I presume the authors have already checked this - it would also be helpful if they report results even if negative.

- Permits for use of the material are increasingly expected to be published as supplementary material. It would be beneficial if the authors could add these to the supplement.

- "4. Rationale for selecting source populations". This part was not easy to follow for me - a revision would help the reader.

Also, I did not understand why cladality was only tested using "HGs". Why not also use e.g. Anatolia_N as test population in f4 analyses?

Table K: I couldn't understand why YUN027-YUN026 and YUN037-YUN038, which share high IBD (Table L), were not represented among READ results?

qpAdm analyses: Would be helpful to explain how "feasible" was defined.

How was the microbial screening performed? Please add to Methods.

The authors could consider comparing relative EHG and WHG affinities in SEE_CA, e.g. by plotting them against each other.

Referee #3 (Remarks to the Author):

This manuscript describes a genome-wide analysis of 134 new ancient individuals from southeastern Europe and the northwestern part of Black Sea region. It focuses on three periods involving key transformation in the archaeological record: the Copper Age, the Eneolithic and the Early Bronze Age. I congratulate the authors on the excellent introduction, it certainly helps the reader to understand the archaeological context of the samples under analysis and the relevant questions. The

laboratory procedures and bioinformatic processing were performed at MPI-EVA, a top aDNA lab fulfilling all the standards in the field. I have some important comments on the admixture modelling (see below), which I believe could be addressed without much trouble. I do question the novelty of the findings, since the Copper Age and Early Bronze Age data consolidate with more samples the results reported in previous publications (e.g. Mathieson et al. 2018, Nature). In the case of Ukraine Eneolithic, Cernavodă I and Usatovo data are reported for the first time in this manuscript, but the findings are quite similar to the ones described in Immel et al. 2020, Scientific Reports for the neighboring Cucuteni-Trypillia, even the titles of the two manuscripts are quite similar.

Major comments:

Throughout the paper, the authors are not consistent in their choice of significance thresholds, in some cases using a p-value threshold of 0.05 ($Z=1.96$) like in qpAdm, in some f4-stats $Z=3$ and in other f4-stats $Z=1$. For instance, in Extended Data Fig. S2, many f4-values are highlighted in orange as significantly different from 0 based on $|Z| \geq 1$, which translates into a p-value threshold of 0.16 (one tail). This seems like a too stringent cut off, as a good portion of these orange f4 deviate from 0 just due to random noise.

177: The outgroups-f3-statistics mentioned here compare CA groups against contemporaneous SEE groups. When trying to test continuity over time, one should compare against the earlier period (in this case the Neolithic), as the authors do in line 181 using the $f_4(\text{SEE_CA}, \text{SEE_N}; \text{CHG}, \text{Mbuti})$ test. More importantly, outgroup-f3-statistics are not good tests for continuity, even when you are comparing against the earlier period. Let's say you have 30% discontinuity, you will likely get high outgroup f3 values against the earlier populations that shares 70% of the ancestry with your test. I agree with the authors that CA groups are largely continuous to the Neolithic groups, but this statistic is not the right one to test for continuity.

qpAdm analysis: Many models (often quite different from each other) seem to work for a given Ukraine Eneolithic group. This is likely because the qpAdm set up has little power to reject non-fitting models:

-KTL B and KTL005 are clearly shifted from the SEE CA cluster in PCA. Thus, unless this shift is the consequence of genetic drift rather than admixture, it should be possible to reject one-way models with SEE CA groups. Instead, several SEE CA groups result in good-fitting one-way models for the ancestry in KTL B and KTL005.

-MAJ can be modelled as a 2-way model with 75% KTL_B and 25% either Khvalynsk or Steppe Maykop, but Khvalynsk and Steppe Maykop are substantially different groups and with the other source population being fixed, they should not both result in good-fitting models. Same goes for the good-fitting distal 3-way models with Anatolia_N+CHG+ EHG or WSHG (the outgroup set is clearly not telling apart EHG-related and WSHG-related ancestry).

I recommend the authors to use a different outgroup set with more resolution, perhaps by using populations with better quality data and/or more individuals, chronologically closer to the Refs, and/or differentially related to the Refs.

186-187: I don't understand how this works. Let's take YUN_CA as an example. You find the Neolithic groups that are cladal with YUN_CA when compared to the four different HG groups, and then you merge all these Neolithic groups and consider this merged group as the best Neolithic proxy for YUN_CA in proximal qpADM modelling. Is that correct? If true, then:

-For YUN_CA, the proximal model features Dhulyunitsa_N but in Sup. Data D, Greece_N and Anatolia_N are also cladal with YUN_CA when compared to the four HG groups. Why aren't Dhulyunitsa_N, Greece_N and Anatolia_N merged and used for the proximal qpADM modelling?

-This is a quite convoluted procedure that results in an array of very similar merged groups, like CA merge 1,2,3,etc, containing slightly different combination of populations and each combination used for modelling a different test group. This leads to unnecessary complex figures like Fig 3. Deciding whether to merge or not two Neolithic groups (and the same goes for the CA merges) should be based on cladality tests including those two Neolithic groups against the HG groups (although you could also test against many other groups, as two Neolithic groups can be cladal against HG groups but differently related to other groups). If two Neolithic or CA groups are cladal to the limits of your resolution and you decide to merge groups for analysis, then they should be merged and used as a possible source irrespective of which of your new test groups you are trying to model. For instance, PIE_CA and YUN_CA are now merged in CA_merge_1,3,6,7,8 but not in CA_merge_2 and 5.

Minor comments

132: The Cernavodă culture appears here for the first time. It would be useful for the reader to comment on this cultural horizon earlier in the manuscript where the archaeological contexts of the different periods are described.

Fig.1a: The published Varna individuals appear on the left of the figure with other published samples, but they display a black outline even though this is used for newly reported individuals.

Fig.1: The legend for the Yamnaya is missing, and same for the Neolithic individual from Pietrele.

178: I am not sure this is entirely true, as previous publications showed that Neolithic groups in present-day Greece did have small amounts of CHG-related ancestry.

181: If you are trying to test for early contributions of steppe ancestry during the SEE_CA, why not directly using the actual steppe Eneolithic groups instead of CHG, which is a much more distal group?

192: This seems to contradict the PCA observation that SEE CA groups are shifted towards the WHG/EHG cline with respect to Neolithic groups (Lines 184-185).

Extended Data Fig. S3: The strong correlation of HG segment locations in the maternal and paternal chromosomes (in several cases with the exact same start-end coordinates) is somehow suspicious, and likely indicates problems either in the imputation/phasing step or in the local ancestry estimation.

Extended Data Fig. S3: In the legend for panel a, please give more information, i.e. what sample is featured. In the legend for panel b, I would remove "Ratio of female (open circles) and male (colour filled squares) individuals" and move it to the end of the legend.

Extended Data Fig. S4c: The PIE061 label at the bottom should be changed to PIE060.

Fig. 3: For consistency/clarity, I would use either Ukraine CA or Ukraine Eneolithic throughout the paper, but not both.

Fig. 3: It would be helpful for the reader if the authors could use a different colour scheme in the proximal modelling because the current colours are very similar to each other (for instance SEE1, PTK CA and CA merge 6).

299: Is this true? I see many cases with $|Z| > 3$ within the last two panels in Extended Data Fig. S6, indicating that the Ukraine Eneolithic groups are not cladal with Steppe Eneolithic/Maykop groups to the exclusion of HG groups.

321: In Extended Data Fig. S8, I do not see any model with ~10% of Steppe Eneolithic/Maykop ancestry for any of the three groups (MAJ, KTL_A and USV), the lowest is ~25%.

403: What does pre-M269 mean here? There are two Ukraine Eneolithic males with R1b, but these are low resolution calls with unknown subtypes. Using pre-M269 could be confusing as some people could understand that these males are ancestral for M269, when in fact we do not know.

415: In the Lazaridis et al. Science 2022 paper they argue that the farmer-related ancestry in Yamnaya is not accompanied by WHG-related ancestry, and thus it must have come from south of the Caucasus, not from mainland Europe. Here, the authors support an origin in eastern Europe, so it seems a good place to discuss both hypotheses and to explain why one of them is preferred over the other.

436: This is true for Britain, but the genetic data do not show a near-complete autosomal genetic turnover in the Iberian Peninsula.

939: I would use "fitting" models rather than "significant" models, because $p > 0.05$ are considered non-significant p-values.

Sup Data C: The title has a couple of typos.

Author Rebuttals to Initial Comments:

Point-by-point Response to Reviewer Comments for Nature manuscript 2022-08-12988

"Early contact between late farming and pastoralist societies in southeastern Europe"

We received **97** specific comments from three reviewers. We present our responses to each comment below (in blue). Changes made to the revised manuscript files are highlighted in yellow.

Referee expertise:

Referee #1: archaeology

Referee #2: evolutionary genetics, aDNA

Referee #3: aDNA

Referee #1 (Remarks to the Author):

A

- 1) Concise summary of the archaeogenetic data and results. The results of the genome-wide analyses are linked to the transfer of innovations and ideas in the 4th millennium BCE, an aspect that is briefly addressed in the 'Discussion' section but not elaborated on.

Thank you. We prefer to remain careful in the interpretation and discussion of the genomic data by providing the archaeological context and an outline of the cultural and socio-economic changes that happened broadly around the time of the genetic transformations. It is unclear what type of elaboration is desired. Furthermore, potential concrete narratives of the form "group X invented technology Y and passed this on to group Z while also mixing with each other" would be too simplistic and close to the 'pots = people' fallacy. Evidence on the archaeological (what was invented where first?) as well as the genetic side (ancestry profiles do not equate concrete populations but rather a momentary stage on a continuum) is too vague to be aligned in a causative model framework.

B

This archaeogenetic study is of great relevance as it provides new insights by presenting genome-wide analyses for the eastern Balkan region and the northwestern Black Sea area during the 5th, 4th and early 3rd millennium BCE. The northwestern Black Sea region has so far been a region for which only a very small amount of archaeogenetic data is available. With 134 sampled individuals from both areas, a new large data base is provided which traces genetic continuities and integrates into archaeological knowledge.

C&D

Lines **162 to 371** present the genetic data analysed using bioinformatics. As an archaeologist, I cannot competently judge the validity of the statistical methods and the quality of the data. Most of the statistical methods are generally known to me from other population genetics articles. Therefore, I can only comment briefly on the presentation of the data, again from an archaeologists' perspective. The data and their results are presented in the text in a well comprehensible way. Frequent reference is made to the relevant figures and tables within the text and in the extended and supplementary materials.

E

- 2) The study with its three main statements is divided into three chronological sections. For the individual chronological sections, there is a bias in the geographical regions from which samples are available. This is certainly related to the availability of suitable sample material, but is not addressed explicitly in the text.

We agree that there is a geographic bias with respect to each chronological section, which is largely caused by the fact that there is a marked hiatus following the demise of the tell settlements in south-eastern Europe; a phenomenon that is well-known in the archaeology of the region, and is explicitly mentioned in the introduction.

In addition, there are not nearly as many graves from the subsequent settlement areas to the north (i.e., Cucuteni-Trypillia) as we would wish to study (see Immel et al. 2019 for a few

notable exceptions), which makes this inherent, taphonomic, geographic and chronological bias difficult to overcome. Other regions are limited by accessibility, insufficient documentation and preservation or conservation conditions (old excavations). Most crucial for our study, the North Pontic region, part of today's Ukraine, will be out of bounds for the foreseeable future.

We argue that our study is not more or less biased by the natural limitations of taphonomy than any other study on ancient human DNA, and provides a solid experimental setup.

The largest data set is available for the eastern Balkan region in the 5th millennium BCE. The genome-wide analyses prove genetic continuity, for single individuals from tell settlements relatedness could be proven. An important additional piece of information from genetics is that no evidence of pathogens has been found. So no evidence for an epi- or pandemic could be provided from genetics which is of interest for the ongoing discussion about the causes of the end of the tell in the Copper Age.

- 3) The second data set is available for the 4th millennium BCE. For this, only individuals from the northwestern Black Sea area were analysed. The fact that there was contact between the cultural groups distinguished by archaeologists in this period, which is also reflected genetically, is not really surprising in view of the regional proximity of the sampled sites. More noteworthy here is the diversity of genetic components that point to Exchange with communities in different directions. There might be a connection here with the rather heterogeneous burial customs, as described by Y. Rassamakin for the steppe Region, which is not addressed in the paper.

Thank you, but we respectfully beg to differ with respect to the statement that our finding is not surprising. While there is indeed evidence for cultural contact in the archaeological assemblages, this is by no means pre-empting interaction at the population level or actual proof thereof.

In fact, our study is the first to show formal and compelling direct evidence for admixture between late farming-associated groups in south-eastern Europe and those of transitional pastoralists from the steppe zone at the population level. In addition, the fact that some of the admixed groups such as Usatovo and Majaky display stabilized mixture proportions attests to biological contact that occurred much earlier than anticipated and involved mutual exchange.

We agree with the notion that the genetic heterogeneity of the Eneolithic sample on the whole might be reflected by the heterogeneous burial customs, but caution that the sample size per burial type is still too small to make informed statements on the latter. We also note that a direct comparison to carriers of these various burials customs is not yet possible.

- 4) For the early 3rd millennium BCE, data are mainly presented from the Balkan area, only from Mayaki in the Odessa region were two individuals of the Early Bronze Age genome-wide analysed. A 'mosaic of ancestries' (**line 425**) has emerged in the Balkans in this period. This too seems to go hand in hand with the archaeological evidence, looking at the new funerary customs in burial mounds associated with the Yamnaya Culture. On the other hand, some tell settlements were being resettled, and their pottery is clearly different from the vessels that are found in Yamnaya graves. In the East European steppe, on the other hand, the 'mosaic of ancestries' described in this study for the 4th millennium BCE for the north-western Black Sea area seems to disappear. The aspect of a relatively uniform genetic 'steppe' or 'Yamnaya' ancestry is not addressed in this paper. Can this change in genetic composition from the 4th to the 3rd millennium BCE be assessed more concretely for the steppe area on the basis of the data presented here?

Of note, we report genome-wide data from three individuals from Early Bronze Age Majaky.

We agree that the question about a possible contribution of the Eneolithic individuals from the early contact zone to the formation of 'steppe' ancestry was not explicitly addressed in the earlier version of the manuscript. We have now included the Ukraine Eneolithic individuals as potential sources for the EBA stratum in the ancestry modelling.

We note that the actual events that led to a uniformity of steppe ancestry type is outside the scope of our manuscript and to our knowledge will be addressed by colleagues in ongoing projects with a special focus on Yamnaya pastoralists. We nonetheless performed additional tests to address the apparent uniformity of 'steppe ancestry' associated with steppe pastoralists, including relevant new EBA data from the western contact zone, namely Majaky_EBA and Boyanovo_EBA individuals. We also explore how this ancestry potentially formed and how it differs from the preceding Steppe Eneolithic ancestry found at sites in the North Caucasus and the Samara region (e.g., Khvalynsk).

Inspired by previous work (Mathieson et al. 2015/18, Wang et al. 2019, and Lazaridis et al. 2022) we explored models involving sources that could have contributed to the formation of this 'steppe' ancestry, and by including the new genomic data from Eneolithic Ukraine presented in this study. We indeed find subtle differences between all EBA pastoralist groups. These findings suggest that despite the apparent genetic homogeneity at face value (in PCA for example), that there is subtle regional structure, which is driven by neighbouring geographic regions, and the respective local substratum.

We now report these findings in the revised main text and respective figures and provide details in the Supplementary Tables.

Overall, the conclusions are fortunately written in a very differentiated way. Too far-reaching interpretations are avoided. I fully agree with the last paragraph, because only by expanding the genetic data sets and their interdisciplinary interpretation complex cultural dynamics in different parts of Eurasia can be understood.

Thank you.

F

- 5) Article - Figure 1b green squares: probably 'Yamnaya' or 'Steppe' ancestry, but not mentioned in the list of published data

Thanks for pointing this out. We have added the correct label to the legend.

- 6) **Extended data - Fig. 1** presents finds from a) Pietrele and b) Usatovo and c) a grave from tell Yunatsite. The illustrations of artefacts have no relation to the text. Why were these finds depicted, what should they show? More appropriate might be illustrations showing the tell settlements in a photograph or topographical plans or a grave of the Usatovo culture. It would probably even make more sense to add illustrations to the sites from which samples were taken and which are described in 'Supplementary information'. In this case Figure 1 in 'extended data' could be dispensed.

Thanks. We agree and have provided additional aerial views of the tell sites, topographic maps or characteristic grave types from sites included in this study (where available). As the Supplementary information document is not supposed to contain additional figures, we also show exemplary grave goods for the Usatovo and Cernavoda cultural groups in Extended Data Fig. S1.

G

The literature cited is well selected and corresponds to the (archaeological) state of research.

- 7) In some places the numbering seems to be mixed up, please check: e.g., ref. 55 probably refers to 54; ref. 56-60 probably to 55-59, etc.

Many thanks for spotting this mix up. The numbering of the references is updated.

H

- 8) A clearly structured and well written paper.

We thank reviewer 1 for the constructive feedback and the positive evaluation.

Referee #2 (Remarks to the Author):

Penske et al. present genome-wide data and C14 dates from 100+ individuals spanning 3000 years from the Early Chalcolithic to the EBA from East Balkans to the North-West Pontic region. The manuscript effectively describes changes in material culture in the region through this period. The authors then perform in depth analysis of their dataset to demonstrate admixture between Anatolia_N-related groups and groups related to pre-Neolithic likely local HG groups in the region, and later, variable levels of Yamnaya ancestry in the broad region. They also report interesting cases of single individuals with distinct ancestry. The work further involves description of IBD sharing within and between Chalcolithic villages.

- 9) The population genomic analyses are comprehensive. However, I felt that the manuscript's main claim, i.e. that it connects major changes in technology and other aspects of material culture during this period, with human movement patterns in the region, needed further support beyond broad parallels. For instance, this could involve testing hypotheses related to sources and timing admixture that could explain specific developments.

While we agree that this would be most desirable, we caution that the scale and power of resolution of the archaeological and the genetic data are insufficient to go beyond broad parallels (see also response to 1).

From a genetic perspective, we can show through genetic ancestry modelling that, for example, many steppe pastoralist groups require more than two sources of ancestry. In addition, the estimation of admixture dates assumes simplified models of single pulses of admixture. Often, several alternative models are supported, and with these the date estimates become increasingly broader or imprecise if more admixture pulses or continuous admixture were the case.

With respect to the archaeological evidence, the time and place of origin of technological, economic and social innovations is also very difficult to define with absolute precision. More often, various innovations, such as the chariot and the wheel, anthropomorphic stela, recipes of metal alloys, or religious symbols spread rapidly over great distances, irrespective of the genetic ancestry of the groups involved, attesting to cultural rather than demic diffusion.

However, details about technological, economic advances and social transformations observed in the archaeological record are provided in the introduction and the Supplementary Information and serve as contextual points of reference. As a result, and despite the limitations of both sources of evidence, the integration of context and genetic data allows speculation as to when (in broader time intervals) contact and exchange was possible, which in occasion can be affirmed by signals of admixture.

Wherever we felt that it was sufficiently supported we provided time windows, the directionality of the transfer, and details about what could have been exchanged in the main text.

#####

I have some questions and suggestions below for the authors:

- 10) **L228:** "and burial rites independent of genetic relatedness (Extended Data Fig. S4c)."

I cannot see how Extended Data Fig. S4c supports this statement. If the authors tested whether rites were unrelated to genetic groups, it would be helpful to provide this.

Thanks. We agree and have rephrased this sentence, which now reads: "However, analysis of the runs of homozygosity (ROH) per individual using hapROH⁵¹ indicates low levels of parental background relatedness suggesting relatively large group (effective population) sizes, consistent with previous observation across early farming societies⁵¹ (Extended Data Fig. S5).

- 11) **L229:** "Taken together, these findings support the cross-regional significance of southeastern European tell sites, and reflect the cultural interconnectedness documented in the archaeological record.

This statement would benefit from some comparative support, e.g. how do the patterns observed differ from those in other regions and/or periods, and how do the patterns suggest "cultural interconnectedness"?

In archaeology, cultural connectedness is primarily inferred from similarities in types and styles of pottery and other artifacts, as well as settlement and funerary architecture. The Gumelnița–Kodžadermen-Karanovo VI complex that is referred to here combines three regional expressions of a highly similar cultural horizon.

We updated the sentence, which now reads: "Taken together, these findings reflect the settlement density and the wide-spread cultural, rather than close genetic, connectedness of the Gumelnița–Kodžadermen-Karanovo VI complex documented in the archaeological record, in line with the cross-regional significance of southeastern European tell sites (Govedarica & Manžura 2010).

- 12) In the Ukrainian Eneolithic, Kartal is described as showing high heterogeneity in ancestry, which is very interesting. But the reader is left wondering what this exactly implies. For example, are there material culture differences between Kartal and Majaky that could be related to differences in ancestry heterogeneity? The described heterogeneity in ancestry among EBA sites could also be studied in the same manner.

Thanks. This is an interesting question, but the underlying assumptions are fairly close to the stereotypical *pots=people* scenario, which archaeogenetic studies are often accused of asserting. As a team of archaeologists and geneticists, we are surprised that we are encouraged to consider this mode of thinking and wonder what exactly the expected implications would be.

In our view, the differences between the three sampled Eneolithic sites with respect to genetic hetero/homogeneity are best explained by the geographic location of the sites. Majaky and Usatovo are close to modern-day Odesa, north of the Dniester River, and are ascribed to the Usatovo culture while Kartal at the Danube River mouth is described to the Cernavodă I culture.

As such, both are located in the interaction zone between economic and cultural traditions associated broadly with Copper Age farmer and steppe pastoralist groups. This is seen in the archaeological record of both Usatovo-, and Cernavodă I-associated groups, where material culture and funerary rites combine elements that are otherwise characteristic for the Eneolithic in the North Pontic steppe as well as Cucuteni-Trypillia and the Gumelnița–Kodžadermen-Karanovo VI complexes.

The archaeological record of the eight Kartal graves studies here indeed documents a heterogeneous mix of elements, but not to the extent that individuals with a higher amount of 'steppe ancestry' were consistent with steppe Eneolithic cultural elements or position/orientation of the bodies/graves. We caution that the number and types of grave goods are too few, and that the number of individuals is too small to allow for meaningful comparison, other than in terms of broad parallels.

Located geographically closer to the late Gumelnița–Kodžadermen-Karanovo VI complex, we seem to capture a snapshot in time that documents the process of contact and exchange that aligns *temporarily* at the cultural and genetic level, even though we caution that this is not a 1:1 match for reasons mentioned above.

By comparison, both the archaeological assembly and the genetic profiles of Usatovo-associated individuals appear more homogenous, which implies (!) that such a process of contact amalgamation had already occurred earlier, i.e., during the late 5th millennium BCE further to the north.

We agree that this was not explained in the original version of the paper. We address this aspect in the revised Discussion section.

- 13) **L269:** "we also tested alternative scenarios of population interactions and cultural influences that link the northern Black Sea to the North Caucasus"

It would be very interesting to test "cultural influences" indeed, but was this really done in this work?

We agree that the phrasing was not clear and we have revised the sentence accordingly. We tested for different admixture scenarios that were formulated on the basis of visible/detectable cultural influences that are described in the archaeological record. In the process of restructuring the results section, we changed the respective sentences. For examples lines 297-299 now reads: "Based on these cultural influences which also link the northern Black Sea via the steppe belt to the North Caucasus region, we also test for potential influence of North Caucasian cultural groups, such as Maykop...".

Lines 351-354: "Since the archaeological records suggests a cultural contribution of both Steppe Eneolithic as well as Caucasus Eneolithic and Maykop-associated groups (Supplementary Information 2.2), we also tested an alternative scenario which involved a possible admixture between both groups north of the Caucasus and independent spreads westwards."

- 14) **L374:** "The genetic continuity observed among the individuals at the four CA sites [...] extended periods of stability for several centuries as attested to by the archaeological record."

This statement suggests lack of inter-regional migration is linked with "stability", which is not necessarily true.

The authors could also explain concretely what the archaeological record is exactly showing (for those not knowledgeable in the field). How does it concretely differ from the Ukrainian Eneolithic and or the EBA?

Thanks. We agree that 'stability' is of course a relative concept.

All tell sites and necropoles under study show evidence for continuous site use or occupation for at least 350 years, and possibly up to 600 years. In Pietrele, for example, a constant sequence of 8 houses with a similar ground plan could be proven for this period. Moreover, the supply of raw materials and resources remains constant, and the style and type of specialized equipment and tools remain similar over long periods of time. This implies a consistent and robust level of social organization, critical for the effective transmission of cultural knowledge, such as architectural features, economic activities and funerary rites, which together have been interpreted as a period of social, political and economic stability. Hence, in the relative terms of this study, an economic and cultural system as presented by the Gumelnița–Kodžadermen-Karanovo VI complex is thus considered 'stable'.

Please see also lines 77-81 in the introduction, where we emphasize this point. We also relativised our statement in lines 481-484, acknowledging the fact that 'stability' is relative and can only reflect the absence of major cultural and genetic transformations. We agree that our data is not suited to detect intra-group conflicts at the local or sub-regional scale.

However, from a genetic perspective, we interpret signals of admixture as more dynamic times, not necessarily less stable, but nonetheless discontinuous. The genetic ancestry cline of the Ukrainian Eneolithic individuals describes such a phase, while method qpAdm can provide estimates of the relative contribution of two or more sources, which can ultimately be reconciled with the emergence of certain aspects visible in the archaeological record.

The subsequent EBA clearly marks another period of change, which is undoubtedly also visible in the archaeological record (i.e., new burial types/rites, artifacts, funerary and settlement architecture, etc.) as well as in discontinuous genetics, resulting in a mosaic of ancestries and/or a gradient of admixture.

- 15) I thought that the results in section "Genetic influence from the east during the Eneolithic in Ukraine" could be explained in a more succinct manner, and the main conclusion, that the

Ukrainian Eneolithic could be explained as admixture between SEE_CA-related groups and Steppe Maykop-related groups, could be more clearly explained (some more specific notes below). Some of the f4 tests are difficult to interpret, at least for me, and I am not sure how they contribute to the picture. Also, whether the intriguing genetic heterogeneity within the Kartal population can be attributed to temporal differences is an obvious question, but is not addressed

In L287-L288 of the original manuscript we described a Spearman correlation test to investigate whether the position of the Ukraine_Eneolithic individuals, and therefore the genetic heterogeneity, is correlated with their respective ¹⁴C dates. The Spearman rank correlation coefficient ρ was 0.113, but since the associated p-value was 0.6656 (> 0.05), the correlation coefficient is not statistically significantly greater than zero. Hence, we find that the structure of genetic heterogeneity and the ¹⁴C dates of our samples are not significantly correlated.

#####

Minor points/suggestions:

16) **Fig 1.** Perhaps indicate that titles of panels c refer also to those of panels b?

Thanks. We updated Fig. 1 by adding dashed lines to make the consistent division into three chronological periods clearer and added a statement to this effect in the figure caption.

17) Perhaps better show the other way round, with older on top?

We decided to follow an archaeological viewpoint, in which chronologically older strata (time periods) are on the bottom and younger ones on the top. This appears also more intuitive to us, but we agree that it is entirely subjective.

18) Dark green small squares - I couldn't find them in the legend. Is it Novosvobodnaya?

Thank you for pointing out the missing groups in the legend. The dark green squares are Yamnaya-associated individuals from the Caucasus, the Samara region and North Pontic in today's Ukraine, and these have now been added to the legend.

19) The authors could also consider multiplying PC1 by -1 to have the axis follow the east-west direction, just to help the reader accustomed to such PCA plots - just a suggestion.

Thank you for your suggestion, but the inversion of PC1 would in fact have the opposite effect. The orientation of the PCA plot as currently shown broadly reflects the first publication that described large-scale human population data from Europe (Novembre et al. 2008, Nature), which has since become a mainstay and is used widely by many groups in the field. We believe that most readers will be familiar with this orientation and are able to infer geography from it.

20) **L154:** numbers appear inconsistent with those in L145. Perhaps the 1324 individuals are both modern-day and ancient?

Thank you for noticing this inconsistency. The larger numbers referred to an older reference dataset that contained more modern-day populations than the reference dataset that we used in the final version of our analysis. We updated the numbers in the caption of Fig. 1.

21) Some numbers related to sample sizes seemed inconsistent:

L140: "we report genome-wide data for 134 (out of 216 attempted individuals)"

L645: "Together, a total of 130 individuals yielded sufficient genomic data for downstream analysis."

Thanks. This was a typo, and we have corrected the number to 135 in the methods sections.

22) **L143:** "114 new radiocarbon dates"

L613: "we obtained direct 14C dates for 122 individuals"

Thanks. The inconsistency arose from different approaches to counting (e.g. including/excluding duplicate dates and duplicate individuals, and/or only including individuals for which we also report successful retrieval of genome-wide data). The correct number of newly obtained 14C dates for successfully typed single individuals (after merging identical individuals) is indeed 113. We have updated the main text, the methods section and the chronology in Fig. 1a, accordingly.

23) "Trypillia culture": The approximate period could be mentioned in the text?

The respective time period for the Cucuteni-Trypillia complex was added to the text (L93), as well as references supporting the date range.

24) **Extended Data Fig. S2** and some of the other supp figures: The f4 configuration is such that positive values (to the *right*) indicate affinity to "test", while the test pop names are on the *left* axis. This was confusing for me.

We apologize for the confusion. Test populations are given on the y-axis and not on the x-axis as erroneously stated. We have revised the figure and corrected the figure caption accordingly.

25) **Fig 2:** The sample used in the figure (test2 populations) should be described in the caption.

Since test2 comprises a large number of different populations that is too large to be listed in the figure caption, we added a note to the caption saying "test2 includes all ancient populations in the respective time bracket" and now also point the reader to Supplementary Data C where all test populations are listed.

26) Honestly, I was also unsure how much information the figure conveyed, so the authors may also consider moving this to the supplement - just a suggestion.

Thank you for the suggestion. We prefer to keep the figure as it better *maps* and thus illustrates the shifting genetic affinities through time with respect to geography, a pattern that can only indirectly be inferred from the PCA.

27) **L176:** "Moreover, outgroup-f3 statistics suggest local genetic continuity throughout the CA in this region"

The f3 pattern looks fine, but would it not be preferable to test this by showing f4-based affinity between SEE CA groups relative to neighbouring groups and/or time periods? Or compare f3 values statistically?

We agree that outgroup-f3 statistics are not an appropriate test for genetic continuity. What we meant instead was genetic *homogeneity* across the wider region over a considerable period of time. We changed the respective parts in the main text, so that it is now clear that we talk about homogeneity and not continuity.

The mentioned f4-statistics with respect to locally preceding SEE N groups were already included (Supplementary Data D).

These f4-statistics provide the rationale for the selection of proximal sources in qpAdm ancestry modelling. The fact that we can model each CA group as a single-source derivation from the local Neolithic substrate implies a reliable extent of genetic continuity through time. We have refined the wording in this paragraph to make our argument clearer.

- 28) **L184:** The mentioned shift on the PC space is not super clear. Perhaps include a more focused version of the PCA?

Thank you. We agree that the shift was indeed barely visible, therefore we added a zoom window into the Copper Age PCA to highlight the genetic difference between the majority of the Neolithic groups and the newly presented groups and individuals.

- 29) **L194:** Perhaps add another figure to point out PIE060 in PC space? Or add a marker on Fig 1 to indicate this individual?

Thanks. We have labelled PIE060 and PIE039 in Fig. 1, as both are mentioned in the main text.

- 30) **L200:** A single outlier individual, PIE060, with 30% WHG ancestry and an admixture estimate of 300 years is somewhat surprising. A 300-year-old admixture event should have diffused within the group. Hence, does this imply the individual was a migrant from a region where WHG admixture was more common?

Thanks. The finding of ~30% HG ancestry taken on its own is not too surprising, since individuals with similarly high proportions of HG ancestry were reported from Malak Preslavets and Dzhulyunitsa, approx. 70km and 140km from Pietrele, respectively.

We agree, however, that these proportions should have diffused under the assumption that the mixture was a single pulse and that all other ancestors of PIE060 were local inhabitants of Pietrele with the predominant SEE_CA ancestry profile.

Based on this premise, we have since revisited the model, using an EEF-related ancestry source that is not restricted to PIE_CA specifically. Using DATES with SEE N and Iron Gates HGs as sources yields a wide interval of admixture times, ranging between ~82 and ~832 years before the 14C date of PIE060. This means we cannot precisely determine if it is a recent or an earlier admixture event based on DATES.

Moreover, we repeated the RFMix analysis using the same source populations. Here, we find indeed indications for multiple pulses of admixture, with some that occurred recently (long haplotype blocks) and some that dated further back in time (short haplotype blocks).

- 31) **L202:** "Haplotype blocks as long as" - perhaps add that these are total numbers, not lengths of single blocks.

In fact, here we show the lengths of specific haplotype blocks and not the total number. We can see that the way we had written it could be confusing. Therefore, we added '...', represented as haplotype blocks in mega base pairs. Haplotype blocks as long as ...' to point out better that we refer to the ancestry proportion and with that to the length of the haplotype block on each chromosome.

- 32) **L208:** "typical Mesolithic [...] Y lineage" - could the authors provide a reference or explanation as to how this "typical" was defined?

Thank you for pointing this out to us. We added a reference that discusses the variation of the human Y chromosome: Kivisild, T. *The study of human Y chromosome variation through ancient DNA*. Human Genetics, 2017.

- 33) **L224:** It was not clear to me why burials in a burial ground would be more closely related than burials within a settlement. Some further explanation would be welcome.

This is a misunderstanding. We did not intend to state this as a universal, *a priori* assumption. We provide site-specific explanations, which describe the specific situation in Yuntsite (a destruction horizon with members of the community inside houses and generally narrow

chronology), Varna (a necropolis with shorter site use) against Pietrele, a site, which combines tell and settlement burials around it along a period of up to 350-400 years (Fig. 1a). We have changed the sentence to emphasize a site-specific rather than a generalized observation, which now reads (lines 240-242): "...which can be explained by structure of the sites (a destruction horizon of households and a burial ground with shorter use, respectively, versus tell and settlement burials used over a period of up to 350-400 years)."

- 34) **L275** and **L292**: the same f4 model is reported, but the text reads as if they were different.

Both use f4(Steppe Eneolithic/Maykop, Ukraine_Eneolithic; test, Mbuti).

Is it a mistake or am I missing something?

In general, this section was a bit confusing to me and might be revised to better explain the analyses and the motivation behind them.

Important note:

During the time of revisions, we were able to increase the coverage of nearly all of the Ukraine_Eneolithic and Ukraine Bronze Age individuals substantially, including the addition of a new individual KTL008 (excluded previously due to insufficient coverage). With this more robust dataset, and reflecting on feedback from the reviewers, we re-ran all relevant analyses and restructured our data analyses profoundly, including the use of slightly different groupings and settings.

As a consequence, some of the reviewers' comments refer to parts and analyses that no longer exist and/or have been revised considerably. In such cases we state this explicitly in our individual responses and refer to updated results.

The f4-statistics referred to above are no longer part of the manuscript. Instead, we added a similar set of f4-statistic of the form f4(test, Ukraine Eneolithic; *cornerstone*, Mbuti) to explore the attraction to the so-called '*cornerstone*' groups who present the most common sources of West Eurasian ancestry (Turkey_N, WHG, EHG, CHG) and in which *test* is conditioning on Steppe Eneolithic, Caucasus Eneolithic/Maykop and SEE CA as test groups (Supplementary Data N, Extended Data Fig. S7).

The relevant f4-statistic to test for the attraction to various Neolithic farmer-associated groups now reads f4(Steppe Eneolithic/ Caucasus Eneolithic/Maykop, Ukraine Eneolithic; *test*, Mbuti) where *test* represents the different Neolithic farmer-associated groups. The results of this test can be found in Supplementary Data P)

Both f4-statistics are more focused and have different 'titles' which makes it easier to distinguish between them.

- 35) **L293** and **L293**: "farmer-related groups" and "HG" - I feel "farmer" and "HG" are somewhat too ambiguous terms. Would be helpful to define these more strictly, e.g. Anatolia_N-related or SEE_N-related.

Thanks. We agree that the terms "farmer-related" and "HG-related" are broad and ambiguous and refer primarily to the subsistence strategy lifestyle of these groups rather than the genetic ancestry profile. Nevertheless, for all the groups mentioned in the manuscript, we require umbrella terms that cover the common denominator (here descriptor), which can also be the predominant lifestyle and only secondarily the shared genetic ancestry profile. To avoid overly strong connotations with single reference groups we only use specific groups such as Anatolia_N (now Turkey_N) when these are tested or referred to explicitly. In all other cases where any closely related group (cultural and/or genetically) could also be used instead (or is subjected to in a specific test) we prefer to keep umbrella terms as working proxies.

- 36) **L298**: "Ukraine_Eneolithic individuals have a higher affinity towards HG groups when compared to Novosvobodnaya Maykop, but are cladal with other 'Steppe Eneolithic/Maykop' groups". I feel that the use of "cladal" here is misleading; perhaps it could be rephrased as

"cladal w.r.t. HG affinity." Further, there is quite some heterogeneity in the f4 scores, so it might be better to be more cautious in interpretation.

Having revised the entire section on the Eneolithic, this test is no longer part of the manuscript. We added a more focused test to find possible proximal HG sources in the Ukraine Eneolithic individuals with respect to Caucasus Eneolithic/Maykop as a possible pre-Yamnaya proxy. The test reads as follows: $f_4(\text{Caucasus Eneolithic/Maykop, Ukraine Eneolithic; HGs, Mbuti})$ and can be found in Supplementary Data O. The results and interpretation have been integrated into the revised main text.

- 37) **L301**: "Only KTL_A produced significant negative statistics". This is in comparison to Khvalynsk, is that correct? If so, I cannot see how this is supported in Extended Data Fig. S6.

This test is no longer part of the manuscript in the original form. We regrouped the Ukraine Eneolithic individuals, where KTL_A now includes all individuals from Kartal who are modelled independently.

We have included a test for shared drift between Ukraine Eneolithic and distal HG groups with respect to Steppe Eneolithic and Caucasus Eneolithic/Maykop (Supplementary Data M, Extended Data Fig. S7) and between Ukraine Eneolithic and proximal HGs with respect to Caucasus Eneolithic/Maykop (Supplementary Data O). The results and interpretation have been integrated into the revised main text.

- 38) **L307**: "Significant negative results ($|Z| \geq 3$) indicate shared drift between KTL_B/KTL005 and the tested HGs which is the case for Tell Kurdu and for two HG groups when compared to YUN_CA." - I'm sorry but I can't see this result in Extended Data Figs. S5 and S6. Would be great if the authors clarify this further.

This specific test is no longer included. Instead, we added f_4 -statistics of the form $f_4(\text{Steppe Eneolithic/ Caucasus Eneolithic/Maykop, Ukraine Eneolithic; test, Mbuti})$ in which *test* represents relevant European Neolithic farmer-related groups. The new results can be found in Supplementary Data N and are discussed in the main text.

- 39) **L304-309**: "KTL005 and KTL_B fell close to the SEE_CA cluster in PCA and therefore we tested for a possible attraction to HGs with the test $f_4(\text{SEE_CA, Ukraine_Eneolithic; HGs, Mbuti})$. Significant negative results ($|Z| \geq 3$) indicate shared drift between KTL_B/KTL005 and the tested HGs which is the case for Tell Kurdu and for two HG groups when compared to YUN_CA. All other tested SEE_CA share either more drift with HGs or are cladal with KTL_B/KTL005 ($|Z| \leq 3$) (Extended Data Fig. S7, Supplementary Data P)."

This part was highly confusing for me. The motivation appears to be finding the HG contributor to KTL_B relative to SEE_CA. Why KTL_A and other Ukraine_Eneolithic groups were not used should be better explained.

Obviously $f_4(\text{SEE_CA, KTL B; HGs, Mbuti})$ is generally not significant, which I felt should be the main result here.

Due to a new grouping scheme of the Ukraine Eneolithic individuals, in which most individuals of the former KTL_A group are now modelled independently, and in which the former KTL_B group now includes KTL002, KTL004 and KTL005, this test no longer exists in its original form. However, the test that explores excess HG ancestry in KTL_B with respect to preceding SEE_CA groups is now addressed in Supplementary Data N. KTL_B returns non-significant results for WHG ($|Z| \leq 3$) but is significant for CHG and EHGW/SHG ($|Z| \geq 3$). These new results are integrated into, and discussed, in the main text.

- 40) Also, Extended Data Fig. S7 includes the test $f_4(\text{Tell Kurdu, KTL B; HGs, Mbuti})$, but obviously Tell Kurdu is not SEE_CA.

Further, $f_4(\text{Tell Kurdu, KTL B; HGs, Mbuti})$ being negative - what this implies was not clear to me.

This test has also been removed from the manuscript and has been replaced by an analogue. We now test for genetic drift between Ukraine Eneolithic groups/individuals and distal HG groups with respect to relevant European Neolithic farmer-related groups in Supplementary Data M and Extended Data Fig. S2. We also focused primarily on the question of shared drift between Ukraine Eneolithic groups/individuals and proximal HGs with respect to Caucasus Eneolithic/Maykop. These results can be found in Supplementary Data N and are integrated and discussed in the main text.

- 41) Finally, to aid the reader, Extended Data Fig. S7 should be referenced at the end of the first sentence.

This figure has also changed and all qpAdm results are now shown in Fig. 3. All new Extended Data Figures have been referenced appropriately in the main text.

- 42) And again, I'm not comfortable with the (for me) loose use of the term "cladal".

We agree and have replaced this term with "are symmetrically related to X with respect to Y" or "form a clade with X with respect to Y".

- 43) **L306**: typo in "Ukraine_Eneolithic"

Thanks for the catch. The typo has been corrected in the revised manuscript.

- 44) **L311**: "explore a variety of models" - please add what is aimed to be explained by the models

We have rewritten the section about the Ukraine Eneolithic and have added a rationale for each model.

- 45) **L316**: "ancestry for the Ukraine_Eneolithic." - perhaps say "for other Ukraine_Eneolithic populations", given that KTL_B and KTL005 are also part of this group, right?

We have rewritten the section about the Ukraine Eneolithic which also includes a regrouping of the Ukraine Eneolithic individuals.

- 46) **L318**: point missing.

Thanks. We have rewritten the section about the Eneolithic.

- 47) **L351**: "we observed significantly negative f4-statistics for Ikiztepe_LC, indicating that YUN_CA possibly carries more CHG-related ancestry than YUN_EBA." - This might simply reflect the fact that YUN_EBA can be modelled as YUN_CA + Baltic ancestry so that the CHG (or perhaps Anatolia_N) ancestry in YUN_CA has been diluted over time. In any case, I find such an interpretation confusing, honestly.

This test has been removed and replaced with a simpler and more informative form in which we directly test the difference in HG affinity between YUN_EBA and the locally preceding YUN_CA group. These results can be found in Supplementary Data T and Extended Data Fig. S8 and have been integrated into the main text.

- 48) **L376**: "Long distance IBD relationships between sites are consistent with the transregional connectivity visible in the material culture."

This is very nice, but to convince the reader, I believe the authors should at least attempt to discuss the genetic and material culture observations more concretely and in a comparative framework (even if qualitatively). For instance, they could provide some examples where

long-distance IBD relationships are rarely observed and transregional connectivity is also not visible.

We agree that this would be the desired ultimate goal, but need to stress that the quality and density of ancient genomic data is still too low to explore this idea systematically. Finding long-distance IBD relationships is a 'needle in a haystack'-exercise and mostly depends on the availability of good quality data (>500k SNPs) in pairs of individuals, while the absence of such relationships is generally much harder to prove (the absence of evidence is not evidence of absence).

In our study, we took advantage of the increased number of individuals available per Chalcolithic tell site/necropolis, which happen to not only be contemporaneous, but also share cultural affinities, which might maximize the chances of finding long-distance IBD relationships. It is important to stress that a comparison with neighbouring regions, where the data is thinner, is not advised as it would be non-conclusive (as of this moment). At this stage, and while the comparative data sets are still growing in size, testing for IBD between individuals and groups are clearly exploratory methods, which have great potential. We have revised the respective sentences to clarify the current intrinsic limitations of this approach.

- 49) **L391**: "A new major finding from our study indicates early contact and admixture between CA farming groups from SEE and Eneolithic groups from the steppe zone in today's southern Ukraine during the 4th millennium BCE at a time when settlement densities shifted further north."

Unfortunately, I don't see how this statement of "early contact" is directly supported by the results.

If the basis is the observation that SEE_CA could be used as a source for modelling Ukrainian Eneolithic, then: 1) this should be explicitly stated, and 2) alternative sources should also be tested, I believe. It is possible I am missing the point here, but it would surely help the authors to clarify this point.

Yes indeed!

We are somewhat surprised how this point could have been missed given that the entire section titled "Genetic influence from the east during the Eneolithic in Ukraine" on pages 10-13 in the original submission described the horizon of "early contact" in great detail, while the ancestries contributing to the Eneolithic groups were explored using numerous sets of formal f4-tests and qpAdm ancestry models shown in Figure 3, Extended Data Figures and Supplementary Data tables. Alternative models were also explored and their rationale explained in the main text.

Irrespective of what we assume must be a simple misunderstanding, we have thoroughly rewritten the section about the Ukraine Eneolithic individuals and made sure this point, which still stands, will not be missed in the revised version of the text.

- 50) **L397**: reference to literature missing.

Thanks. We have added a reference.

- 51) **L402**: "Y-haplogroups indicative of a farming legacy," - again I feel the use of "farming" is too lax; please specify which archaeological groups are referred to explicitly.

We respectfully disagree and would like to keep a broad category (e.g., Neolithic_farmer) that refers to a wider ancestry cluster, which in this case is associated with the spread of farming practices during the Neolithic. The reason is that the use of archaeological groups would imply that Y-chromosome haplogroups are associated with concrete archaeological groups, which results in a reification fallacy that we would clearly like to avoid.

- 52) **Fig 3:** Green, pink and yellow tones are extremely difficult to distinguish. Please add another visual marker (e.g. stripes).

Please briefly mention in the legend what the population names in the panel titles refer to (e.g. "use of Russia Khvalynsk or Russia Steppe Maykop as sources").

We completely revised Figure 3 on the basis of an improved dataset and reduced the number of different colours which will make it easier to distinguish between the different sources used in the ancestry modelling.

- 53) **L683:** Providing READ parameters and details would be helpful - were the P0 values normalized using the median of each site, or region, or the full dataset?

We updated the description in the Methods section and have a paragraph providing more detail in Supplementary Information Part 7.

- 54) **L717:** I presume there was no multiple testing correction applied - would be good to indicate this.

No corrections for multiple hypothesis testing were applied. We added a statement to this effect in the M&M section.

- 55) **L735:** "We attempted to estimate the admixture time for each target population but it resulted in large estimates with standard error being almost as high as the estimate itself."

It would be good to know how many trials were conducted, and how many were "infeasible". If most trials did not work, this actually reduces the reliability of the admixture date estimate reported in L198. (Our experience with DATES has also been similar, with a very high frequency of unreasonable estimates, which leads to the question of whether the "reasonable" looking ones should be taken seriously.)

On the basis of the experience with DATES shared across the wider population genetics group at MPI-EVA, we conclude that the results differ from case to case and very much depend on 1) the depth of genetic differentiation of the two sources, and 2) if, and how badly, the assumption of a single pulse of admixture has been violated. These limitations are also reported by Chintapalati et al. 2022, *eLife*.

Since the previous results obtained for PIE060, in which we used PIE_CA as local substrate, and which resulted in relatively old admixture dates, indicated that individual PIE060 was not a descendent of the Pietrele community with a recent HG ancestry contribution, we repeated DATES using a combination of southeastern European Neolithic individuals as alternative source, but retaining Iron_Gates as a local Danubian HG source. The new date estimates are also wide, in agreement with the parallel RFMix analysis, which show a mix of longer and shorter HG ancestry tracts, which suggest repeated interactions of the different ancestries or mixes thereof.

Of note, PIE060 is the only male buried at Pietrele, who carries Y-chromosome lineage I2a1-L701, indicative of a potential HG introgression on the male side of ancestors. The estimated admixture matches that of other Balkan_Chalcolithic individuals (Chintapalati et al. 2022, *eLife*), and individuals with similarly high proportions of HG have been reported in Mathieson et al. 2018, *Nature*.

- 56) **L900:** "Significant" - better remove or replace by "nominally significant"

We agree and have replaced it by "Z-scores outside the threshold of $|Z| < 1$ ".

- 57) **Extended Data Fig. S1.** This is very nice, but a more comprehensive description of pottery from all the sites involved would have been even more informative.

Thanks. Unfortunately, we are limited with respect to the total number of illustrations, However, we have updated Extended Data Fig. S1 with additional, available images, including relevant pottery from Cernavodă and Usatovo. For a more comprehensive description of the pottery from other sites and cultures, we respectfully refer to the specialized literature provided in the references of the Supplementary Information.

- 58) **Extended Data Fig. S3.** RFmix of what individual? What is the sample in panel b? Also not clear to me how panel b reflects a M:F "ratio".

Thanks for the pointing out the omissions. We have updated the caption of Extended Data Fig. S3.

- 59) **Extended Data Fig. S7:** Why is Tell Kurdu but not e.g. Ikiztepe_LC or Kumtepe used as a possible proxy for southern gene flow?

This test has been removed based on an improved dataset and a regrouping of the Ukraine Eneolithic individuals. Of note, we explore Tell Kurdu and Caucasus_Eneolithic/Maykop as two proxies for a source of "southern" gene flow, which also accounts for Ikiztepe_LC, which itself falls intermediate between both proxies on the genetic cline.

The two published Kumtepe individuals are both SG data and one of them has only low coverage.

- 60) **Extended Data Fig. S8:** Pink tones are too close. Perhaps use stripes or other additional markers? "Maikop" in the figure legend.

After re-analysing the improved dataset., and based on the reviewers' comments, an extended qpAdm figure is no longer necessary for the Ukraine Eneolithic groups and individuals. Results are now included in Figure 3e.

Supplement:

- 61) - The ecological and archaeological background descriptions were generally clear and well-written. Though some parts were confusing, e.g. when the text on the SEE jumps to Northern Pontic evidence - the connections could be better explained (p 4).

Thanks. We have reworked these sections slightly and have better connected both parts.

- 62) It would also be helpful to support the information with maps showing localities and a timeline.

We have included maps and a chronology of the samples and sites in question in Figure 1 of the main text. Additional maps and individual site chronologies can be found in the primary literature cited for each site.

- 63) - "At the moment of their death, these individuals must have been in miserable condition, most probably due to hard physical work." - Further description could be helpful.

The anthropological examination was carried out by Steve Zäuner, whose report is cited in the description. A more detailed anthropological study is currently in preparation.

- 64) Also, the authors could consider providing images to support the skeletal evidence.

Figures are limited to the main text and the Extended Data Figures section. We therefore respectfully refer to the literature provided in the site descriptions for each site in the Supplementary Information.

- 65) - Do differences with respect to social status or conditions of death (trauma) or burial rituals at all correlate with genetic ancestry? I presume the authors have already checked this - it would also be helpful if they report results even if negative.

The broad genetic homogeneity across all newly reported Chalcolithic individuals did not justify a correlation of genetic ancestry, conditions of death or burial rites.

However, we did double-check the archaeological context upon positive indication, that is, when one or more individuals were identified as 'genetic outliers' with respect to the majority of individuals from the same group/site. In these cases, we genetically characterize these individuals (PIE060, BOY019, and YUN041) independently. We note that their respective burial context does not differ from other individuals at the same respective sites.

As far as the destruction horizon at Yunatsite is concerned, we find no genetic differences between individuals from inside and outside of the houses, or other Chalcolithic burials at the site, to the limits of the genetic resolution.

A detailed analysis on wealth and the grave goods from the necropolis of Varna has been conducted and published by Krauß *et al.* 2017, which is cited in the Supplementary Information. We do not detect any genetic differences between the individuals that have more grave goods than others but this also was not expected based on other archaeological findings on cultural exchange and movement of people as we mention in the introduction and elaborate in the discussion.

While we acknowledge that a site like Kartal with genetic heterogeneity would warrant such an explanation, we caution that the number of individuals is too low to yield meaningful statistical support (see also response to point 12).

Overall, while we acknowledge the importance of 'outlier' individuals, the inferences drawn from single observations can only be anecdotal at best.

- 66) - Permits for use of the material are increasingly expected to be published as supplementary material. It would be beneficial if the authors could add these to the supplement.

Permissions for minimally destructive sampling as part of genomic analyses were granted in all cases to the principal investigators and partners in the form of joint collaboration agreements. These are usually not disclosed as supplementary material.

- 67) - "4. Rationale for selecting source populations". This part was not easy to follow for me - a revision would help the reader.

With the help of the reviewers' feedback we were able to narrow down the rationale for selecting the source populations and have updated the respective paragraphs in the Supplementary Information. This should make it easier for the reader to follow our thought process.

- 68) Also, I did not understand why cladality was only tested using "HGs". Why not also use e.g. Anatolia_N as test population in f4 analyses?

In the submitted manuscript we tested for additional HG ancestry after having chosen the best proxy for the SEE_CA groups. However, having realized that this test is redundant we removed it. Instead, we now only test for excess affinity to HG ancestry for PIE_CA and PIE060 since for both no symmetrically related group could be found across all tests (Extended Data Fig. S2 and Supplementary Table D). For both groups/individuals the closest groups were chosen and both returned significantly negative results for WHG, indicating that additional WHG ancestry is needed. Following this observation, we next used $f_4(\text{SEE}_3, \text{PIE_CA}; \text{HGs}, \text{Mbuti})$ and $f_4(\text{SEE_N}, \text{PIE060}; \text{HGs}, \text{Mbuti})$ to find the best HG proxy for both. Further, we argue that an additional test with Turkey_N would be not informative as we already tested 'unadmixed' Neolithic groups using the f_4 -statistics shown in Extended Data Fig. S2, which also includes Turkey_N.

69) **Table K:** I couldn't understand why YUN027-YUN026 and YUN037-YUN038, which share high IBD (Table L), were not represented among READ results?

Thanks for pointing out these inconsistencies.

The READ results for the early Bronze Age, including YUN037-YUN038 had been overlooked and are now added to Supplementary Table J.

Concerning the first-degree relationship between YUN026 and YUN027: we found an error in the naming of the respective bam files, which led to the observed result in IBD. A new IBD run with the correct header of the bam files could solve this issue, yielding consistent results. Both individuals are now marked as unrelated in READ and IBD.

70) **qpAdm analyses:** Would be helpful to explain how "feasible" was defined.

We agree and have added an explanation of what 'infeasible' means to the methods section of the Supplementary Information.

71) How was the microbial screening performed? Please add to Methods.

Apologies for the oversight. We added a paragraph describing the pathogen screening in the Methods section.

72) The authors could consider comparing relative EHG and WHG affinities in SEE_CA, e.g. by plotting them against each other.

Thank you for the suggestion. Since we find groups that are symmetrically related to VAR_CA, YUN_CA and PTK_CA with respect to test populations (Supplementary Data D and Extended Data Fig. S2 respectively), we do not see the need to explore this in addition. For PIE_CA we determined KO1 to be a suitable proxy (Supplementary Data F). PIE060 shows significantly negative results for all tested HGs. However, the f_4 -values and associated Z-scores indicate that PIE060 shares excess drift with WHG-related ancestry over EHG-related ancestry, albeit without excluding a contribution of the latter. This observation was further confirmed in proximal qpAdm ancestry modelling described in Supplementary Data H and shown Fig. 3d.

Referee #3 (Remarks to the Author):

This manuscript describes a genome-wide analysis of 134 new ancient individuals from southeastern Europe and the northwestern part of Black Sea region. It focuses on three periods involving key transformation in the archaeological record: the Copper Age, the Eneolithic and the Early Bronze Age. I congratulate the authors on the excellent introduction, it certainly helps the reader to understand the archaeological context of the samples under analysis and the relevant questions. The laboratory procedures and bioinformatic processing were performed at MPI-EVA, a top aDNA lab fulfilling all the standards in the field. I have some important comments on the admixture modelling (see below), which I believe could be addressed without much trouble.

73) I do question the novelty of the findings, since the Copper Age and Early Bronze Age data consolidate with more samples the results reported in previous publications (e.g. Mathieson et al. 2018, Nature). In the case of Ukraine Eneolithic, Cernavodă I and Usatovo data are reported for the first time in this manuscript, but the findings are quite similar to the ones described in Immel et al. 2020, Scientific Reports for the neighboring Cucuteni-Trypillia, even the titles of the two manuscripts are quite similar.

While we agree that our manuscript builds on previous publications, such as Mathieson et al. 2018, to which our groups also contributed the first results from the CA tell site Yunatsite, we would like to emphasize that we specifically address questions that arose or were left open in previous publications. We do this by contributing sites of great significance (Varna, Pietrele, Yunatsite), which have been excavated according to highest and modern standards and which thus represent critical archaeological and contextual reference points. Archaeological

findings indicate a time of the beginning of warfare but also a time of intensive trade and cultural exchange between the major sites in Bulgaria and Romania. We could show that besides these events, the Copper Age is characterized by genetic continuity from the Neolithic onwards. Testing for genetic interconnectivity using IBD-sharing showed that the people living in tells and large settlements had very little to no genetic connection to each other. This can only be shown by analysing a large number of individuals per site. This is something that Mathieson et al. 2018 did not address in their publication since their aim was a genetic overview over this (still) understudied region. This solid genetic and archaeological foundation is needed to explore the genetic ancestry of the newly reported Cernavodă I and Usatovo data, which are the central novelty of our findings.

We disagree with the comment/notion that the new Ukraine_Eneolithic data are quite similar to the Cucuteni-Trypillia-associated data reported in Immel et al. 2020, *Scientific Reports* or in Mathieson et al. 2018, *Nature*. In fact, we can show that this statement is incorrect.

Mathieson and colleagues already characterized the genetic ancestry of CT-associated individuals correctly as a mix of Anatolian_N, WHG and EHG ancestry (see Figure 3 in Mathieson et al. 2018, *Nature*).

Further, we also find that the CT-associated individuals do not carry any steppe-related ancestry, in contrast to what had been reported described in Immel et al. 2020, and provide additional tests in support of this revised finding. In brief, we re-analysed three individuals from Immel et al. 2020 (Pocrovca 2, the fourth individual, is not available publicly) and have added a detailed paragraph describing the results in the Supplementary Information under point 5.

In fact, the main difference in genetic ancestry from SEE_CA groups is the higher amount of hunter-gatherer ancestry that is best represented by Ukraine_Mesolithic/Neolithic groups.

In sum, we find that Moldova_CTC shows no major additional attraction to either Yamnaya_Samara, Russia_Steppe_Eneolithic, nor CHG (a key component of 'steppe ancestry').

We see that the results presented by Immel et al. 2020 can lead to a misunderstanding and confusion on the reader's side. The main text briefly addresses this issue and refers to the detailed results provided in section 5 of the Supplementary Information.

Of note, the quantity and quality of the contributing source of Eneolithic steppe ancestry is clearly different and visible in the newly reported Cernavodă I and Usatovo-associated individuals of the Ukraine_Eneolithic data. As a result, the findings we present here in this paper, *especially* for KTL_A, MAJ and USV are new, and have not been shown before in any publication.

Major comments:

- 74) Throughout the paper, the authors are not consistent in their choice of significance thresholds, in some cases using a p-value threshold of 0.05 ($Z=1.96$) like in qpAdm, in some f4-stats $Z=3$ and in other f4-stats $Z=1$. For instance, in Extended Data Fig. S2, many f4-values are highlighted in orange as significantly different from 0 based on $|Z|\geq 1$, which translates into a p-value threshold of 0.16 (one tail). This seems like a too stringent cut off, as a good portion of these orange f4 deviate from 0 just due to random noise.

The reviewer makes an excellent point, and we apologize for not being as transparent in the text describing these different "significance cutoffs". However, we must point out that a p-value cut-off of 0.05 for qpAdm and the Z-score cut-off are not directly comparable, for the following reasons. (a) It is not clear what the distribution (or even the family of distributions) of f3- and f4-statistics are, and while research into this has been made, no analytic solutions have been identified. The generally accepted, *conservative* cut-off for the Z-score of $|Z|\geq 3$ was settled upon by the original authors of the paper, and supported by later simulation studies (Patterson *et al.* 2012, Genetics and Harney *et al.* 2021, Genetics). However, while the p-value, to which a Z-score from an "f-test" is mapped to is not clear, it has been shown to be qualitatively reliable. Regarding the p-value obtained from qpAdm, $p>0.05$ is used as the cut-off for which the null hypothesis of a model fit is "not rejected", unlike in classical

statistics, where a $p < 0.05$ is used to reject the null hypothesis. This is all explained in the publication referred to above, but the quality and selection of outgroup sets, and the uniqueness and depth of ancestry sources (among other variables) can skew the null-distribution, meaning that it is not clear that 0.05 is an "analytically sound" cut-off, but, this again has been shown to perform well via simulation.

Note though that the $|Z| \leq 1$ cut-off method (Extended Data Fig. S2) was not a formal testing framework, and instead used as a heuristic search statistic. This concept came about as taking every possible *cladal* Neolithic source (for the Neolithic individual and the Chalcolithic groups) yielded unbalanced sample sizes for statistical testing due to the huge abundance of candidates, and the relative homogeneity of Neolithic ancestry prior to the Chalcolithic. Hence, we chose $|Z| \leq 1$ to conservatively search for potential Neolithic sources, which we then merged as a single source population. Clearly, using $|Z| \leq 3$ would have selected more statistically cladal populations, but the purpose of this analysis was never to identify every potential Neolithic source population. Instead, we wished to find an automated, non-biased method to identify potential "most-cladal" populations to merge into *one* genetic group, without selecting too many populations that we had (a) an unbalanced sample size for just Neolithic ancestry in our modeling, and (b) an unbiased method for choosing of samples. We are aware that "all" Neolithic populations with $|Z| \leq 3$ are, by the above definition, equally "cladal", and hence setting $|Z| \leq 1$ just takes a theoretically random subset from these candidates.

- 75) **177:** The outgroups-f3-statistics mentioned here compare CA groups against contemporaneous SEE groups. When trying to test continuity over time, one should compare against the earlier period (in this case the Neolithic), as the authors do in line 181 using the $f_4(\text{SEE_CA}, \text{SEE_N}; \text{CHG}, \text{Mbuti})$ test. More importantly, outgroup-f3-statistics are not good tests for continuity, even when you are comparing against the earlier period. Let's say you have 30% discontinuity, you will likely get high outgroup f3 values against the earlier populations that shares 70% of the ancestry with your test. I agree with the authors that CA groups are largely continuous to the Neolithic groups, but this statistic is not the right one to test for continuity.

We have rephrased this paragraph, making clear that the outgroup-f3-statistics are neither the sole nor most appropriate test for population continuity. We largely build our argument on the follow up f_4 -statistics and the fact that we can model CA groups directly with preceding, local Neolithic groups. See also response to point 23).

- 76) **qpADM analysis:** Many models (often quite different from each other) seem to work for a given Ukraine Eneolithic group. This is likely because the qpADM set up has little power to reject non-fitting models:

-KTL B and KTL005 are clearly shifted from the SEE CA cluster in PCA. Thus, unless this shift is the consequence of genetic drift rather than admixture, it should be possible to reject one-way models with SEE CA groups. Instead, several SEE CA groups result in good-fitting one-way models for the ancestry in KTL B and KTL005.

Important note:

During the time of revisions, we were able to increase the coverage of nearly all of the Ukraine_Eneolithic and Ukraine Bronze Age individuals substantially, including the addition of a new individual KTL008 (excluded previously due to insufficient coverage). With this more robust dataset, and reflecting on feedback from the reviewers, we re-ran all relevant analyses and restructured our data analyses profoundly, including the use of slightly different groupings and settings.

As a consequence, some of the reviewers' comments refer to parts and analyses that no longer exist and/or have been revised considerably. In such cases we state this explicitly in our individual responses and refer to updated results.

In the process of re-running the analyses we also tested and revised our OG settings. Therefore, we are now able to reduce the amount of well-fit models for many of our groups. Still, the genetic similarity of many single sources has the effect that there are several

alternative models that cannot be rejected, even when using competing or rotating model approaches. However, the new set of models presents a more streamlined and clear approach that is easier to follow. The new results are integrated in the main text and Fig. 3, and were added to revised versions of the Supplementary Data tables.

- 77) -MAJ can be modelled as a 2-way model with 75% KTL_B and 25% either Khvalynsk or Steppe Maykop, but Khvalynsk and Steppe Maykop are substantially different groups and with the other source population being fixed, they should not both result in good-fitting models. Same goes for the good-fitting distal 3-way models with Anatolia_N+CHG+ EHG or WSHG (the outgroup set is clearly not telling apart EHG-related and WSHG-related ancestry).

I recommend the authors to use a different outgroup set with more resolution, perhaps by using populations with better quality data and/or more individuals, chronologically closer to the Refs, and/or differentially related to the Refs.

Thanks. In the process of re-running the analyses we also tested and revised our OG settings. Related to the concern raised here, we added Russia_Siberia_UP to the OG set as a possible way to distinguish between EHG and WSHG ancestry in the distal modelling.

However, some of the targets still return equally well-fit models for Turkey_N+EHG+CHG+WHG or Turkey_N+CHG+WHG+WSHG. This can be explained by the fact that WSHG and EHG ancestries are very similar, with EHG ancestry forming an intermediate position on a genetic cline (drift path) between WHG and WSHG. This also becomes apparent from the modelled proportions where a combination of EHG+WHG always requires less WHG ancestry since this is already accounted for in parts by EHG, whereas the proportion of WHG is higher when combined with WSHG. Thus, to avoid confusion for the reader we plot only the EHG+WHG models in Fig. 3b but for transparency we show all other results in Supplementary Data P.

Further adjustments involved the inclusion of Russia_Steppe_Maikop (Maykop) to the OG set. In a previous approach we had added Russia_Steppe_Eneolithic to the OG and used Russia_Steppe_Maikop as a possible source which resulted in the rejection of all models, and suggests that Russia_Steppe_Eneolithic is the better suited source instead. On this note, the tests shown in Extended Data Fig. S6 and Supplementary Data L shows that all Ukraine Eneolithic groups and individuals share more drift with Russia_Steppe_Eneolithic and Russia_Khvalynsk_Eneolithic than with Russia_Steppe_Maikop. Importantly, the use of rotating model approaches also helped to narrow down the set of possible well-fit models.

- 78) **186-187**: I don't understand how this works. Let's take YUN_CA as an example. You find the Neolithic groups that are cladal with YUN_CA when compared to the four different HG groups, and then you merge all these Neolithic groups and consider this merged group as the best Neolithic proxy for YUN_CA in proximal qpADM modelling. Is that correct? If true, then:

-For YUN_CA, the proximal model features Dhulyunitsa_N but in Sup. Data D, Greece_N and Anatolia_N are also cladal with YUN_CA when compared to the four HG groups. Why aren't Dhulyunitsa_N, Greece_N and Anatolia_N merged and used for the proximal qpADM modelling?

Yes, it is correct that Greece_N and Anatolia_N are also symmetrically related with YUN_CA with respect to all four HG groups. We had indeed run qpAdm initially combining Greece_N, Anatolia_N and Dzhulyunitsa_N as one source, which surprisingly did not result in good model fit for YUN_CA. However, when we used only Dzhulyunitsa_N as the group that is closest in time and space to model YUN_CA the model fit improved significantly. Using a preceding group that is closest in time and space is consistent with the archaeological evidence of the region and this can be supported by the source populations for the other tested CA groups. We realize that we neither specified this appropriately in the main text nor in the Supplementary Information and therefore added more information in the revised documents.

79) -This is a quite convoluted procedure that results in an array of very similar merged groups, like CA merge 1,2,3,etc, containing slightly different combination of populations and each combination used for modelling a different test group. This leads to unnecessary complex figures like Fig 3. Deciding whether to merge or not two Neolithic groups (and the same goes for the CA merges) should be based on cladality tests including those two Neolithic groups against the HG groups (although you could also test against many other groups, as two Neolithic groups can be cladal against HG groups but differently related to other groups). If two Neolithic or CA groups are cladal to the limits of your resolution and you decide to merge groups for analysis, then they should be merged and used as a possible source irrespective of which of your new test groups you are trying to model. For instance, PIE_CA and YUN_CA are now merged in CA_merge_1,3,6,7,8 but not in CA_merge_2 and 5.

We fully agree more and have therefore revised this section of the manuscript to make it easier to follow. For the CA groups we still use the f_4 -statistics to narrow down the Neolithic groups with the highest affinity for further analyses. This is explained in more detail now in the main text and the Supplementary Information. Moreover, the adjustments made to the qpAdm settings and selection of outgroups for the Eneolithic and EBA resulted in fewer well-fit models. Most importantly, we could also reduce the number of different sources. In cases where one or two well-fit models were observed we decided to plot the one with the higher p -value and mention in the respective Supplementary Data section which model had been chosen and why. Overall, the reanalysis resulted in a much clearer and more focused version of Fig. 3, and respective changes to the main text.

Minor comments

80) **132:** The Cernavodă culture appears here for the first time. It would be useful for the reader to comment on this cultural horizon earlier in the manuscript where the archaeological contexts of the different periods are described.

Thanks. We now mention Cernavodă I and Usatovo earlier in the introduction where the developments of the 5th and 4th millennium BCE are described.

81) **Fig.1a:** The published Varna individuals appear on the left of the figure with other published samples, but they display a black outline even though this is used for newly reported individuals.

Thanks. We have adjusted the labels accordingly.

82) **Fig.1:** The legend for the Yamnaya is missing, and same for the Neolithic individual from Pietrele.

We have added Yamnaya as well as the Neolithic individual from Pietrele to the legend.

83) **178:** I am not sure this is entirely true, as previous publications showed that Neolithic groups in present-day Greece did have small amounts of CHG-related ancestry.

We agree that the wording in this sentence was misleading. We have rephased it to avoid confusion and to focus more on the point that CA groups did not receive any extra CHG ancestry. This part now reads: "Thus, f_4 -statistics of the form $f_4(\text{test}, \text{CA}; \text{HGs}, \text{Mbuti})$ were used to find Neolithic groups that form a clade with SEE CA with respect to HGs ($|Z| \leq 1$) and then grouped (Extended Data Fig. S2, Supplementary Data D, E, Supplementary Information Point 5). Of note, we observe no excess affinity to CHG compared to SEE Neolithic groups (Supplementary Data D), a critical component of the later 'steppe ancestry' during the Eneolithic in the North Pontic region, and therefore rule out early contributions of 'steppe-related' ancestry during the SEE CA."

- 84) **181:** If you are trying to test for early contributions of steppe ancestry during the SEE_CA, why not directly using the actual steppe Eneolithic groups instead of CHG, which is a much more distal group?

The test in this form is not part of the manuscript anymore. Instead, we answer the question now indirectly with the f_4 -statistics shown in Extended Data Fig. S2 and Supplementary Data D of the revised version, where we can show that none of the comparisons with CHG as test group result in significantly negative values ($|Z| \leq 3$) indicating that there is no additional CHG ancestry contribution.

We generally use a step-wise exploration, in which we first use distal sources (“cornerstone” ancestries) and then follow up with more proximal groups in both f_4 -statistics and qpAdm modelling. In the particular case above, we use CHG instead of Steppe Eneolithic, which consists largely of CHG+EHG. If we used Steppe Eneolithic in such a test, we would have the confounding issue that a significant attraction to Steppe Eneolithic could also be caused by shared EHG ancestry and thus result in a false positive signal when asking for possible attraction to CHG ancestry specifically.

- 85) **192:** This seems to contradict the PCA observation that SEE CA groups are shifted towards the WHG/EHG cline with respect to Neolithic groups (Lines 184-185).

Not exactly, but we understand that our previous statement might have sounded contradicting. We have revised this paragraph, explaining the rationale of the f_4 -statistics that were used to find the best local proxy among the published Neolithic groups.

Of note, we can model SEE_CA groups also with a single source model since we have chronologically and spatially proximal sources that were themselves already admixed with small amounts of HG ancestry.

- 86) **Extended Data Fig. S3:** The strong correlation of HG segment locations in the maternal and paternal chromosomes (in several cases with the exact same start-end coordinates) is somehow suspicious, and likely indicates problems either in the imputation/phasing step or in the local ancestry estimation.

We agree and have reanalysed the data using the updated source populations from the revised qpAdm ancestry modelling approach (see also response to points 30 and 55).

We removed the number of generations since the admixture date. This allowed us to assess the admixture in PIE060 more broadly, not being tied to the *a priori* number of generations since the admixture event. Moreover, the *-reanalyzereference* option allowed us to take into account the possibility of an already-admixed source.

The new RFMix results show some segments with overlap of longer haplotype blocks on the paternal and maternal chromosomes, indicating a possible recent admixture time. Other segments show shorter haplotype blocks that are non-overlapping indicating a possible admixture date further back in time. These results and the results from DATES lead us to the conclusion that ancestors of PIE060 experienced multiple pulses of admixture.

The updated RFMix graph is now shown in Extended Data Figure SX.

- 87) **Extended Data Fig. S3:** In the legend for panel a, please give more information, i.e. what sample is featured. In the legend for panel b, I would remove “Ratio of female (open circles) and male (colour filled squares) individuals” and move it to the end of the legend.

Thanks. We have added more information for panel a and also changed the order of the sentence in panel b.

- 88) **Extended Data Fig. S4c:** The PIE061 label at the bottom should be changed to PIE060.

Thanks. This is done.

89) **Fig. 3:** For consistency/clarity, I would use either Ukraine CA or Ukraine Eneolithic throughout the paper, but not both.

We agree and have changed it in every instance to Ukraine Eneolithic.

90) **Fig. 3:** It would be helpful for the reader if the authors could use a different colour scheme in the proximal modelling because the current colours are very similar to each other (for instance SEE1, PTK CA and CA merge 6).

We agree and would have done so in the original plot. Fortunately, the re-analysis of the new data resulted in a much simpler version of Fig. 3.

91) **299:** Is this true? I see many cases with $|Z| > 3$ within the last two panels in Extended Data Fig. S6, indicating that the Ukraine Eneolithic groups are not cladal with Steppe Eneolithic/Maykop groups to the exclusion of HG groups.

The mentioned test is not part of the revised manuscript anymore. With our improved dataset and the helpful feedback from the reviewers' we were able to perform more informative tests which we have included in the main text as well as in the Supplementary Data O.

Since all Ukraine Eneolithic groups are symmetrically related with Steppe Eneolithic when compared to WHG (Supplementary Data M, Extended Data Fig. S7) but not when compared to Caucasus Eneolithic/Maykop we only use f_4 (Caucasus Eneolithic/Maykop, Ukraine Eneolithic; HGs, Mbuti) to narrow down potential HG proxies.

92) **321:** In Extended Data Fig. S8, I do not see any model with ~10% of Steppe Eneolithic/Maykop ancestry for any of the three groups (MAJ, KTL_A and USV), the lowest is ~25%.

This figure does not exist anymore due to an improved dataset and revised qpAdm modelling approach. The main results of the qpAdm modelling are now shown in Fig. 3.

93) **403:** What does pre-M269 mean here? There are two Ukraine Eneolithic males with R1b, but these are low resolution calls with unknown subtypes. Using pre-M269 could be confusing as some people could understand that these males are ancestral for M269, when in fact we do not know.

It is true that these Y-haplogroups were only called to the resolution of R1b and R1b1 for KTL005 and MAJ009, respectively. What we did not make clear is that both shallow calls had no further derived SNP calls, but both yielded ancestral SNP calls for the pre-M269 branch (P297) and the M269 branch, respectively, meaning that both are definitely pre-M269. We have amended the Y-haplogroup calls to be R1b/M343(xP297) and R1b1/L754(xM269) to better reflect this and to follow the most widely accepted nomenclature. We also noticed a typo in the calls, with MAJ009 accidentally labelled as R1b1a2a2, even though the terminal SNP was correct.

We have added these details to the results section of the Ukraine Eneolithic individuals, so that we safely refer to 'pre-M269' in the Discussion.

94) **415:** In the Lazaridis et al. Science 2022 paper they argue that the farmer-related ancestry in Yamnaya is not accompanied by WHG-related ancestry, and thus it must have come from south of the Caucasus, not from mainland Europe. Here, the authors support an origin in eastern Europe, so it seems a good place to discuss both hypotheses and to explain why one of them is preferred over the other.

Yes, we are aware of Lazaridis et al. 2022, which came out after the initial submission of our manuscript. In line with point 4) and the general revisions of our manuscript we are now addressing this point in greater detail in the revised section on the Early Bronze Age.

- 95) **436**: This is true for Britain, but the genetic data do not show a near-complete autosomal genetic turnover in the Iberian Peninsula.

Thank you for pointing this out. Our wording was misleading and we have revised it accordingly, making the difference between the autosomal and the Y-chromosome turnover clearer.

- 96) **939**: I would use "fitting" models rather than "significant" models, because $p > 0.05$ are considered non-significant p-values.

The plot does not exist anymore since we were able to narrow down possible models and were able to plot everything in Fig. 3. in the main text.

- 97) **Sup Data C**: The title has a couple of typos.

Thanks for the catch. The typo is corrected.

Reviewer Reports on the First Revision:

Referees' comments:

Referee #1 (Remarks to the Author):

A

Thank you very much. I appreciate your careful and differentiated argumentation in your paper and have tried to make this clear in various places in the review. My comment - links between archaeogenetic results and technological innovations - is based on the fact that you explicitly refer to this point in the last sentence of your summary, but then write only very cautiously about it in the 'Discussion'. I can understand your reasons to do so, but the summary raises certain expectations, at least for me. However, I don't think this is essential and I completely agree if you don't want to change this point.

B

No comments or questions in the previous review that would have necessitated changes.

C

No comments or questions in the previous review that would have necessitated changes.

D

No comments or questions in the previous review that would have necessitated changes.

E

Thank you for the numerous additions you have made to the text as well as your helpful clarifications regarding my comments. I have no further questions or comments.

F

Thank you for the changes. I have no further comments.

G

Thank you for the changes. I have no further comments.

H

I already stated in the previous review, that this is a well-structured and argued paper. But it improved by the revision.

Referee #2 (Remarks to the Author):

I'd like to thank the authors for their effort in revising the manuscript and addressing my points. I have two main and some minor suggestions.

In general, I'd recommend revising the text to increase accessibility to a wider audience. For instance, instead of saying "all EBA individuals are significantly negative for WHG and EHG/WSHG when compared to Caucasus Eneolithic/Maykop" (L440), it would be easier to follow "WHG and EHG/WSHG share a higher genetic affinity to all EBA individuals compared to Caucasus Eneolithic/Maykop". I think such a revision could significantly benefit the manuscript. Likewise, when reporting analyses, explaining the motivation and logic clearly would be helpful for the reader less familiar with the methodology and the populations and period in question. Additional

information on the populations could also help the reader; such as "Ukraine N" actually representing Ukraine foragers.

The admixture analysis on PIE060 and the conclusion about "repeated admixture" are very interesting. But I'd like to follow up on Rev 3's point on the maternal and paternal WHG-related segments overlapping (even if not at the same exact boundaries) at unexpectedly high rates (>50%) in Ext Data Fig 3.

An explanation / discussion of this pattern is needed, I believe. The only biological explanation I could imagine would be the individual being inbred, but whether the identified ROH levels could explain the segment overlap is not obvious (Ext Data Fig 4). Could the authors perhaps test the non-WHG tracts for ROH? In any case, if the reason for the overlap is not clarified, I'd suggest removing this result from the main text (the authors could report the analysis in the Supplement with the discussion here).

Suggestions and minor points:

Fig 1: Uses Anatolia_N instead of Turkey_N.

Fig 1: Adding key events, e.g. Yunatsite destruction, to the timeline could be nice.

Fig 1: The blue diamond of Ukr Eneol - should it not belong to EBA, given the timeline?

Fig 1: Could the authors also show the Ozera individual separately in the PCA and the timeline? Reporting its location would also be helpful.

Fig 2: Adding the date ranges (as in Fig. 1c) would be helpful.

Fig 3: CHG missing in legend.

Fig 3: Marking both in the legend and the text that some models did not fit well would be helpful. The authors should at least indicate cases with $p < 0.05$ also in panel a.

Fig 3: Mention that these are results using allSNPs:NO in the legend.

Fig 3: Colors are difficult to trace.

qpAdm results: reporting overlapping SNP numbers would be helpful.

L127: "reference" a leftover?

L148: You could also report the range and average of SNP coverages in the sample, and also the number of genomes you ran imputation on. This is useful information, I think.

L179: "we were then used"

L179: Perhaps mention already here that you are using apparent symmetry wrt HGs as an indication of being "most genetically close". It is stated 2 lines further, but my suggestion would help the reader follow your approach, I think.

L181: saying "apparently symmetrically related" or something that indicates uncertainty instead of just "symmetrically related" would be more prudent.

L192: Here something like "To investigate this further" would make more sense than "Thus", I think.

L192-204: What is done here (the logic of each analysis and the outcomes) could be better explained, I felt. I'd suggest the following steps:

- A possible HG effect is detected.
- qpAdm with HG and Turkey_N sources supports this.
- But this change could have happened already earlier. Perhaps earlier Neol groups already carried this ancestry?
- Run the f4-based grouping.
- Find support in proximal analysis.

L284: "However" might not be necessary. The authors might directly say "we tested for a temporal effect, and found none", which would be easier to follow.

Extended Data Fig 4: panel b: why are n_IBD values not integers?

Ext Dat Fig 6: 3rd panel: "Russia Steppe Eneolithic" repeated twice in D formula. Different Russia Steppe Eneolithic individuals used as X and Y? Explanation would be helpful.

L306: "show gene flow from all HG groups" - gene flow being only one explanation for asymmetry, better not formulate this result in this way, but e.g. say "higher affinity to all HG groups" or something along those lines.

L328-9: Should be Data P not O, and KTL003 instead of KTL001.

L334: "scenarios involving potential gene flow from the North Caucasus would require an additional source carrying WHG-like ancestry." - I felt this is confusing to the reader, as the background is not explained yet - i.e. the archaeological connection with the Maykop and the hypothesis of farmer ancestry being contributed via N Caucasus instead of SEE. Please revise this section for sake of clarity.

L340: "For individuals KTL001 and KTL007, and KTL003, KTL006 and KTL008" - reads a bit awkward.

Data J: UBK and USV are both used in the manuscript - might be better to stick to a single one (e.g. in Ext Dat Fig 4 and Data J)

L423: parenthesis missing.

L530: "Gene flow from the contact zones to the steppe could also explain the small amounts of farmer-related ancestry in the emerging Yamnaya pastoralists, which differentiates them from the Steppe Eneolithic substrate" - this is a nice point, based on the result in Extended Data Fig. S10, I believe. But I'm wondering if an alternative scenario, where the Turkey_N affinity to EBA relative to Steppe Eneolithic may be explained by Maykop ancestry instead of SEE CA-related ancestry, could also explain the picture.

L927: Parenthesis missing. Also, I think you should reference Mathieson et al 2018 here?

L980: More information on the imputation process, e.g. whether any filtering or MAF threshold was applied, would be useful.

Supp Data S: The cell "Selected proximal qpAdm modelling for Ukrain Eneolithic groups and individuals with a rotating model approach and the parameter 'AllSNPs: NO'" probably should be 'AllSNPs: YES'

L962: Supp Data table names appear mixed up.

Both 134 and 135 new genomes are reported through the text. It would be good to clarify.

Supplement:

p48: I believe $|Z| \geq 1$ should be $|Z| < 1$, and $(|Z| < 1)$ should be $(|Z| > 1)$. Plus wrong ref to Supplementary Data C.

p49: wrong ref to Supplementary Data G

Grammatical / spelling errors:

"Since there was initially little to no admixture between local hunter-gatherer groups and expanding Anatolian Neolithic farmers, and this early farmer-associated ancestry is still prevailing at the beginning of the Copper Age, approximately 1000-2000 years after the first farmers arrived."

"made up off"

Referee #3 (Remarks to the Author):

The authors have addressed most of my comments and have performed a heavy restructuring of the ancestry modelling. This section and associated figures are now simpler and much easier to follow, especially for non-experts.

Although this is an excellent manuscript, I still think that it might not represent a significant breakthrough of the degree required by Nature, nor would they appeal to a wide enough audience to deserve a Nature publication.

-Comment 73. The authors reanalysed the data from the Immel et al study, concluding that the

signal in the Cucuteni Trypillia Culture individuals from that paper is different as compared to the Ukraine Eneolithic newly reported samples, with the signal in the former being more EHG-like and the latter being more Yamnaya-like.

I am convinced that this is actually the case, but to be fair this was already presented as one of the possible scenarios in the Immel et al study, for instance in the abstract "Three of the specimens also showed considerable amounts of steppe-related ancestry, suggesting influx into the CTC gene-pool from people affiliated with, for instance, the Ukraine Mesolithic" or discussion "One likely source population that could have introduced the steppe ancestry component into the CTC gene-pool might have been individuals associated with the eastern Eurasian Mesolithic, e.g. the Ukraine Mesolithic people, Eastern hunter-gatherers or even later-dating Yamnaya steppe pastoralists "

(Note that in Immel et al, "steppe ancestry" refers to geography, not to a specific mixture of ancestries such as EHG+CHG).

Thus, although the authors state that "One of the main claims of the Immel et al study was an early contact event between early farmers and pastoralist groups from the steppe regions", other scenarios not involving contact with Yamnaya pastoralist groups were also considered in that paper. In summary, I agree that the findings for KTL A, MAJ and USV are new and demonstrate a stronger genetic signal from the Steppe of a different nature than the Cucuteni Trypillia samples, but they both provide evidence of early contacts between southeastern Europe and the Steppe.

-Comment 86. Yes, longer haplotype blocks indicate a possible recent admixture event, and shorter blocks more ancient admixture event, but my initial worry (which still persists) was that recombination points are very correlated in the maternal and paternal copies. One expects to find some correlation because recombination is not random along the genome, but I cannot think of a reason that would make most of the HG tracts to have the same or very similar start -end coordinates in the maternal and paternal copies.

-Figure 3. Panels a,b,c. The colour legend for CHG is missing. I am not sure if the bars accurately represent the distal qpADM ancestry proportions in the Supplementary Tables, because none of the a,b,c panels have any bar for WSHG. For instance, in Supplementary Table R, one of the two models for KTL001 has 18% WSHG ancestry but this is not shown in Fig. 3b.

-Related to the above, in line 341. "For individuals KTL001 and KTL007, and KTL003, KTL006 and KTL008 we find support for two alternative models: one in line with the visible cline in PCA (Turkey_N+EHG+CHG), and an alternative model (Turkey_N+CHG+WHG), to the exclusion of EHG ancestry." I might not be looking at the right table, but in Supplementary Table R, the two alternative models for KTL001 are Turkey_N+EHG+CHG+WHG and Turkey_N+CHG+WHG+WSHG.

-Line 382. I believe it should be Supplementary Table S.

-Line 463. Typo "of of".

Author Rebuttals to First Revision:

Point-by-point Response to Reviewer Comments for Nature manuscript 2022-08-12988A

"Early contact between late farming and pastoralist societies in southeastern Europe"

We received additional comments from two of the three reviewers, which we address below (our responses in blue). Changes made to the revised manuscript files are highlighted in yellow.

Referees' comments:

Referee #1 (Remarks to the Author):

A

Thank you very much. I appreciate your careful and differentiated argumentation in your paper and have tried to make this clear in various places in the review. My comment - links between archaeogenetic results and technological innovations - is based on the fact that you explicitly refer to this point in the last sentence of your summary, but then write only very cautiously about it in the 'Discussion'. I can understand your reasons to do so, but the summary raises certain expectations, at least for me. However, I don't think this is essential and I completely agree if you don't want to change this point.

Many thanks for your understanding.

We provide information about the technological innovations and archaeological assemblages in the contact zones in the introduction (lines 97-114) and the Supplementary Information. Throughout the main text, we explicitly test, and show evidence for, contact and genetic admixture between late farming and pastoralist communities, which was in all likelihood facilitated through these contact zones during time periods such as the Eneolithic of the 4th millennium BCE. However, while there is clear evidence for exchange at the cultural and biological level, we do not (yet) have the power of resolution nor temporal precision to state which technology or knowledge was transferred at which time exactly. We feel that the discussion, esp. lines 511-531, covers this aspect well and provides balanced information about the broader time frames, the dynamics and the directionality of events.

B

No comments or questions in the previous review that would have necessitated changes.

C

No comments or questions in the previous review that would have necessitated changes.

D

No comments or questions in the previous review that would have necessitated changes.

E

Thank you for the numerous additions you have made to the text as well as your helpful clarifications regarding my comments. I have no further questions or comments.

F

Thank you for the changes. I have no further comments.

G

Thank you for the changes. I have no further comments.

H

I already stated in the previous review, that this is a well-structured and argued paper. But it improved by the revision.

We thank reviewer 1 for the positive evaluation.

Referee #2 (Remarks to the Author):

I'd like to thank the authors for their effort in revising the manuscript and addressing my points. I have two main and some minor suggestions.

In general, I'd recommend revising the text to increase accessibility to a wider audience. For instance, instead of saying "all EBA individuals are significantly negative for WHG and EHG/WSHG when compared to Caucasus Eneolithic/Maykop" (L440), it would be easier to follow "WHG and EHG/WSHG share a higher genetic affinity to all EBA individuals compared to Caucasus Eneolithic/Maykop". I think such a revision could significantly benefit the manuscript. Likewise, when reporting analyses, explaining the motivation and logic clearly would be helpful for the reader less familiar with the methodology and the populations and period in question. Additional information on the populations could also help the reader; such as "Ukraine N" actually representing Ukraine foragers.

We thank the reviewer for the suggestion. Across all f_4 -statistics reported in this study we tried to strike a balance between reporting the four-population formula (which includes a statement of the motivation in this setting), the actual results (positive, negative or not deviating from 0), and an interpretation of this outcome (e.g., shared drift/genetic affinity of pop X with pop Y), which taken together explains both the motivation AND the principle/logic of these tests. Removing the results and just reporting the outcome ("...share a higher genetic affinity to all EBA individuals") would in our opinion not help in conveying the underlying principle, but would indeed be easier to follow once the principle was explained properly. We carefully went through the text and double-checked whether at the first mention of f_4 -statistics, the motivation and principle is explained appropriately, so that the outcome of these tests are easier to follow. Once explained, we only report the actual finding. We hope that this has improved readability and accessibility to the reader.

Regarding the populations and groups used for comparison, we are aware of the challenges to keep track of the published data and the naming and renaming of the latter, which often requires encyclopedic knowledge, similar to archaeological cultures and time periods. The length restrictions unfortunately do not leave much room for additional information on all groups used in this study, and we therefore refer respectfully to the primary literature, which first reported this data (here Mathieson et al. 2018). We are afraid that renaming the groups for the purpose of our study would only add further confusion. However, we provided extra context to the main text wherever we could.

The admixture analysis on PIE060 and the conclusion about "repeated admixture" are very interesting. But I'd like to follow up on Rev 3's point on the maternal and paternal WHG-related segments overlapping (even if not at the same exact boundaries) at unexpectedly high rates (>50%) in Ext Data Fig 3.

An explanation / discussion of this pattern is needed, I believe. The only biological explanation I could imagine would be the individual being inbred, but whether the identified ROH levels could explain the segment overlap is not obvious (Ext Data Fig 4). Could the authors perhaps test the non-WHG tracts for ROH? In any case, if the reason for the overlap is not clarified, I'd suggest removing this result from the main text (the authors could report the analysis in the Supplement with the discussion here).

We thank both reviewers for the thorough inspection of our work. We fully agree about his point and have investigated the situation further. RFMix was clearly developed for high-coverage modern-day population data (Maples *et al.* 2013, *The American Journal of Human Genetics*). As the method requires phasing, which can only be done reliably with high-quality data, our low coverage ancient DNA data with high missingness likely introduces a large number of switch errors, which result in the biased pattern, while maintaining the modeled ancestry proportions. In fact, using ancient DNA data as target AND sources makes the situation even worse.

Exploring the method with a few additional tests that were run with the same parameters as our analysis, we gained further insights: below we provide two anonymized individuals from a Medieval pedigree who represent mixed European and East Asians ancestries, and using high-quality modern-day East Asian (EAS) and Europeans (EUR) as reference data. Since the paternal segments of EAS ancestry rarely overlap in both individuals tells us that this method can be applied in exceptional circumstances, that is when two genetically very distinct groups of high-quality modern data are used as reference.

However, since both conditions cannot be met in our case, we decided to exclude this result from the manuscript. Instead, we retain the DATES estimate. By assessing the developed but flat decay curve (now included in Extended Data Fig. 3a instead of the RFMix graph) together with the context and the overall amount of HG ancestry, we conclude that the admixture must have happened relatively recently. Our initial motivation was to find independent evidence for the recent admixture, but we realise that our data is not of sufficient quality to achieve this through RFMix analysis.

We have thus changed the part in the manuscript about the dating of the admixture event of PIE060 accordingly.

Suggestions and minor points:

Fig 1: Uses Anatolia_N instead of Turkey_N.

Thank you for the catch. We have changed the name to Turkey_N.

Fig 1: Adding key events, e.g. Yunatsite destruction, to the timeline could be nice.

Thank you for the suggestion. We would rather keep Fig. 1 without key events since it already contains a lot of information. The Figure is already structured by three main temporal ranges, and while the destruction horizon at Yunatsite clearly marks an event within the period of the demise of the Copper Age settlements it is very specific to this Tell site. The number of ^{14}C dates that end around 4200 BCE marks the demise of the Copper Age and is visible from the chronology in panel a. To make a more obvious connection between the demise of the Tell sites and a lack of individuals from the subsequent time window we now refer to Fig. 1a in the main text in this sentence: "Consequently, the roughly simultaneous abandonment of the numerous tell settlements and cemeteries such as Varna around 4250/4200 BCE appears enigmatic (Fig. 1a)."

Fig 1: The blue diamond of Ukr Eneol - should it not belong to EBA, given the timeline?

Yes, this published individual dates to the beginning of the early Bronze Age. The label comes from the original publication and has since been changed to 'Ukraine_Eneolithic_oHG' in the latest annotation file of the public data repositories (e.g. AADR v54.1.p1; <https://reich.hms.harvard.edu/allen-ancient-dna-resource-aadr-downloadable-genotypes-present-day-and-ancient-dna-data> or POSEIDON; <https://poseidon-framework.github.io/#/>). Hence, we would prefer to use the name as it is, and have added the suffix _oHG where needed (PCA, Supplementary Data). More information can be found in Supplementary Data B. We are aware that the naming of samples in the field of ancient DNA is somewhat inconsistent due to initial naming of samples.

Fig 1: Could the authors also show the Ozera individual separately in the PCA and the timeline? Reporting its location would also be helpful.

We added the location of Ozera to the map for the early Bronze Age, its date to the chronology in Figure 1 and gave the data point a different symbol in the PCA and thus an extra mention in the legend.

Fig 2: Adding the date ranges (as in Fig. 1c) would be helpful.

We agree that it would be helpful to the reader to see the date ranges again. We added them to the figure.

Fig 3: CHG missing in legend.

Thank you. We added CHG to the figure legend.

Fig 3: Marking both in the legend and the text that some models did not fit well would be helpful. The authors should at least indicate cases with $p < 0.05$ also in panel a.

Thank you. We changed the labels for poorly-fitting models to *Italics* and also changed the colour transparency of the ancestry proportions to reflect this.

Fig 3: Mention that these are results using allSNPs:NO in the legend.

We added the following sentence to the legend: "All results shown here were obtained from analyses run with the parameter 'allSNPs: NO'."

Fig 3: Colors are difficult to trace.

The colours in qpAdm reflect the colours used for the respective groups in PCA. We agree that the colours used for the Neolithic and the WHG groups were very similar and thus have added symbols to better distinguish between the different groups plotted.

qpAdm results: reporting overlapping SNP numbers would be helpful.

Thank you for this suggestion. We agree that the number of overlapping SNPs would be helpful but we think it would add another information to the figure which is already conveying a lot of information. Therefore, we added the number of overlapping SNPs for allSNPs NO to the Supplementary Data G, H, R, S, X and Y and made a reference to these in the figure caption.

L127: "reference" a leftover?

Yes, how embarrassing ;)

L148: You could also report the range and average of SNP coverages in the sample, and also the number of genomes you ran imputation on. This is useful information, I think.

Thank you for the suggestion. We agree, and added the following sentence: "The number of obtained SNPs ranges from ~61,000 up to ~947,000 with an average coverage between 0.01X and 3.4X. Additionally, we used a cut-off of 400,000 SNPs for hapROH and imputation and filtered more stringently using >550,000 SNPs for IBD analyses (Supplementary Data A, Materials and Methods)."

L179: "we were then used"

Thank you. We removed 'we'.

L179: Perhaps mention already here that you are using apparent symmetry wrt HGs as an indication of being "most genetically close". It is stated 2 lines further, but my suggestion would help the reader follow your approach, I think.

We agree that it would be helpful for the reader if we stated our motivation early on and thus changed the sentence as follows: "We used f_4 -statistics of the form $f_4(\text{test}, \text{PIE039}; \text{HG})$,

Mbuti), where 'test' are different Neolithic groups, to identify the most genetically close groups with respect to an underlying HG ancestry in Neolithic individuals, which were then used as local proxies for quantitative ancestry modelling."

L181: saying "apparently symmetrically related" or something that indicates uncertainty instead of just "symmetrically related" would be more prudent.

Symmetrically related refers to the f_4 -statistic being non-significant (in this case $|Z| \leq 1$). This is an observation that is independent of any interpretation and therefore, 'apparently' would be an interpretation to the result which is not used here. In the sentence before we state that we are looking for proxies which these two groups are for PIE039 based on a symmetrical relatedness between the groups.

We understand the point made and for clarity we changed the sentence to read: "Here, we found Hungary_LN_Sopot and Malak Preslavets N to be symmetrically related to PIE039 with respect to all HG comparisons ($|Z| \leq 1$) and thus combined them into the group SEE 1 which is used as a proxy." to create a direct connection with the sentence before.

L192: Here something like "To investigate this further" would make more sense than "Thus", I think.

Thank you for the advice. We agree and have changed the structure of this section slightly which integrates the sentence differently.

L192-204: What is done here (the logic of each analysis and the outcomes) could be better explained, I felt. I'd suggest the following steps:

- A possible HG effect is detected.
- qpAdm with HG and Turkey_N sources supports this.
- But this change could have happened already earlier. Perhaps earlier Neol groups already carried this ancestry?
- Run the f_4 -based grouping.
- Find support in proximal analysis.

We thank the reviewer for this nice idea and we integrated it into the main text and therefore changed the structure of this part according to the reviewer's suggestion.

L284: "However" might not be necessary. The authors might directly say "we tested for a temporal effect, and found none", which would be easier to follow.

Thank you for the suggestion. We can see that it would be easier to understand our point and changed the sentence as follows: "We tested for a correlation between positions of the Ukraine Eneolithic individuals in PC2 and their ^{14}C dates and found none (Spearman's $\rho=0.113$, $p=0.6656$)."

Extended Data Fig 4: panel b: why are n_{IBD} values not integers?

In both panels for Fig. S4, these values are the site-wise averages of the n_{IBD} and sum_{IBD} values, and the mean of a sample of integers need not be an integer itself. The y-axis labels have been updated. Thanks for helping us to provide clarity here.

Ext Dat Fig 6: 3rd panel: "Russia Steppe Eneolithic" repeated twice in D formula. Different Russia Steppe Eneolithic individuals used as X and Y? Explanation would be helpful.

Thank you for the catch. It was supposed to be the comparison between Russia Steppe Eneolithic and Russia Khvalynsk Eneolithic. The formula in Extended Data Fig. S6 is changed accordingly.

L306: "show gene flow from all HG groups" - gene flow being only one explanation for asymmetry, better not formulate this result in this way, but e.g. say "higher affinity to all HG groups" or something along those lines.

Thank you for the suggestion. We changed this part of the sentences as follows. "First, compared to Turkey_N, Ukraine Eneolithic individuals show excess affinity to all HG groups, ..."

L328-9: Should be Data P not O, and KTL003 instead of KTL001.

Thank you. We changed the reference to the Supplementary Data P and after double checking we removed KTL001 and did not change it to KTL003 since KTL003 is significant when compared to all groups except Tell Kurdu, and is already mentioned in the first part of the sentence.

L334: "scenarios involving potential gene flow from the North Caucasus would require an additional source carrying WHG-like ancestry." - I felt this is confusing to the reader, as the background is not explained yet - i.e. the archaeological connection with the Maykop and the hypothesis of farmer ancestry being contributed via N Caucasus instead of SEE. Please revise this section for sake of clarity.

Thank you. We understand the concern raised and have revised the sentence which now reads: "This affinity towards WHG/EHG is absent when Steppe Eneolithic is used (Supplementary Data N), which implies that scenarios involving potential gene flow from the Caucasus would require an additional source carrying WHG-/EHG-like ancestry as this ancestry is not sufficiently represented by SEE CA or Caucasus Maykop groups."

The background and rationale for these tests is provided in the beginning of the Eneolithic section: "Based on cultural influences which link the northern Black Sea via the steppe belt to the North Caucasus region⁵⁸⁻⁶⁰, we also test for potential influence of North Caucasian cultural groups, such as Maykop."

L340: "For individuals KTL001 and KTL007, and KTL003, KTL006 and KTL008" - reads a bit awkward.

We agree that this sentence was confusing and not conveying what we actually intended to say. The sentence now reads: "Using distal qpAdm modelling we can confirm a four-way mixture with Turkey_N, EHG, CHG and WHG for KTL001, KTL007, MAJ and USV (Fig. 3b, Supplementary Data R). For individuals KTL003, KTL006 and KTL008 we find support for an alternative model in line with the visible cline in PCA (Turkey_N+EHG+CHG), while the three

KTL_B individuals can only be modelled with Turkey_N (~60%), CHG (~28%), and WHG (~12%) ancestry.””

Data J: UBK and USV are both used in the manuscript - might be better to stick to a single one (e.g. in Ext Dat Fig 4 and Data J)

We agree that it is unfortunate that we have two different abbreviations for the same site. Yet, we use UBK only when we work with the actual data, like in Extended Data Fig. S5 for example. We add a note in Supplementary Data A explaining that we have two different names for these samples. It was only made clear to us AFTER we had processed these samples that the individuals belong to the very same site. The fact that all processing steps contain the different names (from the first sample tube to the final fastq and bam files), make a renaming after the full processing cumbersome and will only lead to complication with our LIMS (Laboratory Information Management System). That is the reason that we are stuck with the two different site IDs and have added aliases to indicate the main site name. We hope that, since we added this to the Supplementary Data, it will be clear to the reader that we are speaking about the same site. To avoid further confusion we have added additional information at the beginning of the second part (Genetic influence from the east during the Eneolithic in Ukraine) where we now write (revised version L337): “The five individuals from Majaky (MAJ), of which four tentatively date to the early Eneolithic (~4500-4000 BCE) and one to the late Eneolithic (~4000-3400 BCE), are genetically more homogeneous and fall together with the four individuals from the late Eneolithic Usatovo type-site (USV/UBK, Supplementary Data A) in the middle of the ‘Kartal cline’.”

L423: parenthesis missing.

Thank you for the catch. We added the parenthesis.

L530: "Gene flow from the contact zones to the steppe could also explain the small amounts of farmer-related ancestry in the emerging Yamnaya pastoralists, which differentiates them from the Steppe Eneolithic substrate" - this is a nice point, based on the result in Extended Data Fig. S10, I believe. But I'm wondering if an alternative scenario, where the Turkey_N affinity to EBA relative to Steppe Eneolithic may be explained by Maykop ancestry instead of SEE CA-related ancestry, could also explain the picture.

Yes, indeed. The point about trade, exchange and potential gene flow with farming communities in the contact zones in the west (northwestern Black Sea region) and the south of the steppe zone (Caucasus) is highlighted in the introduction, which formulates a central aim of our study. In fact, in both results sections about the Eneolithic and the Early Bronze Age we go to great lengths to explore both possible scenarios by using source populations from both regions that could equally likely account for the affinity to early farmer-related ancestry in Ukraine Eneolithic as well as Early Bronze Age steppe pastoralists. The ancestry modelling section about the Eneolithic even contains a competing model test, with which we attempt to distinguish between the highly similar sources.

We now include a rationale for these tests at various points, hoping this will improve the accessibility.

L927: Parenthesis missing. Also, I think you should reference Mathieson et al 2018 here?

Thank you for the catch. We added the parenthesis and the reference suggested.

L980: More information on the imputation process, e.g. whether any filtering or MAF threshold was applied, would be useful.

Thank you for the suggestion. We added the following sentences to the Material and Methods section about Imputation: "Samples with more than 0.5x coverage on the 1240k positions (~550k SNPs) after imputation were included in the analysis. No MAF filtering was performed, since only 1240k SNP positions were retained after imputation."

Supp Data S: The cell "Selected proximal qpAdm modelling for Ukrain Eneolithic groups and individuals with a rotating model approach and the parameter 'AllSNPs: NO'" probably should be 'AllSNPs: YES'

Thank you. We changed it to 'YES'.

L962: Supp Data table names appear mixed up.

Thank you. We changed them to the correct Supplementary Data (G, H, R, S, X and Y).

Both 134 and 135 new genomes are reported through the text. It would be good to clarify.

Thank you for pointing that out to us. This must have been a leftover from the previous version. The correct number is 135.

Supplement:

p48: I believe $|Z| \geq 1$ should be $|Z| < 1$, and $(|Z| < 1)$ should be $(|Z| > 1)$. Plus wrong ref to Supplementary Data C.

Thank you. We changed the symbols to the correct ones and also updated the references to the correct Supplementary Data.

p49: wrong ref to Supplementary Data G

Yes, indeed. The reference is now updated to Supplementary Data F.

Grammatical / spelling errors:

"Since there was initially little to no admixture between local hunter-gatherer groups and expanding Anatolian Neolithic farmers, and this early farmer-associated ancestry is still prevailing at the beginning of the Copper Age, approximately 1000-2000 years after the first farmers arrived."

Thank you for the catch. We changed the sentence to: "Since there was initially little to no admixture between local hunter-gatherer groups and expanding Anatolian Neolithic farmers, this early farmer-associated ancestry is still prevailing at the beginning of the Copper Age,

approximately 1000-2000 years after the first farmers arrived.” , which now makes sense and we also corrected the spelling mistake.

"made up off"

Corrected.

Referee #3 (Remarks to the Author):

The authors have addressed most of my comments and have performed a heavy restructuring of the ancestry modelling. This section and associated figures are now simpler and much easier to follow, especially for non-experts.

Although this is an excellent manuscript, I still think that it might not represent a significant breakthrough of the degree required by Nature, nor would they appeal to a wide enough audience to deserve a Nature publication.

Thank you for the feedback.

-Comment 73. The authors reanalysed the data from the Immel et al study, concluding that the signal in the Cucuteni Trypillia Culture individuals from that paper is different as compared to the Ukraine Eneolithic newly reported samples, with the signal in the former being more EHG-like and the latter being more Yamnaya-like.

This is correct, albeit with a critical chronological nuance: we clearly state that the Ukraine Eneolithic individuals carry a Steppe Eneolithic-like component, consisting of EHG+CHG-like ancestry. This component is later also found in Yamnaya-associated individuals as part of a different blend of ancestries as we elaborate on, but - importantly - it is not a direct contribution of Yamnaya.

I am convinced that this is actually the case, but to be fair this was already presented as one of the possible scenarios in the Immel et al study, for instance in the abstract “Three of the specimens also showed considerable amounts of steppe-related ancestry, suggesting influx into the CTC gene-pool from people affiliated with, for instance, the Ukraine Mesolithic” or discussion “One likely source population that could have introduced the steppe ancestry component into the CTC gene-pool might have been individuals associated with the eastern Eurasian Mesolithic, e.g. the Ukraine Mesolithic people, Eastern hunter-gatherers or even later-dating Yamnaya steppe pastoralists ”

(Note that in Immel et al, "steppe ancestry" refers to geography, not to a specific mixture of ancestries such as EHG+CHG).

It is rather unfortunate that the phrasing in the abstract of Immel et al. 2020, Scientific Reports, is not clear, and - as in this case - rather misleading. “Steppe ancestry” was defined as a mix of EHG and CHG ancestries, and was always understood as such by the scientific community. (Note that in addition to EHG ancestry a ‘southern ancestry type’ from the Caucasus region was suggested as a modern-day proxy in 2015 by Haak et al., Nature,

while in the same year Jones et al. 2015, Nat Communications, reported with CHG ancestry a more fitting proxy.)

Importantly, "Steppe ancestry" never referred to geography and this re-attribution appears to us like an attempt to shift goalposts.

In fact, the increase of hunter-gatherer ancestry following the initial wave of expanding farming communities was broadly described as 'hunter-gatherer resurgence' and has been reported in many regions across Europe (Haak et al. 2015, Mathieson et al. 2015/2018, Lipson et al. 2017, Rivollat et al. 2020, and many others). In eastern Europe the hunter-gatherer contribution of this 'resurgence' is increasingly more of the EHG type or a mixed form of WHG/EHG ancestry, such as the one found in the Iron Gates, the Ukrainian "Mesolithic/Neolithic", and thus likely also north of the Carpathians. We notice a similar but very subtle signal of this assimilation process between farming-associated and late hunter-gatherer-fisher communities in the studied Chalcolithic communities, in particular at Varna. Moreover, the fact that we document a continued increase in HG-ancestry in the Early Bronze Age individuals from Yunatsite shows that this is an independent and ongoing process, but which is separate from the clearly distinguishable influence of Steppe Eneolithic-related ancestry we report in the northwestern Pontic region.

Thus, although the authors state that "One of the main claims of the Immel et al study was an early contact event between early farmers and pastoralist groups from the steppe regions", other scenarios not involving contact with Yamnaya pastoralist groups were also considered in that paper.

In summary, I agree that the findings for KTL A, MAJ and USV are new and demonstrate a stronger genetic signal from the Steppe of a different nature than the Cucuteni Trypillia samples, but they both provide evidence of early contacts between southeastern Europe and the Steppe.

Many thanks for the comment and the acknowledgment of the new findings.

-Comment 86. Yes, longer haplotype blocks indicate a possible recent admixture event, and shorter blocks more ancient admixture event, but my initial worry (which still persists) was that recombination points are very correlated in the maternal and paternal copies. One expects to find some correlation because recombination is not random along the genome, but I cannot think of a reason that would make most of the HG tracts to have the same or very similar start -end coordinates in the maternal and paternal copies.

We thank reviewer 3 for this comment and we would like to point to our response to reviewer 2, who had raised the same issue.

-Figure 3. Panels a,b,c. The colour legend for CHG is missing. I am not sure if the bars accurately represent the distal qpADM ancestry proportions in the Supplementary Tables, because none of the a,b,c panels have any bar for WSHG. For instance, in Supplementary Table R, one of the two models for KTL001 has 18% WSHG ancestry but this is not shown in Fig. 3b.

Thanks. We have added CHG to the legend.

The reason why we do not plot the results with WSHG in any of the cases is simply for better readability of the plot but we report them as well-fit models in the supplement information where these are highlighted in bold print. The result and interpretation do not change with WSHG, only the proportions of WHG and EHG/WSHG change, since EHG already carries some WHG ancestry, and therefore needs less in a well-fit model compared to a model with only WSHG.

We see that this might be misleading or confusing for the reader and added the following sentences to Supplementary Information 6: "For better readability of the main Figure 3 we decided to reduce the number of well-fit models displayed. However, all well-fit models are marked in 'bold' in their respective Supplementary Data. We only excluded well-fit models from the plot when the result and interpretation of well-fit models were very similar and thus interchangeable (see e.g., distal qpAdm modeling with either EHG or WSHG)."

We also adapted the title of Supplementary Information 6 to fit the information given in this section: "Rationale for plotting and grouping of distal and proximal source population in qpAdm".

Lastly, we removed WSHG from the legend to avoid confusion and added a reference to Supplementary Information 6 to the caption of Fig. 3.

-Related to the above, in line 341. "For individuals KTL001 and KTL007, and KTL003, KTL006 and KTL008 we find support for two alternative models: one in line with the visible cline in PCA (Turkey_N+EHG+CHG), and an alternative model (Turkey_N+CHG+WHG), to the exclusion of EHG ancestry." I might not be looking at the right table, but in Supplementary Table R, the two alternative models for KTL001 are Turkey_N+EHG+CHG+WHG and Turkey_N+CHG+WHG+WSHG.

Yes, this is correct. We corrected that sentence, which now reads: "Using distal qpAdm modeling we can confirm a four-way mixture with Turkey_N, EHG, CHG and WHG for KTL001, KTL007, MAJ and USV (Fig. 3b, Supplementary Data R). For individuals KTL003, KTL006 and KTL008 we find support for an alternative model in line with the visible cline in PCA (Turkey_N+EHG+CHG), while the three KTL_B individuals can only be modeled with Turkey_N (~60%), CHG (~28%), and WHG (~12%) ancestry."

-Line 382. I believe it should be Supplementary Table S.

Thank you. We changed it to Supplementary Data S.

-Line 463. Typo "of of".

Thank you. We deleted one 'of'.

Reviewer Reports on the Second Revision:

Referees' comments:

Referee #2 (Remarks to the Author):

I thank the authors for their careful work and the changes made.

I would only like to mark that it would be commendable if the authors submitted their data right away and thus could open it directly upon acceptance, for the benefit of the community.

Three minor points:

L265: "This indicates possible admixture between CA farmer-related groups and transitional Eneolithic steppe groups, as described in the archaeological record" might better be "This indicates possible admixture between CA farmer-related groups and transitional Eneolithic steppe groups, in line with cultural interactions described in the archaeological record", or something along those lines?

Fig 3d: The use of faint colors for non-significant results should be noted in the legend. If the authors could find another strategy (e.g. using different border colors to distinguish significant and not) might be even better as this is somewhat confusing.

Ref 15 has a typo.

Referee #3 (Remarks to the Author):

I don't have further comments, besides congratulating the authors for engaging with the reviewers' comments and criticisms.

Author Rebuttals to Third Revision:

Referee #2 (Remarks to the Author):

I thank the authors for their careful work and the changes made.

I would only like to mark that it would be commendable if the authors submitted their data right away and thus could open it directly upon acceptance, for the benefit of the community.

Thank you. The data is uploaded to ENA and readily available upon publication.

Three minor points:

L265: "This indicates possible admixture between CA farmer-related groups and transitional Eneolithic steppe groups, as described in the archaeological record" might better be

"This indicates possible admixture between CA farmer-related groups and transitional Eneolithic steppe groups, in line with cultural interactions described in the archaeological record", or something along those lines?

Thanks. We revised the sentence accordingly.

Fig 3d: The use of faint colors for non-significant results should be noted in the legend. If the authors could find another strategy (e.g. using different border colors to distinguish significant and not) might be even better as this is somewhat confusing.

Thanks for spotting this omission. We had indicated rejected models or those with weak support with Italic p values, and now state this and the use of faint colours (now further enhanced) explicitly in the figure caption.

Ref 15 has a typo.

Ref 15 has been removed in the process of reducing references to a limit of 55 for the main text.